# BEYOND BENCHMARKS: TOWARD CAUSALLY FAITHFUL EVALUATION OF LARGE LANGUAGE MODELS

## ABSTRACT

Current large language models (LLMs) evaluations overlook that measured LLM performance is produced on a full evaluation system, including many indispensable components, such as workloads, prompting methods, decoding parameters, and the supporting software–hardware stack. Without an explicit, controlled specification of the evaluation system, attributing performance differences to the model itself is unreliable. Our experiments reveal that uncontrolled testing may lead to accuracy variations of up to 70%. To address this urgent issue, we introduce LLM evaluatology, a principled methodology that reduces the evaluation problem to accurately attributing the outcomes to the effect of the evaluated LLM, which is a high-dimensional causal-attribution problem. Empirical results demonstrate that LLM evaluatology not only enhances interpretability and causal validity, but also yields evaluations that are more robust, reproducible, and trustworthy than prevailing benchmarks.

## 1 INTRODUCTION

Current LLM evaluation practices are fragmented and ad-hoc, spanning standardized test–style benchmarks (Hendrycks et al., 2020; Huang et al., 2023; Rein et al.; Suzgun et al., 2023; AIME, 2025), human preference–based benchmarks (Chiang et al.; OpenCompass, 2025; Xu et al., 2023), and dynamic or continuously refreshed benchmarks (Jain et al.; Jimenez et al.; White et al.; Zhu et al.; Li et al.). Yet all largely treat the model in isolation, neglecting that measured performance arises from the entire evaluation system, including workloads, prompts, decoding, and even the software–hardware stack. In reality, LLM evaluation is inherently a high-dimensional problem, as these interacting components jointly shape outcomes and complicate attribution. As recent studies show, results can vary sharply with dataset artifacts (Long et al., 2024; Liu et al., 2025), prompt formatting (He et al., 2024), decoding strategies (Shi et al., 2024), or annotator biases (Das et al., 2024). But such analyses remain piecemeal, each targeting a single component without quantifying their combined impact or enabling principled attribution. What is missing is a rigorous methodology that disentangles intrinsic model capability from confounding influences and establishes a reliable foundation for evaluation.

Even under a fully specified evaluation system, LLMs differ fundamentally from traditional single-task or deterministic systems such as conventional algorithms or CPUs. For CPUs, workloads in domains like desktop computing or high-performance computing exhibit well-characterized patterns, allowing evaluation to focus on representative hotspots while treating less common cases as secondary. In contrast, LLM workloads are effectively open-ended: each user can define new tasks across languages, domains, and usage styles. Some tasks resemble those seen during training, others require analogical transformation from familiar patterns, and yet others are entirely novel. This diversity eliminates the notion of a single "typical" workload, making isolated evaluation on a few canonical examples insufficient. Here we adopt the term "workload" from CPU benchmarking, using it to denote a question or instance within a benchmark that the LLM is required to solve. In addition, LLMs may produce fluent responses without genuine reasoning or knowledge, so-called hallucinations, meaning that correctly solving one instance does not guarantee mastery of the underlying skill. Consequently, reliable evaluation must consider multiple task variations, from familiar to analogical to novel, in order to disentangle true capability from surface-level correctness. Interpreting performance and attributing capability is therefore both a high-dimensional and a content-sensitive challenge, further amplified by the confounding inherent in the evaluation system.

This paper introduces LLM evaluatology (Fig. 1), a principled methodology for the rigorous evaluation of LLMs based on Evaluatology (Zhan, 2024; Zhan et al., 2024). At its core, we construct a Minimal Evaluation System (MES), which explicitly defines the evaluated object (e.g., standalone LLM or LLM service), the indispensable components influencing performance, and the evaluation conditions (the configuration space formed by admissible settings of these components). By providing a well-defined, controllable system, MES enables systematic exploration of the evaluation configuration space, capturing how different components jointly affect performance and allowing accurate attribution of model capabilities – a solution to the high-dimensional nature of LLM evaluation. To address content sensitivity, we further extend MES into an Augmented MES (A-MES), which transforms existing workloads and generates new instances along semantically related themes. This approach ensures evaluation coverage across three workload layers: workloads the model is likely to have seen, workloads requiring analogical transformation, and entirely novel workloads, thereby mitigating the risks of superficial correctness and hallucination. A-MES offers a structured, reproducible, and automatable framework that disentangles intrinsic model competence from confounding influences while accommodating the diversity and dynamism of real-world user interactions.

Our experiments reveal several important findings. First, by constructing A-MES, we observe that the accuracy of Doubao varies dramatically with configuration, ranging from 0 to 0.8, highlighting the substantial impact of evaluation settings. Notably, Doubao-1.5-pro ranks first under MES but drops to sixth under A-MES, with a significant gap from the top model, indicating limited generalization ability. Within the Qwen series, we find that the smaller model ranks higher under MES but is surpassed by the larger model under A-MES, suggesting that A-MES provides a more faithful reflection of scaling properties. By contrast, DeepSeek-V3 consistently achieves strong accuracies across all MES and A-MES scenarios, demonstrating the strongest robustness among the tested models. Second, leveraging analysis of variance (ANOVA), xgboost, and linear models, we quantify the impact of each component on model accuracy. All components show measurable influence, with Question Format and COT emerging as the most sensitive, followed by max_tokens, Shot, and Multi Turn. Furthermore, models exhibit heterogeneous sensitivity to languages: for example, DeepSeek-V3 is most sensitive to Arabic, where its accuracy reaches the lowest among all languages tested. Finally, we validate that our proposed LLM evaluatology provides the closest approximation to the accuracy ground truth, significantly outperforming traditional single-configuration evaluations in reliability and robustness.

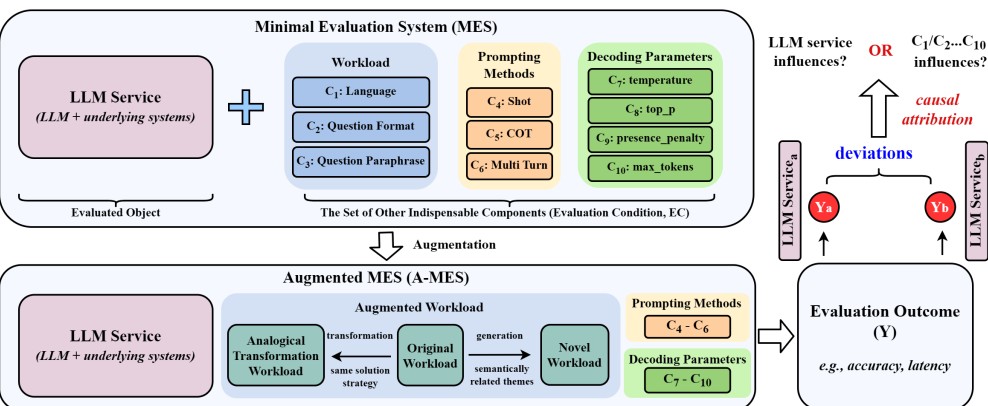

Figure 1: LLM Evaluatology: Measured performance arises from an Augmented Minimal Evaluation System (A-MES), which enables disentangling intrinsic model capability from confounding influences. Here, the evaluation object is defined as the LLM service, comprising the LLM and its underlying systems. When evaluating a standalone LLM, the underlying systems are instead treated as part of the evaluation conditions (EC).

## 2  RELATED WORK

Broadly, existing benchmarks can be grouped into the following three categories. Standardized test–style benchmarks present problems in the form of test questions, with model outputs compared against reference answers. Representative examples include MMLU (Hendrycks et al., 2020) and

its extensions MMLU-Pro (Wang et al., 2024b) and MMLU-Redux (Gema et al., 2025), as well as C-Eval (Huang et al., 2023) and CMMLU (Li et al., 2024) in the Chinese context. GPQA (Rein et al.) targets graduate-level science, while other datasets focus on specific capabilities such as reasoning (BBH (Suzgun et al., 2023), HellaSwag (Zellers et al., 2019), Winogrande (Sakaguchi et al., 2021)), mathematics (GSM8K (Cobbe et al., 2021), MATH (Hendrycks et al.), AIME (AIME, 2025)), coding (HumanEval (Chen et al., 2021), MBPP (Austin et al., 2021), Aider-polyglot (Aider, 2025), MultiPL-E (Cassano et al., 2023)), long-context understanding (L-Eval (An et al., 2024), LongBench (Bai et al., 2024), ∞Bench (Zhang et al., 2024a), HELMET (Yen et al., 2025)), safety (SafetyBench (Zhang et al., 2024b), Toxigen (Hartvigsen et al., 2022)), instruction-following (IFE-val (Zhou et al., 2023), Multi-Challenge (Sirdeshmukh et al., 2025)), and multimodality (MMBench (Liu et al., 2024), MMMU (Yue et al., 2024), MathVista (Lu et al.)).

Human preference–based benchmarks evaluate models in interactive settings, collecting user judgments instead of relying on fixed test sets. Chatbot Arena (Chiang et al.) is the most prominent example, where pairwise votes are aggregated via Elo ratings. CompassArena (OpenCompass, 2025) apply similar designs in the Chinese context.

Dynamic or continuously refreshed benchmarks aim to avoid data contamination by relying on newly released or procedurally generated tasks. Examples include LiveCodeBench (Jain et al.) (recent programming contests), SWE-bench (Jimenez et al.) (GitHub issues and PRs), LiveBench (White et al.) (rolling monthly refresh), DyVal (Zhu et al.) (procedural reasoning via DAGs), and Arena-Hard (Li et al.) (real-time crowdsourced challenges).

Table 1: Evaluation Settings on Different Benchmarks (Lang. = Language, Format = Question Format, Para. = Question Paraphrase, M-turn = Multi Turn, Temp. = temperature, PP = presence_penalty, MaxTok = max_tokens, ori = original, y = yes, n = no)

| Model | Lang. | Format | Para. | Shot | COT | M-turn | Temp. | top_p | PP | MaxTok |
|---|---|---|---|---|---|---|---|---|---|---|
| MMLU | English | ori | n | 0/3/5 | y/n | n | 0.0/0.3/0.5/0.6/0.7 | 0.8/0.95 | 0/1.5 | 1024/8192/32768 |
| AIME | English | ori | n | 0 | y/n | n | 0.0/0.6/0.7 | 0.8/0.95 | 0/1.5 | 8192/32768/38912 |
| GPQA | English | ori | n | 0/5 | y/n | n | 0.4/0.5/0.6/0.7 | 0.8/0.95 | 0/1.5 | 1024/8192/32768 |
| MATH | English | ori | n | 0/8 | y/n | n | 0.0/0.6/0.7 | 0.8/0.95 | 0/1.5 | 8192/32768 |
| SWE-bench | English | ori | n | 0 | y/n | n | 0.0/0.8 | 0.95 | x | 8192/16384 |
| IFEval | English | ori | n | 0 | y/n | n | 0.0/0.6/0.7 | 0.8/0.95 | 0,1.5 | 8192/16384 |
| Arena-Hard | English | ori | n | 0 | y/n | n | 0.0/0.6/0.7 | 0.8/0.95 | 0/1.5 | 8192/32768 |
| Human Eval | English | ori | n | 0 | y/n | n | 0.3 | 0.95 | x | 8192/32768 |

## 3 MOTIVATION

**The flaw of existing LLM evaluation methodology.** Existing LLM benchmarks define workload formats and scoring rules, but leave crucial indispensable components uncontrolled, e.g., decoding parameters and prompting methods. As a result, reported evaluation outcomes often do not allow a direct comparison of model differences and may conflate intrinsic capability with arbitrary component settings. To make this issue concrete, we systematically reviewed major benchmarks and compiled a taxonomy of which components are explicitly defined and which are left open (Table 1). Strikingly, many widely used benchmarks, including AIME, specify only a subset of variables while leaving key components underspecified. To quantify the implications, we reconstructed the AIME evaluation space by enumerating plausible settings of uncontrolled components (e.g., COT, temperature, top-p, presence penalty, max tokens), yielding 162 distinct evaluation conditions. Accuracy under these conditions varied by as much as 70% across settings, and the distributions often diverged substantially from the single numbers reported in technical documentation. On some models, the median relative change between our measured accuracy and the accuracy reported in the technical report reached as high as 50%(see Figure 2). Comparable inconsistencies are evident in MMLU (Appendix A.2) and other flagship benchmarks, suggesting that the problem is not dataset-specific but structural across current LLM evaluation methodologies. These findings reveal a fundamental flaw in current practice: a benchmark score is often not a property of the model alone but of the loosely specified evaluation system surrounding it. Without principled control over these confounding components, evaluation becomes unstable, attribution unreliable, and comparisons across models misleading.

**The challenges of using Evaluatology for LLM evaluation.** Zhan et al. conceptualize evaluation as constructing a minimal system that integrates the evaluation object with indispensable components while considering user requirements (Zhan, 2024; Zhan et al., 2024). Wang et al. illustrate this

approach for CPUs, where a Minimal Evaluation System (MES) isolates CPU behavior from confounding components (Wang et al., 2024a). However, extending Evaluatology to LLMs presents a qualitatively deeper challenge than in the case of CPUs or other deterministic systems. For such conventional artifacts, workloads can be reasonably characterized and stabilized: standardized benchmarks capture dominant usage scenarios and once confounders are controlled, evaluation outcomes largely reflect intrinsic system differences. By contrast, LLM workloads are inherently open-ended and socially constructed, shaped by heterogeneous users, diverse linguistic and cultural contexts, and the continual emergence of novel use cases. In this setting, even the "unit of evaluation" becomes unstable: what qualifies as mainstream, extrapolative, or atypical shifts across communities and over time. To illustrate, consider the following problem from AIME: "Let $A$, $B$, $C$, and $D$ be points on the hyperbola $\frac{x^2}{20} - \frac{y^2}{24} = 1$ such that $ABCD$ is a rhombus whose diagonals intersect at the origin. Find the greatest real number that is less than $BD^2$ for all such rhombi." When evaluated on nine LLMs including deepseek, doubao, gpt series, moonshot, mistral, qwen series, etc.,

five were able to solve this original (seen) workload correctly. However, after performing analogical transformations through inserting distractor: "In a geometric study, we often encounter various shapes and their properties. Also, the concept of symmetry plays an important role in analyzing the relationships between different geometric figures. Let $A$, $B$, $C$, and $D$ be points on the hyperbola $\frac{x^2}{20} - \frac{y^2}{24} = 1$ such that $ABCD$ is a rhombus whose diagonals intersect at the origin. Find the greatest real number that is less than $BD^2$ for all such rhombi.", none of these models produced correct solutions. This striking contrast illustrates why A-MES is essential: performance on a single workload can be misleading, as models may succeed on problems they have effectively memorized yet fail when the same reasoning must be applied under slightly altered conditions.

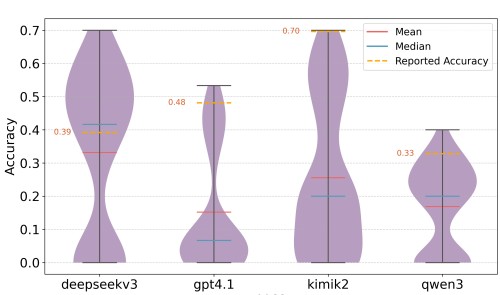

Figure 2: Accuracy deviations on AIME when evaluating with identical workloads across 162 combinations of component settings (COT, temperature, top-p, presence penalty, and max tokens).

## 4 LLM EVALUATOLOGY

LLM evaluatology consists of three essential steps: (1) defining MES, (2) defining A-MES, and (3) evaluating MES/A-MES and attributing evaluation outcomes.

### 4.1 DEFINING MINIMAL EVALUATION SYSTEM (MES)

We define the Minimal Evaluation System (MES) for LLM evaluation as the smallest independently runnable system that includes the evaluated object and all indispensable components that materially affect the evaluation outcome. The evaluated object $O$ is not limited to a bare LLM; it can also encompass the broader deployed LLM service that fuses the model with its supporting software and hardware stack. For example, when evaluating through an API, the LLM and its underlying systems should be treated as an inseparable whole, whereas for locally deployed open-source models, the surrounding system environment may either be incorporated into $O$ or explicitly modeled as part of the other indispensable components. Thus, the first step of defining MES is to rigorously define the evaluated object.

The second step in defining MES is to identify the indispensable components that shape evaluation outcomes and to establish their value ranges, collectively denoted as evaluation conditions (EC). We organize EC into three layers, covering workload, prompting method, and decoding parameters, which together yield 10 key factors ($C_1$–$C_{10}$). *Workload* captures data-related variations, including Language, Question Format, and Question Paraphrase ($C_1$–$C_3$). Note that Question Paraphrase is introduced as a key component to mitigate hallucination and data contamination, referring to reformulating questions without altering their semantics or correct answers. *Prompting method* accounts for interaction styles, namely Shot, COT(chain-of-thought), and Multi Turn ($C_4$–$C_6$). *De-*

Table 2: Evaluation Conditions: Indispensable Components and Value Ranges

| Variable | Value Range |
|---|---|
| Language | Chinese, English, Japanese, Arabic, French, Russian |
| Question Format | Multiple-choice, Fill-in-the-blank |
| Question Paraphrase | Yes, No |
| Shot | Yes, No |
| COT | Yes, No |
| Multi Turn | Yes, No |
| temperature | 0.0, 1.0, 2.0 |
| top_p | 0.2, 0.6, 1.0 |
| presence_penalty | -0.5, 0.5, 1.5 |
| max_tokens | 10, 100, 4000 |

*coding parameters* represent inference controls, including temperature, top_p, presence_penalty, and max_tokens ($C_7$–$C_{10}$). Each component is instantiated with representative values to balance coverage of real-world variability against configuration space tractability. The indispensable components and their value ranges are summarized in Table 2, with both components and their value ranges configurable based on the evaluation object and user-defined requirements. Each MES instance is then specified as $MES = EC \times O$, ensuring that performance measurements are attributed correctly while systematically controlling for confounding factors introduced by indispensable components.

## 4.2 Constructing Augmented MES (A-MES)

To further overcome the limitations of traditional evaluation, we extend MES into an augmented form (A-MES) by expanding the workload subspace. Specifically, an MES is defined as $EC \times O$, where the evaluation conditions factorize as $EC = W(workload) \times P(prompting\_methods) \times D(decoding\_parameters)$. We do augmentation in workload $W$ and leave the non-workload EC components $P$ and $D$ unchanged when building A-MES. Thus $A\text{-}MES = O \times EC_A$, with $EC_A = W_A \times P \times D, W_A = A(W)$. $A()$ is the augmentation operator that expands the original workload $W$ into an enriched workload $W_A$. Practically, $A(W)$ is constructed by partitioning and extending items from the original workload into three purpose-built categories—*original (seen)*, *analogical transformation (transformed)*, and *novel (newly-synthesized)* workloads—as shown in Fig. 3:

In our implementation, the latter two categories of augmented workloads are constructed through five systematically defined, script-driven transformation pipelines including: three analogical (distractor insertion, numeric substitution, conditional recomposition) and two novel (recent-source adaptation and conceptual synthesis) pipelines. For each pipeline, we fix general prompts and scaffolding, and then run the entire process automatically through LLM API calls (e.g., GPT-5), lightweight verification scripts and other auxiliary tooling. This setup scales to large workloads and produces diverse variants, without any per-item manual rewriting or hand-crafting of individual problems. Below we outline the overall automatic transformation process; detailed procedures, transformation examples, and algorithmic pseudocode are provided in Appendix A.3.

**(1) Analogical Pipelines**

- **Distractor Insertion.** Distractor insertion augments an original question by adding redundant sentences at a random position. We systematically divide redundant information into three categories: (i) context-irrelevant redundancy that is completely unrelated to the problem content; (ii) context-relevant explanatory redundancy that explains concepts already appearing in the problem; and (iii) context-relevant misleading redundancy that is logically related to the problem but deliberately nudges the solver toward an incorrect strategy. By providing the LLM with transformation examples, the correct answer, and (optionally) solution steps, all three types of redundancy are generated via similar structured prompts and automatically inserted at random positions in the problem statement. We empirically evaluated multiple candidate LLMs for this task and selected the one that most consistently respects these constraints, GPT-5, as our transformation executor.

- **Numeric Substitutions.** Numeric substitution augments a problem by systematically perturbing its key numerical parameters. We leverage the correct answer, solution sketches, and information from a pre-built formula library to prompt the LLM to generate a Python solver that explicitly parameterizes the key numbers in the problem. We then execute this solver locally and, if any error or mismatch is detected, feed the error messages back to the LLM for iterative refinement until the code passes verification. Once a reliable solver is obtained, we automatically sample new parameters within a predefined range and invoke the solver to compute the

corresponding new answers, thereby creating a family of numeric variants without manually editing each instance or recomputing answers by hand.

- **Conditional Recomposition** Conditional recomposition augments a problem by constructing "inverse" variants where the original answer is treated as a given condition and some of the original conditions become the new target quantities. We first prompt the LLM to identify which conditions and target quantities can be interchanged, and then, following a procedure similar to numeric substitutions, use a large language model to generate a Python solver that explicitly parameterizes the key numerical quantities in the problem. The solver is iteratively refined until it passes verification. Once a verified solver is obtained, we perturb the new input conditions within a reasonable range and automatically generate multiple "conditionally recomposed" variants of the original problem.

(2) Novel Pipelines

- **Recent-source Adaptation** Recent-source adaptation augments a problem by aligning it with thematically similar questions drawn from recent real-world exams. Given an original problem, we first use an LLM to extract its core knowledge points. We then query a public exam-question repository, indexed by year, region, subject, and knowledge point, to retrieve recent (e.g., 2025) exam questions that match these knowledge points. The retrieved questions are subsequently paraphrased via LLM, and can optionally be further transformed using the three analogical pipelines described above. In this way, we obtain recent-source adapted problems that remain aligned with the original item at the knowledge-point level while being entirely new instances.

- **Conceptual Synthesis** Conceptual synthesis augments a problem by generating conceptual questions that target the underlying concepts. Based on authoritative textbooks in PDF form, we build a structured knowledge base in which each concept is associated with its definitions, theorems, phenomena, and canonical examples extracted from the textbooks. For a given problem, we use an LLM to identify its primary knowledge points and retrieve the corresponding concept entries from the knowledge base. If the corresponding concept entries is missing, we trigger a textbooks crawling and parsing step to expand the knowledge base and try again. We then prompt the LLM to synthesize new conceptual questions grounded in these entries, yielding problems that probe conceptual understanding underlying the origin item.

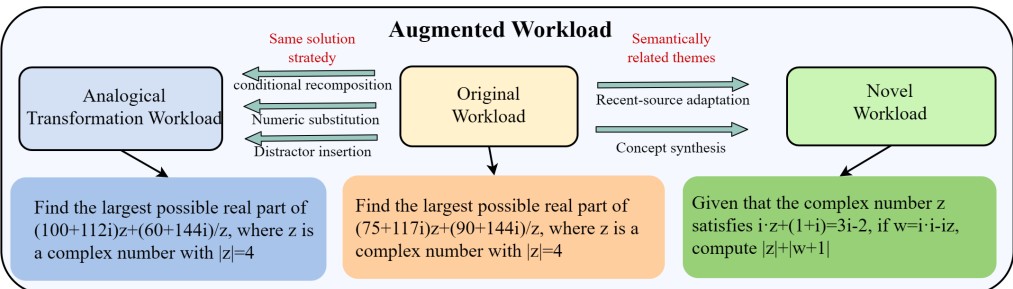

Figure 3: Augment the original workload into analogical transformation and novel workloads.

### 4.3 EVALUATING ON A-MES AND ATTRIBUTING EVALUATION OUTCOMES

MES samples from the full space defined by 10 variables, whereas A-MES starts from this same space and, for each question within a benchmark, applies all seven mechanisms to construct the full space of augmented variants, filtering out transformation attempts that fail (e.g., numeric substitutions or conditional recomposition without a stable solver). Evaluation then samples directly from this augmentation space, ensuring that no invalid transformations are ever selected; the small number of discarded variants has negligible impact on overall coverage or robustness.

Given the exponentially large space of workload, prompting methods, and decoding parameters, exhaustive testing is generally infeasible. Evaluation on MES/A-MES balances the trade-off between evaluation accuracy and evaluation cost by systematically sampling the configuration space of evaluation conditions under a joint convergence- and LLN-based stopping rule. Specifically, we generate

the full list of configurations once and shuffle it with a fixed random seed. This single globally shuffled list is then shared across all models and workloads. For any given model and benchmark, use a sample size $N$ by selecting the first $N$ configurations from this global list, without any further re-shuffling. We process this shuffled list sequentially in fixed-size batches (e.g., 10). After every batch, we recompute the running mean accuracy over all configurations seen so far, together with the corresponding 95% confidence interval. We stop sampling when two conditions are satisfied simultaneously: (i) the absolute changes in the running mean accuracy for the last three updates are all smaller than 0.002, and (ii) the length of the 95% confidence interval is smaller than 0.06. The number of configurations evaluated when these criteria are first satisfied is denoted $N_{\text{conv}}$. After convergence, we further apply a simple Law of Large Numbers–based estimation: using the empirical variance of the current results, we compute the minimal sample size required to achieve the desired error tolerance and confidence level, obtaining an LLN-based sample size $N_{\text{LLN}}$. If $N_{\text{LLN}}$ is larger than $N_{\text{conv}}$, we continue sampling along the same shuffled order until $N_{\text{LLN}}$ configurations have been evaluated; otherwise, we stop at $N_{\text{conv}}$. Combining the convergence criterion with the LLN-based check ensures that our sample sizes are both empirically stable and theoretically justified.

The sampled evaluation conditions are then used to test the evaluation object, yielding performance outcomes under diverse settings. The final reported evaluation score for each model is then the mean performance over the sampled instances, together with 95% and 99% confidence intervals, providing a stable summary metric that balances comprehensiveness with practical efficiency. One approach to isolate component effects is to use equivalent evaluation conditions, where all component settings are held constant except for the factor of interest; differences in measured performance can thus be attributed directly to that component, effectively mitigating confounding. An alternative and complementary approach is to apply ANOVA (analysis of variance) across the sampled configurations, quantifying the proportion of performance variance explained by each component and enabling systematic attribution of effects. Together, these strategies provide both controlled and statistical means to disentangle intrinsic model capability from the influence of evaluation conditions.

## 5 EVALUATION

In this section, we evaluate the proposed methodology using mainstream LLMs that are publicly accessible, including deepseek-v3, doubao-1.5-pro-32k, gpt-3.5, gpt-4.1, moonshot-v1-8k, mistral-large, mistral-medium, qwen-plus and qwen2.5-32b-instruct. We have three targets. 1) Demonstrate the necessity of constructing MES and A-MES for LLM evaluation by varying the settings of each indispensable component within MES and A-MES. 2) Quantify the contribution of each indispensable component to overall performance variance and identify the key components affecting LLM behavior using ANOVA. 3) Compare LLM evaluatology with traditional LLM evaluation methods and show how it enables accurate attribution of performance differences to specific components.

For online testing, we primarily access the models through their official APIs; however, since the official API for Deepseek v3 has been discontinued, we instead use the API provided by a third-party server deployment. This study employs several widely used and representative benchmark datasets—MMLU, GPQA, and AIME—as the basis for evaluation. Note that due to the page limit, the results of MMLU and GPQA are listed in Appendix A.4. MMLU covers 57 subjects and contains a large collection of multiple-choice questions, widely used to assess models' general knowledge and reasoning abilities. GPQA consists of 448 challenging multiple-choice questions developed and validated by experts in biology, physics, and chemistry, designed to evaluate AI models' reasoning ability on complex scientific problems. AIME is a highly selective U.S. high school mathematics competition, well known for its challenging problems that test deep mathematical reasoning. It is worth noting that our methodology is not tied to any specific benchmark and can be applied to the evaluation of any LLM.

### 5.1 THE NECESSITY OF CONSTRUCTING MES AND A-MES

This section demonstrates that LLMs exhibit significant performance variations across different MES and A-MES configurations, thereby underscoring the inadequacy of single-configuration evaluations in accurately capturing their true capabilities.

Table 3: Performance Rankings of LLMs (deepseek = deepseek-v3, doubao = doubao-1.5-pro-32k, mistralL = mistral-large, mistralM = mistral-medium, kimi = moonshot-v1-8k, qwenP = qwen-plus)

| Type | 1. | 2 | 3 | 4 | 5 | 6 | 7 | 8 | 9 |
|---|---|---|---|---|---|---|---|---|---|
| Original | deepseek(0.4) | gpt4.1(0.07) | doubao(0) | gpt3.5(0) | mistralL(0) | mistralM(0) | kimi(0) | qwenP(0) | qwen2.5(0) |
| A-MES | deepseek(0.45) | qwenP(0.43) | mistralL(0.38) | gpt4.1(0.37) | mistralM(0.36) | doubao(0.34) | qwen2.5(0.26) | kimi(0.18) | gpt3.5(0.13) |
| MES | doubao(0.25) | deepseek(0.21) | qwen2.5(0.21) | mistralL(0.20) | mistralM(0.20) | qwenP(0.18) | gpt4.1(0.16) | gpt3.5(0.10) | kimi(0.08) |

Note: Models are sorted alphabetically by name when accuracy equals zero.

In the MES experiments, we conducted 500 random samplings without replacement from the MES configuration space described in Section 4.1. The specific components and their corresponding value ranges are summarized in Table 2. We determined 500 as a conservative sample-size upper bound by combining a convergence-based stopping rule with LLN-guided estimates.

In the A-MES experiments, to verify the effectiveness of the Augmented Minimal Evaluation System (A-MES) proposed in Section 4.2 and comprehensively evaluate the performance of LLMs across diverse task scenarios, we conducted a comparative analysis of their performance on two types of datasets: the original AIME workload and four augmented workloads derived from this original workload. For the analogical transformation workload, we employed two specific methods: the first involves inserting redundant information into the stem of the original question, information that is irrelevant to the problem-solving logic and methods yet consistent with the question scenario, to interfere with the output results of LLMs; The second method involves numeric substitutions. For novel (newly-synthesized) workloads, this study designs two core strategies: the first is a knowledge point-based question generation strategy, which specifically generates new tasks based on the core knowledge points covered in the original questions and combined with the conceptual system and expression paradigm of relevant textbook chapters; The second is an adaptation and transformation strategy based on college entrance examination (gaokao) questions, which involves selecting the latest gaokao questions that match the target knowledge points and generating new tasks by adjusting the scenario of the question, questioning logic, and other aspects.

The experimental results of this study are presented in Figure 4. As shown in Figure 4, significant variations in accuracy trends are observed across different models and configuration spaces. For instance, the accuracy of the deepseek-v3 model fluctuates within a range of 0 to 0.78 under MES experiments, and from 0.23 to 0.7 under A-MES experiments. As shown in Table 3, we have also generated performance rankings for large models based on the original workload, MES workload, and A-MES workload. For the MES and A-MES scenarios, we employ their average accuracy as the performance metric for the LLMs. It is crucial to note that when the accuracy is zero, we sort the models alphabetically based on their names. Drawing insights from the rankings, we observe three key conclusions: first, the original evaluation methodology demonstrates limited effectiveness in benchmarking large language models (LLMs) due to its inability to distinguish performance beyond two models achieving non-zero accuracy scores; second, DeepSeek consistently outperforms all competing models across diverse evaluation conditions, underscoring its robustness and superior generalization capabilities; third, model performance rankings exhibit contextual sensitivity, as evidenced by Doubao's inferior performance relative to DeepSeek in both Original and A-MES workloads, yet its top-ranking achievement in MES, thereby highlighting the non-transitive nature of LLM performance across varying task formulations and data distributions.

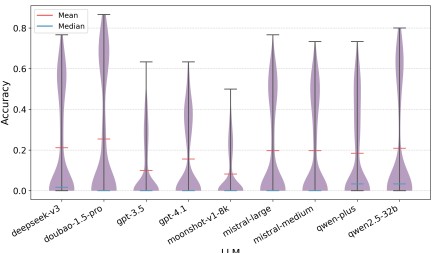
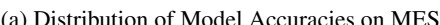
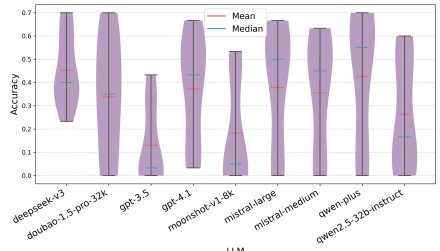

(a) Distribution of Model Accuracies on MES      (b) Distribution of Model Accuracies on A-MES

Figure 4: Distribution of Model Accuracies

Table 4: ANOVA results on DeepSeek-V3 (sorted by effect size in descending order)

| Factor | Effect Size $\eta^2$ | $p$-value |
|---|---|---|
| Question Format | 0.399643 | 0.000 |
| Question Format - COT | 0.161394 | 0.000 |
| COT | 0.080156 | 0.000 |
| max_tokens | 0.028099 | 0.000 |
| Question Format - Shot | 0.011101 | 0.000 |
| Language - Question Format | 0.008178 | 0.006 |
| COT - max_tokens | 0.006721 | 0.010 |
| Language - COT | 0.004345 | 0.038 |
| Multi Turn - max_tokens | 0.003841 | 0.050 |
| Language | 0.003841 | 0.046 |
| Shot - max_tokens | 0.003669 | 0.046 |
| Question Format - max_tokens | 0.002687 | 0.100 |
| Language - Multi Turn | 0.002600 | 0.066 |
| temperature - top_p | 0.002082 | 0.178 |
| Question Format - Multi Turn | 0.001321 | 0.244 |

## 5.2 QUANTIFY THE CONTRIBUTION OF EACH INDISPENSABLE COMPONENT TO OVERALL PERFORMANCE VARIANCE

In LLM evaluation, a key challenge lies in effectively evaluating the contribution of each components illustrated in Fig. 1 to overall performance variance. Given the enormous number of possible EC configuration combinations, exhaustively testing every configuration is computationally infeasible. To address this, we selected a limited number of experimental points from the full space, allowing us to systematically and evenly examine the effects of multiple components and their levels on performance with significantly fewer trials. This design reduces experimental cost while maintaining scientific rigor and representativeness.

To quantify the proportion of performance variance explained by each MES component, we adopted an analysis of variance (ANOVA) approach. Specifically, for component $C_1$–$C_{10}$, we selected two levels ("high" and "low") within their respective ranges, with these ranges given in Table 2, thereby constructing a subspace of size $2^{10} = 1024$. For the Language component, we selected Chinese and English, while for three-valued components we used their maximum and minimum values. Within this subspace, variance decomposition was used to quantify the contributions of different components and their interactions to variations in accuracy. Moreover, we employed a permutation test to evaluate statistical significance, enabling a more robust assessment of component importance without relying on additional distributional assumptions. This procedure yields both the relative importance and the statistical significance of all components.

Taking the DeepSeek-V3 model as an example, Table 4 reports the main effects and two-way interactions that significantly influence its accuracy on the AIME'24 benchmark, with the complete ANOVA results provided in Appendix A.5. Overall, Question type, COT, max_tokens, and their interactions with other components exhibit the most significant effects. Shot, Multi turn, and Language also show significant effects, while the remaining components have only limited impact.

Consistent patterns were observed across other LLMs (see Appendix A.5). Using $p < 0.05$ as the significance threshold, we found that the main effects of Question format and COT, or their interactions with other components, were consistently significant across all LLMs. Furthermore, max_tokens, Shot, and Multi turn also reached significance for the vast majority of models. In addition to these five core components, Language, top_p, and temperature were significant for some models. It is worth noting that for the remaining two components, Question Paraphrase and presence_penalty, the $p$-values did not meet the significance threshold, but reached 0.19 and 0.16, respectively, on GPT-4.1. This suggests that they may exert some influence on model performance, although the evidence is not sufficient for a definitive conclusion.

## 5.3 COMPARE LLM EVALUATOLOGY WITH TRADITIONAL LLM EVALUATION METHODS AND ATTRIBUTE THE PERFORMANCE DIFFERENCES TO SPECIFIC COMPONENTS

This section demonstrates that evaluating models under a single configuration fails to capture their true capabilities, while LLM evaluatology not only yields results in strong agreement with the ground truth, but also attributes the performance differences to specific components.

Based on the randomly sampled data collected from the complete configuration space spanned by the components in Table 2, we estimated the overall average accuracies of different models on the same benchmark using their 95% and 99% confidence intervals. As illustrated in Figure 5a, we

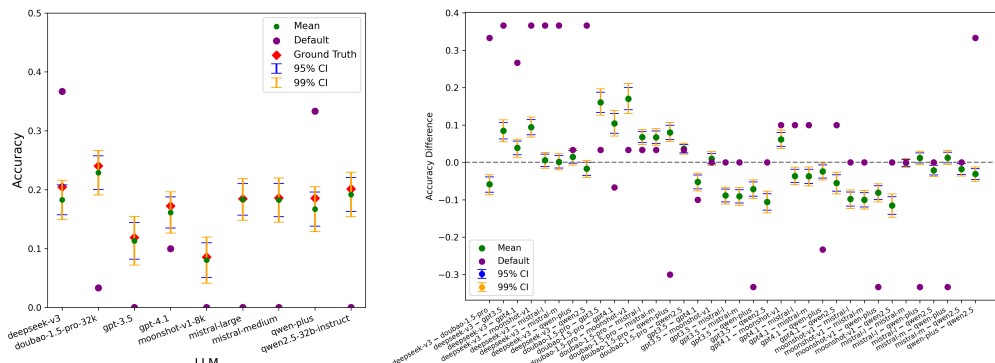

(a) Accuracy confidence intervals of different LLMs on AIME

(b) Accuracy difference confidence intervals of different models on AIME

Figure 5: Comparison of LLM Evaluatology and Traditional Method on AIME

report the performance of different models on AIME'2024, where the purple dots denote the test results under the commonly adopted default setting, using the original workloads without optimized prompting methods and with default decoding parameters. It can be observed that the purple dots are far from the confidence intervals (interval estimation of the population mean) obtained through random sampling, showing that evaluating a model under a single configuration is unreliable. Note that accuracy values of 0 in Figure 5a are not due to missing data, but to the high difficulty of AIME problems, which are challenging even for human contestants.

Figure 5b further presents, on AIME'2024, the accuracy differences between two models under the default configuration, together with the 95% and 99% confidence intervals constructed from accuracy differences observed across sampled equivalent evaluation configurations. In 10 cases, the confidence intervals and the default accuracy differences fall on opposite sides of the zero line, revealing contradictions in the conclusions regarding model superiority. For instance, the 99% confidence interval for the mean accuracy difference between Doubao-1.5-pro and GPT-4.1 lies entirely above the zero line, implying that overall Doubao-1.5-pro outperforms GPT-4.1. However, if one were to rely on the result of a single experiment under the default configuration, the accuracy difference would fall below the zero line, leading instead to the opposite conclusion that GPT-4.1 outperforms Doubao-1.5-pro. This "conclusion reversal" highlights the limitations of relying solely on single-configuration testing. More detailed results on additional benchmarks including MMLU and GPQA can be found in the Appendix.

Furthermore, we selected the five most influential components for a cost-efficient accuracy test on each LLM, based on the ANOVA data in Section 5.2. We then constructed the configuration subspace restricted to these components and conducted exhaustive testing within this subspace. The mean performance obtained was taken as a "restricted-space ground truth." As shown by the red diamond in Figure 5a, for all models, this reference truth fell within the confidence intervals estimated from random sampling, thereby demonstrating both the validity and the robustness of the proposed LLM evaluatology method.

# 6 CONCLUSION

LLM Evaluatology establishes a principled methodology for assessing LLMs through an Augmented Minimal Evaluation System (A-MES), explicitly accounting for both intrinsic model capabilities and the many confounding components that shape observed performance, thereby enabling accurate attribution of performance differences to their true sources. Our analysis reveals that meaningful evaluation of LLMs requires careful consideration of both workload heterogeneity and the vast space of evaluation condition (EC) configurations. We advocate for the adoption of evaluatology as a foundational paradigm, encouraging the community to develop richer workload augmentation strategies and robust evaluation practices that mirror the complexity of actual deployment scenarios.

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

# A  APPENDIX

## A.1  THE USE OF LARGE LANGUAGE MODELS (LLMS)

During the manuscript preparation, we leveraged large language models (LLMs) to assist in refining and polishing the text. Specifically, the LLM was used to improve sentence clarity and enhance linguistic fluency, while all scientific content, reasoning, and results were independently authored and verified by the researchers. This approach facilitated more concise and readable presentation without affecting the technical accuracy.

## A.2  EVALUATION SETTING ON DIFFERENT BENCHMARKS

Table 5: Evaluation Settings Reported in Technical Reports of Different LLMs. (Lang. = Language, Q-type = Question type, Para. = Paraphrase, M-turn = Multi turn, Temp. = Temperature, PP = presence_penalty, MaxTok = max_tokens)

(a) Evaluation Settings on AIME'2024

| Model | Lang. | Q-type | Para. | Shot | COT | M-turn | Temp. | top_p | PP | MaxTok |
|---|---|---|---|---|---|---|---|---|---|---|
| DeepSeek-R1 | english | origin | origin | 0 | yes | x | 0.6 | 0.95 | x | 32768 |
| DeepSeek-V3 | english | origin | origin | x | x | x | 0.7 | x | x | 8192 |
| Kimi K2 | english | origin | origin | x | no | x | 0.0 | fixed | x | 8192 |
| Kimi K1.5 | english | origin | origin | x | yes | x | x | x | x | x |
| Qwen2 | | | | | Not evaluated on AIME | | | | | |
| Qwen2.5 | | | | | Not evaluated on AIME | | | | | |
| Qwen3 | english | origin | origin | x | no | x | 0.7 | 0.8 | 1.5 | 32768 |
| GPT-4 | | | | | Not evaluated on AIME | | | | | |
| GPT-4.1 | english | origin | origin | x | x | x | x | x | x | x |
| GPT-5 | | | | | Not evaluated on AIME | | | | | |
| Claude Opus 4 | | | | | Not evaluated on AIME | | | | | |
| Mistral Small3.1 | | | | | Not evaluated on AIME | | | | | |
| Mistral Medium3 | | | | | Not evaluated on AIME | | | | | |
| Mistral Large2 | | | | | Not evaluated on AIME | | | | | |

(b) Evaluation Settings on MMLU

| Model | Lang. | Q-type | Para. | Shot | COT | M-turn | Temp. | top_p | PP | MaxTok |
|---|---|---|---|---|---|---|---|---|---|---|
| DeepSeek-R1 | english | origin | origin | 0 | yes | x | 0.5 | x | x | 1024 |
| DeepSeek-V3 | english | origin | origin | 0 | yes | x | 0.5 | x | x | 1024 |
| Kimi K2 | english | origin | origin | x | no | x | 0.0 | fixed | x | 8192 |
| Kimi K1.5 | english | origin | origin | x | yes | x | x | x | x | x |
| Qwen2 | english | origin | origin | 5 | x | x | x | x | x | x |
| Qwen2.5 | english | origin | origin | 5 | x | x | x | x | x | x |
| Qwen3 | english | origin | origin | 5 | x | x | x | x | x | x |
| GPT-4 | multiple | origin | origin | 5/3 | no | x | x | x | x | x |
| GPT-4.1 | multiple | origin | origin | x | x | x | x | x | x | x |
| GPT-5 | | | | | Not evaluated on MMLU | | | | | |
| Claude Opus 4 | multiple | origin | origin | x | yes/no | x | x | x | x | x |
| Mistral Small3.1 | english | origin | origin | x | x | x | x | x | x | x |
| Mistral Medium3 | | | | | Not evaluated on MMLU | | | | | |
| Mistral Large2 | multiple | origin | origin | x | x | x | x | x | x | x |

(c) Evaluation Settings on GPQA

| Model | Lang. | Q-type | Para. | Shot | COT | M-turn | Temp. | top_p | PP | MaxTok |
|---|---|---|---|---|---|---|---|---|---|---|
| DeepSeek-R1 | english | origin | origin | 0 | yes | x | 0.5 | x | x | 1024 |
| DeepSeek-V3 | english | origin | origin | 0 | yes | x | 0.5 | x | x | 1024 |
| Kimi K2 | english | origin | origin | x | no | x | 0.0 | fixed | x | 8192 |
| Kimi K1.5 | english | origin | origin | x | yes | x | x | x | x | x |
| Qwen2 | english | origin | origin | x | x | x | x | x | x | x |
| Qwen2.5 | english | origin | origin | x | x | x | x | x | x | x |
| Qwen3 | english | origin | origin | x | yes/no | x | 0.6/0.7 | 0.95/0.8 | 0/1.5 | 32768 |
| GPT-4 | | | | | Not evaluated on GPQA | | | | | |
| GPT-4.1 | english | origin | origin | x | x | x | x | x | x | x |
| GPT-5 | english | origin | origin | x | 0/1 | x | x | x | x | x |
| Claude Opus 4 | english | origin | origin | x | 0/1 | x | x | x | x | x |
| Mistral Small3.1 | english | origin | origin | x | x | x | x | x | x | x |
| Mistral Medium3 | english | origin | origin | 5 | 1 | x | x | x | x | x |
| Mistral Large2 | | | | | Not evaluated on GPQA | | | | | |

(d) Evaluation Settings on MATH

| Model | Lang. | Q-type | Para. | Shot | COT | M-turn | Temp. | top_p | PP | MaxTok |
|---|---|---|---|---|---|---|---|---|---|---|
| DeepSeek-R1 | english | origin | origin | 0/8 | yes/no | x | 0 | x | x | 32768 |
| DeepSeek-V3 | english | origin | origin | 0/8 | yes/no | x | 0 | x | x | 8192 |
| Kimi K2 | english | origin | origin | x | no | x | 0.0 | fixed | x | 8192 |
| Kimi K1.5 | english | origin | origin | x | yes | x | x | x | x | x |
| Qwen2 | english | origin | origin | x | x | x | x | x | x | x |
| Qwen2.5 | english | origin | origin | x | x | x | x | x | x | x |
| Qwen3 | english | origin | origin | x | yes/no | x | 0.6/0.7 | 0.95/0.8 | 0/1.5 | 32768 |
| GPT-4 | | | | | Not evaluated on MATH | | | | | |
| GPT-4.1 | | | | | Not evaluated on MATH | | | | | |
| GPT-5 | | | | | Not evaluated on MATH | | | | | |
| Claude Opus 4 | | | | | Not evaluated on MATH | | | | | |
| Mistral Small3.1 | english | origin | origin | x | x | x | x | x | x | x |
| Mistral Medium3 | english | origin | origin | 0 | 0 | x | x | x | x | x |
| Mistral Large2 | english | origin | origin | 0 | 0 | x | x | x | x | x |

**(e) Evaluation Settings on SWE-bench**

| Model | Lang. | Q-type | Para. | Shot | COT | M-turn | Temp. | top_p | PP | MaxTok |
|---|---|---|---|---|---|---|---|---|---|---|
| DeepSeek-R1 | english | origin | origin | x | x | x | 0.8 | x | x | x |
| DeepSeek-V3 | english | origin | origin | x | x | x | 0.8 | x | x | x |
| Kimi K2 | english | origin | origin | x | no | x | 0.0 | fixed | x | 8192/16384 |
| Kimi K1.5 | | | | | | | Not evaluated on SWE-bench | | | |
| Qwen2 | | | | | | | Not evaluated on SWE-bench | | | |
| Qwen2.5(pre) | | | | | | | Not evaluated on SWE-bench | | | |
| Qwen3(pre) | | | | | | | Not evaluated on SWE-bench | | | |
| GPT-4 | | | | | | | | | | |
| GPT-4.1 | english | origin | origin | x | x | x | x | x | x | x |
| GPT-5 | english | origin | origin | x | 0/1 | x | x | x | x | x |
| Claude Opus 4 | english | origin | origin | x | 0/1 | x | x | 0.95 | x | x |
| Mistral Small3.1 | | | | | | | Not evaluated on SWE-bench | | | |
| Mistral Medium3 | | | | | | | Not evaluated on SWE-bench | | | |
| Mistral Large2 | | | | | | | Not evaluated on SWE-bench | | | |

**(f) Evaluation Settings on IFEval**

| Model | Lang. | Q-type | Para. | Shot | COT | M-turn | Temp. | top_p | PP | MaxTok |
|---|---|---|---|---|---|---|---|---|---|---|
| DeepSeek-R1 | english | origin | origin | 0 | 0 | 0 | x | x | x | x |
| DeepSeek-V3 | english | origin | origin | 0 | 0 | 0 | x | x | x | x |
| Kimi K2 | english | origin | origin | x | no | x | 0.0 | fixed | x | 8192 |
| Kimi K1.5 | english | origin | origin | x | yes | x | x | x | x | x |
| Qwen2 | english | origin | origin | x | x | x | x | x | x | x |
| Qwen2.5 | english | origin | origin | x | x | x | x | x | x | x |
| Qwen3 | english | origin | origin | x | yes/no | x | 0.6/0.7 | 0.95/0.8 | 0/1.5 | 32768 |
| GPT-4 | | | | | | | Not evaluated on IFEval | | | |
| GPT-4.1 | english | origin | origin | x | x | x | x | x | x | x |
| GPT-5 | | | | | | | Not evaluated on IFEval | | | |
| Claude Opus 4 | | | | | | | Not evaluated on IFEval | | | |
| Mistral Small3.1 | | | | | | | Not evaluated on IFEval | | | |
| Mistral Medium3 | english | origin | origin | 0 | 0 | x | x | x | x | x |
| Mistral Large2 | | | | | | | Not evaluated on IFEval | | | |

**(g) Evaluation Settings on Arena-Hard**

| Model | Lang. | Q-type | Para. | Shot | COT | M-turn | Temp. | top_p | PP | MaxTok |
|---|---|---|---|---|---|---|---|---|---|---|
| DeepSeek-R1 | english | origin | origin | 0 | 0 | 0 | config | default | default | user-set |
| DeepSeek-V3 | english | origin | origin | 0 | 0 | 0 | config | default | default | user-set |
| Kimi K2 | english | origin | origin | x | no | x | 0.0 | fixed | x | 8192 |
| Kimi K1.5 | | | | | | | Not evaluated on Arena-Hard | | | |
| Qwen2 | english | origin | origin | x | x | x | x | x | x | x |
| Qwen2.5 | english | origin | origin | x | x | x | x | x | x | x |
| Qwen3 | english | origin | origin | x | yes/no | x | 0.6/0.7 | 0.95/0.8 | 0/1.5 | 32768 |
| GPT-4 | | | | | | | Not evaluated on Arena-Hard | | | |
| GPT-4.1 | | | | | | | Not evaluated on Arena-Hard | | | |
| GPT-5 | | | | | | | Not evaluated on Arena-Hard | | | |
| Claude Opus 4 | | | | | | | Not evaluated on Arena-Hard | | | |
| Mistral Small3.1 | | | | | | | Not evaluated on Arena-Hard | | | |
| Mistral Medium3 | | | | | | | Not evaluated on Arena-Hard | | | |
| Mistral Large2 | english | origin | origin | x | x | x | x | x | x | x |

**(h) Evaluation Settings on HumanEval**

| Model | Lang. | Q-type | Para. | Shot | COT | M-turn | Temp. | top_p | PP | MaxTok |
|---|---|---|---|---|---|---|---|---|---|---|
| DeepSeek-R1 | english | origin | origin | 0 | 0 | 0 | varied | 0.95 | x | 32768 |
| DeepSeek-V3 | english | origin | origin | 0 | 0 | 0 | varied | 0.95 | x | 8192 |
| Kimi K2 | | | | | | | Not evaluated on HumanEval | | | |
| Kimi K1.5 | english | origin | origin | x | yes | x | x | x | x | x |
| Qwen2 | english | origin | origin | x | x | x | x | x | x | x |
| Qwen2.5 | english | origin | origin | x | x | x | x | x | x | x |
| Qwen3 | | | | | | | Not evaluated on HumanEval | | | |
| GPT-4 | english | origin | origin | 0 | 0 | x | 0.3 | x | x | x |
| GPT-4.1 | | | | | | | Not evaluated on HumanEval | | | |
| GPT-5 | | | | | | | Not evaluated on HumanEval | | | |
| Claude Opus 4 | | | | | | | Not evaluated on HumanEval | | | |
| Mistral Small3.1 | english | origin | origin | x | x | x | x | x | x | x |
| Mistral Medium3 | english | origin | origin | 0 | 0 | x | x | x | x | x |
| Mistral Large2 | english | origin | origin | x | x | x | x | x | x | x |

## A.3 A-MES CONSTRUCTION PIPELINE

**Analogical: Distractor Insertion**

For distractor insertion, we define three explicit, controllable categories of redundancy and implement all instances via LLM prompting. To ensure that the inserted distractors strictly follow our predefined specifications, we empirically test several candidate LLMs and choose the one that most consistently adheres to these constraints (GPT-5). This selection is made solely to guarantee transformation fidelity rather than to compare model capabilities. For each item to be transformed, the chosen LLM is invoked through an API and, guided by our structured prompts, automatically produces and inserts the required redundant content. The concrete implementation is as follows.

(1) Context-irrelevant redundancy.

- Provide the LLM with an example containing an original question and a version with added context-irrelevant redundancy.
- Instruct the LLM to insert one sentence at a random position that is completely unrelated to the target question.

---
**Algorithm 1** Context-irrelevant redundancy insertion

---
1: **function** INSERTCONTEXTIRRELEVANTDISTRACTOR($problem\_text$)
2:     $EXAMPLE\_PAIR \leftarrow (orig\_example, example\_with\_irrelevant\_context)$
3:     $PROMPT \leftarrow$ BUILDPROMPTIRRELEVANT($EXAMPLE\_PAIR, problem\_text$)
4:     $RESPONSE \leftarrow$ LLM_CALL($PROMPT$)
5:     $transformed\_text \leftarrow$ PARSETRANSFORMEDPROBLEM($RESPONSE$)
6:     **if not** BASICSANITYCHECK($problem\_text, transformed\_text$) **then**
7:         **return** FAILURE
8:     **end if**
9:     **return** $transformed\_text$
10: **end function**

---

(2) Context-relevant, explanatory redundancy.

- Provide the LLM with an example of an original question and a version with added explanatory redundancy.
- Instruct the LLM to insert a redundant sentence at a random position in each target question that explains a concept already appearing in the target question.

---
**Algorithm 2** Context-relevant explanatory redundancy insertion

---
1: **function** INSERTEXPLANATORYDISTRACTOR($problem\_text$)
2:     $EXAMPLE\_PAIR \leftarrow (orig\_example, example\_with\_explanatory\_sentence)$
3:     $PROMPT \leftarrow$ BUILDPROMPTEXPLANATORY($EXAMPLE\_PAIR, problem\_text$)
4:     $RESPONSE \leftarrow$ LLM_CALL($PROMPT$)
5:     $transformed\_text \leftarrow$ PARSETRANSFORMEDPROBLEM($RESPONSE$)
6:     **if not** BASICSANITYCHECK($problem\_text, transformed\_text$) **then**
7:         **return** FAILURE
8:     **end if**
9:     **return** $transformed\_text$
10: **end function**

---

(3) Context-relevant, misleading redundancy.

- Provide the LLM with an example containing an original question and a version with added misleading but logically related redundancy.
- Supply the model with the correct answer and several correct solution approaches, and instruct it to avoid directly hinting at these correct strategies when crafting the misleading cue. The official answer and solution approaches are provided by the user, and providing solution approaches is optional.
- Instruct the model to insert a redundant sentence that nudges the reader toward an incorrect strategy or line of reasoning, without explicitly revealing that it is "misleading" or "distracting".

---
**Algorithm 3** Context-relevant misleading redundancy insertion

---
1: **function** INSERTMISLEADINGDISTRACTOR($problem\_text, answer\_gold, solution\_sketches$)
2:     $EXAMPLE\_PAIR \leftarrow (orig\_example, example\_with\_misleading\_sentence)$
3:     $PROMPT \leftarrow$ BUILDPROMPTMISLEADING(example_pair = $EXAMPLE\_PAIR$, target_problem = $problem\_text$, answer_gold = $answer\_gold$, solution_sketches = $solution\_sketches$)
4:     $RESPONSE \leftarrow$ LLM_CALL($PROMPT$)
5:     $transformed\_text \leftarrow$ PARSETRANSFORMEDPROBLEM($RESPONSE$)
6:     **if not** BASICSANITYCHECK($problem\_text, transformed\_text$) **then**
7:         **return** FAILURE
8:     **end if**
9:     **return** $transformed\_text$
10: **end function**

---

In practice, the selected LLM produces variations that are more diverse and linguistically natural than manual editing. In particular, its context-relevant misleading redundancies tend to hint at incorrect heuristics in a more subtle way than hand-written versions, while still strictly adhering to the predefined category constraints. The entire process involves no per-item manual editing. The three examples of redundancy for three types generated by the above procedure are illustrated as follows:

1. Context-irrelevant redundancy example:

> *The weather today seems quite pleasant, and it might be a great day for a picnic.* Find the number of triples of nonnegative integers $(a, b, c)$ satisfying $a + b + c = 300$ and $a^2b + a^2c + b^2a + b^2c + c^2a + c^2b = 6{,}000{,}000$.

Here, the weather is entirely unrelated to the math content.

2. Context-relevant, explanatory redundancy example:

> There exist real numbers $x$ and $y$, both greater than 1, such that $\log_x(y^x) = \log_y(x^{4y}) = 10$. *A logarithm is a way to express how many times a base must be multiplied by itself to get a certain number.* Find $xy$.

The added sentence explains the notion of a logarithm while leaving the underlying problem unchanged.

3. Context-relevant, misleading redundancy example:

> Alice and Bob play the following game. A stack of $n$ tokens lies before them. The players take turns with Alice going first. On each turn, the player removes either 1 token or 4 tokens from the stack. *Many players adopt a greedy approach here: always take 4 whenever possible to shorten the game and restrict the opponent's replies.* Whoever removes the last token wins. Find the number of positive integers $n$ less than or equal to 2024 for which there exists a strategy for Bob that guarantees that Bob will win the game regardless of Alice's play.

The extra sentence about the "greedy approach" is logically related to the game but suggests a flawed strategy, intentionally nudging the solver toward an incorrect line of reasoning.

**Analogical: Numeric Substitutions**

For numeric substitutions, we use a uniform pipeline built around LLM-generated Python solvers and automatic verification scripts, rather than manually changing a few numbers:

- We first call an LLM to extract the primary knowledge points tested by the original problem, and query a pre-constructed formula library indexed by knowledge point to retrieve potentially relevant formulas.
- We feed the original problem, its official answer, the retrieved formulas, and (where available) multiple correct solution sketches into the LLM. The official answer and solution sketches are provided by the user, and providing solution sketches is optional.
- The LLM is prompted to:
  - analyze the problem's solution strategy, using the provided solution sketches when available;
  - write a Python solution program where problem-specific numbers are extracted as explicit variables with reasonable value ranges.
- We then ask another LLM to inspect the generated Python code and verify that it implements a general computational procedure for solving the problem, rather than relying on hard-coded instance-specific outputs or trivial pattern matching.
- We import the LLM-generated Python code as a local module and call its `solve()` function with the original numeric values as inputs, checking whether the resulting output matches the official answer.
- If the code fails (wrong answer or runtime error), we return the error message and incorrect output to the LLM, asking it to refine the code; we repeat this refinement–verification loop for up to five attempts and keep the Python code if it passes on the original instance.

After obtaining a correct solver, we automatically sample new numeric configurations within the validated ranges to generate analogical variants of the same underlying problem.

This "knowledge-point extraction → formula retrieval → analyze → code → verify → resample" pipeline is identical across all problems.

---

**Algorithm 4** Numeric substitutions

---

1: **function** BUILDNUMERICSOLVER(*problem_text*, *answer_gold*, *solution_sketches*, *max_iter* = 5, *max_refine* = 5)
2:     *knowledge_points* ← LLM_EXTRACTKNOWLEDGEPOINTS(*problem_text*)
3:     *retrieved_formulas* ← RETRIEVE_FORMULAS(*knowledge_points*)
4:     *history* ← EMPTY_LIST
5:     **for** *iter* = 1 **to** *max_iter* **do**
6:         *PROMPT* ← BUILDPROMPTCODEGEN(*problem_text*,*answer_gold*,*solution_sketches*,*retrieved_formulas*)
7:         APPEND(*history*, (*PROMPT*, NONE))
8:         (*CODE*, *param*, *value_ranges*) ← LLM_CALL(*PROMPT*)
9:         *is_hard_code* ← LLM_HARD_CODE_CHECK(*CODE*)
10:         **if** *is_hard_code* **then**
11:             **continue**
12:         **end if**
13:         **for** *refine_step* = 0 **to** *max_refine* **do**
14:             (*output*, *error*) ← RUN_PYTHON(*CODE*, input = *param.value*)
15:             APPEND(*history*, (*CODE*, (*output*, *error*)))
16:             **if** *error* = NONE **and** VERIFY(*output*, *answer_gold*) **then**
17:                 **return** (*CODE*, *param*, *value_ranges*)
18:             **end if**
19:             **if** *refine_step* = *max_refine* **then**
20:                 **break**
21:             **end if**
22:             *PROMPT_refine* ← BUILDPROMPTCODEREFINE(*problem_text*, *answer_gold*, *history*)
23:             (*CODE*, *param*, *value_ranges*) ← LLM_CALL(*PROMPT_refine*)
24:         **end for**
25:     **end for**
26:     **return** FAILURE
27: **end function**
28: **function** GENERATENUMERICVARIANTS(*problem_text*, *answer_gold*, *solution_sketches*)
29:     (*solver_code*, *param*, *param_ranges*) ← BUILDNUMERICSOLVER(*problem_text*, *answer_gold*, *solution_sketches*)
30:     **if** *solver_code* = FAILURE **then**
31:         **return** NONE
32:     **end if**
33:     *new_param* ← SAMPLE(*param_ranges*)
34:     *new_problem_text* ← INSTANTIATENUMERICPROBLEMTEXT(*problem_text*, *param*, *new_param*)
35:     (*output*, *error*) ← RUN_PYTHON(*solver_code*, input = *new_param*)
36:     *new_answer_gold* ← *output*
37:     **return** (*new_problem_text*, *new_answer_gold*)
38: **end function**

---

An example of numeric substitutions is given as follows.

- **Original:**

  Find the largest possible real part of $(75 + 117i)z + \frac{96+144i}{z}$ where $z$ is a complex number with $|z| = 4$. A common shortcut is to take $z$ to be a positive real number, since for a fixed modulus the real part is often largest when the argument of $z$ is zero.

- **Numeric variant:**

  Find the largest possible real part of $(100 + 112i)z + \frac{60+144i}{z}$ where $z$ is a complex number with $|z| = 4$. A common shortcut is to take $z$ to be a positive real number, since for a fixed modulus the real part is often largest when the argument of $z$ is zero.

**Analogical: Conditional Recompositions**

For conditional recompositions, we again adopt a general and automatable pipeline built around LLM-generated Python solvers and automatic verification scripts, rather than manually rewriting statements:

- We first call an LLM to extract the primary knowledge points tested by the original question, and query a pre-constructed formula library indexed by knowledge point to retrieve potentially relevant formulas.
- We feed the original problem, its official answer, the retrieved formulas, and (where available) multiple correct solution sketches into the LLM. The official answer and solution sketches are provided by the user, and providing solution sketches is optional.
- The LLM is prompted to:
  - identify the key conditions and the target quantity;
  - determine whether some of these conditions and the target can be interchanged—i.e., whether knowing the original answer allows us to infer some of the original conditions (an invertible relationship).
- When such an invertible relationship exists, the LLM is asked to write a Python solution program for the recomposed problem, where the original target now appears as an input condition and (a subset of) the original conditions become the new target.
- We then ask another LLM to inspect the generated Python code and verify that it implements a general computational procedure for solving the problem, rather than relying on hard-coded instance-specific outputs or trivial pattern matching.
- We import the LLM-generated Python code as a local module and call its `solve()` function, plugging the original answer value into the new "condition" slot and checking whether the returned output correctly recovers the original condition values.
- Any discrepancy or runtime error is fed back to the LLM for iterative refinement, just as in the numeric substitutions pipeline. We repeat this refinement–verification loop for up to five attempts and keep the Python code if it passes on the instance.

Once a correct solver for the recomposed version is obtained, we can further vary the new input variables within reasonable ranges to generate additional condition-recomposed variants.

This "knowledge-point extraction → formula retrieval → analyze → code → verify → resample" pipeline is identical across all problems.

---

**Algorithm 5** Conditional recompositions

---

1: **function** BUILDRECOMPOSEDSOLVER(*problem_text*, *answer_gold*, *solution_sketches*, *max_iter* = 5, *max_refine* = 5)
2:     *knowledge_points* ← LLM_EXTRACTKNOWLEDGEPOINTS(*problem_text*)
3:     *retrieved_formulas* ← RETRIEVE_FORMULAS(*knowledge_points*)
4:     *history* ← EMPTY_LIST
5:     *ANALYSIS_PROMPT* ← BUILDPROMPTINVERTIBLEANALYSIS(*problem_text*, *answer_gold*, *solution_sketches*, *retrieved_formulas*)
6:     *ANALYSIS_RESPONSE* ← LLM_CALL(*ANALYSIS_PROMPT*)
7:     (*invertible*, *cond_as_unknown*, *target_as_given*, *recomposed_problem_text*) ← PARSEINVERTIBLESTRUCTURE(*ANALYSIS_RESPONSE*)
8:     **if** *invertible* = FALSE **then**
9:         **return** FAILURE
10:     **end if**
11:     **for** *iter* = 1 **to** *max_iter* **do**
12:         *PROMPT* ← BUILDPROMPTRECOMPOSEDCODEGEN(origin_problem_text = *problem_text*, new_problem_text = *recomposed_problem_text*, origin_answer_gold = *answer_gold*, new_answer_gold = *cond_as_unknown.original_values*, solution_sketches = *solution_sketches*, retrieved_formulas = *retrieved_formulas*)
13:         APPEND(*history*, (*PROMPT*, NONE))
14:         (*CODE*, *value_ranges*) ← LLM_CALL(*PROMPT*)
15:         *is_hard_code* ← LLM_HARD_CODE_CHECK(*CODE*)
16:         **if** *is_hard_code* **then**
17:             **continue**
18:         **end if**
19:         **for** *refine_step* = 0 **to** *max_refine* **do**
20:             (*output*, *error*) ← RUN_PYTHON(*CODE*, input = *answer_gold*)
21:             APPEND(*history*, (*CODE*, (*output*, *error*)))
22:             **if** *error* = NONE **and** VERIFY(*output*, *cond_as_unknown.original_values*) **then**
23:                 **return** (*CODE*, *cond_as_unknown*, *target_as_given*, *value_ranges*, *recomposed_problem_text*)
24:             **end if**
25:             **if** *refine_step* = *max_refine* **then**
26:                 **break**
27:             **end if**
28:             *PROMPT_refine* ← BUILDPROMPTRECOMPOSEDCODEREFINE(problem_text = *recomposed_problem_text*, answer_gold = *cond_as_unknown.original_values*, history = *history*)
29:             (*CODE*, *value_ranges*) ← LLM_CALL(*PROMPT_refine*)
30:         **end for**
31:     **end for**
32:     **return** FAILURE
33: **end function**
34: **function** GENERATERECOMPOSEDVARIANTS(*problem_text*, *answer_gold*, *solution_sketches*)
35:     (*solver_code*, *cond_as_unknown*, *target_as_given*, *value_ranges*, *recomposed_problem_text*) ← BUILDRECOMPOSEDSOLVER(*problem_text*, *answer_gold*, *solution_sketches*)
36:     **if** *solver_code* = FAILURE **then**
37:         **return** NONE
38:     **end if**
39:     *new_param* ← SAMPLE(*value_ranges*)
40:     *new_problem_text* ← INSTANTIATERECOMPOSEDPROBLEMTEXT(*recomposed_problem_text*, *target_as_given*, *new_param*)
41:     (*output*, *error*) ← RUN_PYTHON(*solver_code*, input = *new_param*)
42:     *new_answer_gold* ← *output*
43:     **return** (*new_problem_text*, *new_answer_gold*)
44: **end function**

---

An example of conditional recompositions is given as follows.

- **Original:**

  Rectangles $ABCD$ and $EFGH$ are drawn such that $D, E, C, F$ are collinear. Also, $A, D, H, G$ all lie on a circle. If $BC = 16$, $AB = 107$, $FG = 17$, and $EF = 184$, what is the length of $CE$?

- **Conditional recomposition:**

Rectangles $ABCD$ and $EFGH$ are drawn such that $D, E, C, F$ are collinear. Also, $A, D, H, G$ all lie on a circle. If $BC = 16$, $AB = 107$, $CE = 104$, and $EF = 184$, what is the length of $FG$?

**Novel: recent-source adaptation**

In the "novel" branch, the first mechanism is recent-source adaptation, which is also fully scriptable:

1. We first use an LLM to extract the primary knowledge points tested by a given source problem.
2. We query open-access repositories of centralized exam questions that index items by region, year, subject, and knowledge point, and crawl the most recent 2025 exam problems matching the extracted knowledge points.
3. The retrieved problems are paraphrased by the LLM and can be further transformed using the three analogical methods (redundancy insertion, numeric substitution, and conditional recomposition).

This yields a set of new, recent-source problems that are structurally aligned at the knowledge level but clearly distinct in surface form and provenance. The entire workflow is driven by scripts and general prompts, without hand-curating individual items.

---

**Algorithm 6** Recent-Source Adaptation

---

1: **function** RECENTSOURCEADAPTATION(*problem_text*, *metadata*, *K*)
2:     *KP_PROMPT* ← BUILDPROMPTEXTRACTKNOWLEDGEPOINTS(*problem_text*, *metadata*)
3:     *KP_RESPONSE* ← LLM_CALL(*KP_PROMPT*)
4:     *KPs* ← PARSEKNOWLEDGEPOINTS(*KP_RESPONSE*)
5:     **if** *KPs* = NONE **then**
6:         **return** NONE
7:     **end if**
8:     *year_range* ← {2025}
9:     *candidate_item* ← RETRIEVE_EXAMS(knowledge_points = *KPs*, year_range = *year_range*, subject = *metadata.subject*)
10:     **if** *candidate_item* = NONE **then**
11:         **return** NONE
12:     **end if**
13:     *PARA_PROMPT* ← BUILDPROMPTPARAPHRASE(*candidate_item.text*, *candidate_item.answer_gold*, *KPs*)
14:     *PARA_RESPONSE* ← LLM_CALL(*PARA_PROMPT*)
15:     *paraphrased_item* ← PARSEPARAPHRASEDPROBLEM(*PARA_RESPONSE*)
16:     *adapted_item* ← {
17:         *text* : *paraphrased_item.text*,
18:         *answer* : *candidate_item.answer_gold*,
19:         *KPs* : *KPs*,
20:         *provenance* : {
21:             *source_exam* : *candidate_item.metadata*,
22:             *transform* : "paraphrase"
23:         }
24:     }
25:     **return** *adapted_item*
26: **end function**

---

For example, one recent-source adaptation question generated for the concept of *logarithms* is:

Given $2^{\log_2 a} = 3$ and $\log_5 5^b = 2$, find $a - b$

**Novel: conceptual synthesis**

The second "novel" mechanism is conceptual synthesis from authoritative textbooks. We first crawl a large collection of authoritative textbooks across different subjects from the web, and then use the LLM API's built-in functionality for parsing local PDF files to extract their content. Based on the

extracted content, we build a structured knowledge base in which each concept is associated with definitions, properties, theorems, phenomena, and canonical examples extracted from the textbooks.

1. Given a problem to be augmented, we use an LLM to identify its main knowledge points, and then retrieve the corresponding entries from the structured knowledge base. If the subject-specific knowledge base is missing, we trigger the textbook crawling and parsing step to expand the knowledge base, and then retrieve the corresponding entries from it.
2. Conditioned on these entries, the LLM is prompted to generate new conceptual questions targeting the underlying knowledge points, rather than copying any existing problem.

This pipeline turns textbook content into fresh conceptual questions that align with the original topic but are novel in form and focus.

---

**Algorithm 7** Conceptual Synthesis

---

1: **function** CONCEPTUALSYNTHESIS(*problem_text*, *metadata*)
2:     *KP_PROMPT* ← BUILDPROMPTEXTRACTKNOWLEDGEPOINTS(*problem_text*, *metadata*)
3:     *KP_RESPONSE* ← LLM_CALL(*KP_PROMPT*)
4:     *KPs* ← PARSEKNOWLEDGEPOINTS(*KP_RESPONSE*)
5:     **if** *KPs* = NONE **then**
6:         **return** NONE
7:     **end if**
8:     *kb_entry* ← RETRIEVE_KB_ENTRY(knowledge_points = *KPs*, subject = *metadata.subject*)
9:     **if** *kb_entry* = NONE **then**
10:         CRAWL_AND_PARSE_TEXTBOOKS(knowledge_points = *KPs*, subject = *metadata.subject*)
11:         *kb_entry* ← RETRIEVE_KB_ENTRY(knowledge_points = *KPs*, subject = *metadata.subject*)
12:         **if** *kb_entry* = NONE **then**
13:             **return** NONE
14:         **end if**
15:     **end if**
16:     *GEN_PROMPT* ← BUILDPROMPTCONCEPTUALQUESTIONGENERATION(*kb_entry*, *KPs*)
17:     *GEN_RESPONSE* ← LLM_CALL(*GEN_PROMPT*)
18:     *raw_item* ← PARSEGENERATEDCONCEPTUALQUESTIONS(*GEN_RESPONSE*)
19:     *conceptual_item* ← {
20:         *text* : *raw_item.text*,
21:         *answer* : *raw_item.answer*,
22:         *KPs* : *KPs*
23:     }
24:     **return** *conceptual_item*
25: **end function**

---

For example, one conceptual-synthesis question generated for the concept of *logarithms* is:

What kind of mathematical idea/method turns exponentiation and multiplication into multiplication and addition?

A.4 RESULTS ON MMLU, GPQA

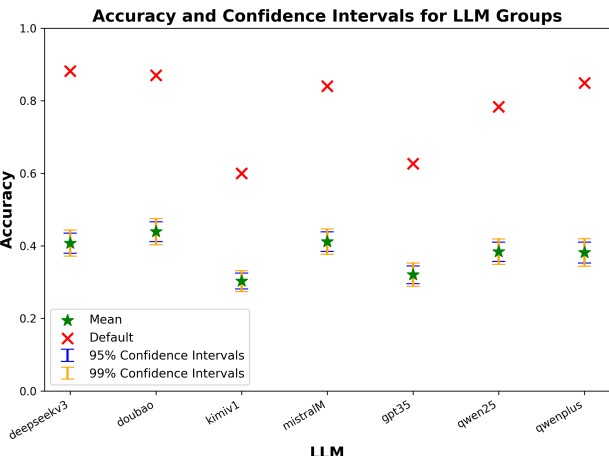

Figure 6: Accuracy confidence intervals of different LLMs on MMLU

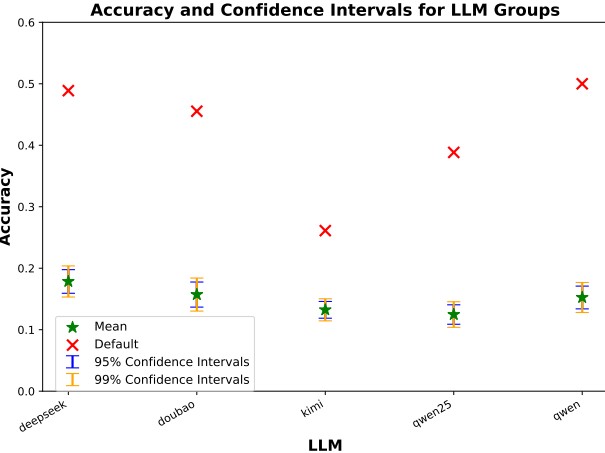

Figure 7: Accuracy confidence intervals of different LLMs on GPQA

## A.5 ANOVA ANALYSIS RESULTS ON DIFFERENT LLMs

Table 6: Complete ANOVA results on LLMs (sorted by effect size in descending order)

(a) Complete ANOVA results on DeepSeek-V3

| Factor | Effect Size $\eta^2$ | $p$-value |
|---|---|---|
| Question Format | 0.399643 | 0.000 |
| Question Format-COT | 0.161394 | 0.000 |
| COT | 0.080156 | 0.000 |
| max_tokens | 0.028099 | 0.000 |
| Question Format-Shot | 0.011101 | 0.000 |
| Language-Question Format | 0.008178 | 0.006 |
| COT-max_tokens | 0.006721 | 0.010 |
| Language-COT | 0.004345 | 0.038 |
| Multi Turn-max_tokens | 0.003841 | 0.050 |
| Language | 0.003841 | 0.046 |
| Shot-max_tokens | 0.003669 | 0.046 |
| Question Format-max_tokens | 0.002687 | 0.100 |
| Language-Multi Turn | 0.002600 | 0.066 |
| temperature-top_p | 0.002082 | 0.178 |
| Question Format-Multi Turn | 0.001321 | 0.244 |
| Question Paraphrase | 0.001240 | 0.252 |
| Question Format-temperature | 0.001201 | 0.262 |
| Shot-temperature | 0.000926 | 0.326 |
| Multi Turn | 0.000793 | 0.364 |
| temperature | 0.000587 | 0.396 |
| COT-Multi Turn | 0.000587 | 0.466 |
| COT-top_p | 0.000573 | 0.482 |
| Language-Shot | 0.000534 | 0.466 |
| presence_penalty | 0.000435 | 0.488 |
| Question Format-Question Paraphrase | 0.000411 | 0.562 |
| Question Paraphrase-max_tokens | 0.000207 | 0.626 |
| Language-Question Paraphrase | 0.000198 | 0.660 |
| Shot-COT | 0.000161 | 0.646 |
| COT-temperature | 0.000140 | 0.698 |
| Language-max_tokens | 0.000140 | 0.712 |
| COT-presence_penalty | 0.000140 | 0.668 |
| Shot | 0.000134 | 0.706 |
| Question Format-presence_penalty | 0.000133 | 0.708 |
| Shot-top_p | 0.000109 | 0.736 |
| max_tokens-presence_penalty | 0.000092 | 0.768 |
| max_tokens-top_p | 0.000086 | 0.790 |
| top_p | 0.000058 | 0.812 |
| Shot-Multi Turn | 0.000054 | 0.802 |
| temperature-max_tokens | 0.000038 | 0.836 |
| Language-presence_penalty | 0.000035 | 0.854 |
| top_p-presence_penalty | 0.000032 | 0.902 |
| Multi Turn-temperature | 0.000029 | 0.874 |
| Multi Turn-top_p | 0.000026 | 0.884 |
| Question Paraphrase-top_p | 0.000018 | 0.898 |
| temperature-presence_penalty | 0.000013 | 0.902 |
| Question Paraphrase-presence_penalty | 0.000011 | 0.910 |
| Language-top_p | 0.000008 | 0.942 |
| Question Paraphrase-Shot | 0.000006 | 0.930 |
| Multi Turn-presence_penalty | 0.000004 | 0.946 |
| Language-temperature | 0.000004 | 0.940 |
| Question Format-top_p | 0.000003 | 0.962 |
| Question Paraphrase-Multi Turn | 0.000003 | 0.972 |
| Shot-presence_penalty | 0.000001 | 0.966 |
| Question Paraphrase-COT | 0.000001 | 0.996 |
| Question Paraphrase-temperature | 0.000000 | 0.994 |

(b) Complete ANOVA results on Doubao-1.5-pro-32k

| Factor | Effect Size $\eta^2$ | $p$-value |
|---|---|---|
| Question Format | 0.467626 | 0.000 |
| Question Format-COT | 0.259657 | 0.000 |
| COT | 0.130549 | 0.000 |
| max_tokens | 0.009192 | 0.002 |
| COT-max_tokens | 0.008943 | 0.006 |
| max_tokens-presence_penalty | 0.000671 | 0.388 |
| Shot | 0.000638 | 0.408 |
| Shot-Multi Turn | 0.000605 | 0.424 |
| Question Paraphrase-max_tokens | 0.000502 | 0.466 |
| Multi Turn | 0.000473 | 0.466 |
| Multi Turn-max_tokens | 0.000464 | 0.472 |
| Language-temperature | 0.000455 | 0.468 |
| COT-Multi Turn | 0.000427 | 0.496 |
| Question Format-Shot | 0.000400 | 0.546 |
| Question Format-max_tokens | 0.000392 | 0.538 |
| temperature | 0.000392 | 0.534 |
| Question Format-temperature | 0.000358 | 0.530 |
| Question Paraphrase-presence_penalty | 0.000349 | 0.552 |
| Language-top_p | 0.000326 | 0.584 |
| temperature-top_p | 0.000326 | 0.588 |
| Language-presence_penalty | 0.000310 | 0.578 |
| Language-Multi Turn | 0.000295 | 0.562 |
| Multi Turn-presence_penalty | 0.000272 | 0.604 |
| Language-COT | 0.000265 | 0.586 |
| Shot-COT | 0.000224 | 0.628 |
| Question Paraphrase-top_p | 0.000205 | 0.642 |
| Question Format-Multi Turn | 0.000158 | 0.704 |
| COT-presence_penalty | 0.000147 | 0.696 |
| max_tokens-top_p | 0.000147 | 0.674 |
| presence_penalty | 0.000117 | 0.730 |
| temperature-max_tokens | 0.000108 | 0.740 |
| Question Format-presence_penalty | 0.000090 | 0.764 |
| Question Format-Question Paraphrase | 0.000067 | 0.792 |
| Question Paraphrase | 0.000054 | 0.836 |
| Shot-top_p | 0.000047 | 0.818 |
| Shot-presence_penalty | 0.000047 | 0.792 |
| Language | 0.000045 | 0.814 |
| Language-Question Paraphrase | 0.000044 | 0.840 |
| Multi Turn-temperature | 0.000036 | 0.832 |
| COT-top_p | 0.000036 | 0.854 |
| COT-temperature | 0.000034 | 0.834 |
| Language-Shot | 0.000034 | 0.850 |
| Shot-max_tokens | 0.000024 | 0.864 |
| temperature-presence_penalty | 0.000015 | 0.906 |
| top_p-presence_penalty | 0.000013 | 0.902 |
| Question Paraphrase-COT | 0.000007 | 0.908 |
| Language-Question Format | 0.000002 | 0.956 |
| Question Paraphrase-temperature | 0.000001 | 0.960 |
| Shot-temperature | 0.000001 | 0.962 |
| Question Paraphrase-Multi Turn | 0.000001 | 0.966 |
| Language-max_tokens | 0.000001 | 0.970 |
| Question Paraphrase-Shot | 0.000001 | 0.980 |
| top_p | 0.000001 | 0.970 |
| Question Format-top_p | 0.000000 | 0.984 |
| Multi Turn-top_p | 0.000000 | 0.992 |

(c) Complete ANOVA results on GPT-3.5

| Factor | Effect Size ($\eta^2$) | $p$-value |
|---|---|---|
| Question Format | 0.417428 | 0.000000 |
| Question Format-COT | 0.199209 | 0.000000 |
| COT | 0.199208 | 0.000000 |
| temperature | 0.006046 | 0.016000 |
| Question Format-temperature | 0.005781 | 0.020000 |
| Shot-Multi Turn | 0.005715 | 0.010000 |
| Shot-COT | 0.005651 | 0.026000 |
| max_tokens | 0.005396 | 0.024000 |
| Question Format-max_tokens | 0.004434 | 0.044000 |
| temperature-top_p | 0.004320 | 0.052000 |
| top_p | 0.003119 | 0.074000 |
| Question Format-top_p | 0.002570 | 0.104000 |
| COT-max_tokens | 0.001773 | 0.170000 |
| Question Format-Shot | 0.000853 | 0.344000 |
| COT-Multi Turn | 0.000828 | 0.352000 |
| Language-Multi Turn | 0.000779 | 0.364000 |
| Language | 0.000687 | 0.428000 |
| Language-Question Format | 0.000600 | 0.484000 |
| COT-top_p | 0.000443 | 0.496000 |
| COT-temperature | 0.000425 | 0.508000 |
| COT-presence_penalty | 0.000408 | 0.508000 |
| Question Paraphrase-Multi Turn | 0.000407 | 0.534000 |
| Question Paraphrase-COT | 0.000391 | 0.506000 |
| Shot | 0.000296 | 0.578000 |
| Shot-temperature | 0.000281 | 0.598000 |
| Language-Shot | 0.000253 | 0.598000 |
| max_tokens-presence_penalty | 0.000201 | 0.628000 |
| Question Paraphrase-top_p | 0.000177 | 0.696000 |
| Shot-max_tokens | 0.000135 | 0.720000 |
| Language-top_p | 0.000115 | 0.690000 |
| Question Paraphrase-presence_penalty | 0.000115 | 0.714000 |
| Language-temperature | 0.000106 | 0.750000 |
| Language-max_tokens | 0.000106 | 0.740000 |
| Shot-presence_penalty | 0.000089 | 0.786000 |
| Multi Turn-temperature | 0.000081 | 0.784000 |
| Language-COT | 0.000074 | 0.766000 |
| Multi Turn-presence_penalty | 0.000074 | 0.788000 |
| Question Format-Question Paraphrase | 0.000067 | 0.776000 |
| Question Paraphrase | 0.000053 | 0.808000 |
| Question Format-Multi Turn | 0.000047 | 0.826000 |
| presence_penalty | 0.000047 | 0.854000 |
| temperature-max_tokens | 0.000036 | 0.858000 |
| Question Paraphrase-max_tokens | 0.000036 | 0.854000 |
| temperature-presence_penalty | 0.000031 | 0.884000 |
| Language-Question Paraphrase | 0.000027 | 0.858000 |
| Question Format-presence_penalty | 0.000027 | 0.900000 |
| Question Paraphrase-temperature | 0.000027 | 0.882000 |
| Multi Turn | 0.000018 | 0.872000 |
| Multi Turn-max_tokens | 0.000009 | 0.916000 |
| Question Paraphrase-Shot | 0.000009 | 0.920000 |
| Multi Turn-top_p | 0.000007 | 0.942000 |
| top_p-presence_penalty | 0.000007 | 0.948000 |
| Shot-top_p | 0.000007 | 0.942000 |
| Language-presence_penalty | 0.000002 | 0.970000 |
| max_tokens-top_p | 0.000000 | 0.978000 |

(d) Complete ANOVA results on GPT-4.1

| Factor | Effect Size ($\eta^2$) | $p$-value |
|---|---|---|
| Question Format | 0.289086 | 0.000000 |
| Question Format-COT | 0.180162 | 0.000000 |
| COT-max_tokens | 0.054845 | 0.000000 |
| max_tokens | 0.053399 | 0.000000 |
| COT | 0.027181 | 0.000000 |
| temperature-top_p | 0.006685 | 0.010000 |
| Question Format-Shot | 0.006529 | 0.030000 |
| temperature | 0.005619 | 0.016000 |
| Shot | 0.004963 | 0.028000 |
| max_tokens-top_p | 0.004019 | 0.044000 |
| Language-max_tokens | 0.003907 | 0.062000 |
| Question Format-temperature | 0.003732 | 0.072000 |
| top_p | 0.003364 | 0.098000 |
| temperature-max_tokens | 0.002990 | 0.140000 |
| Question Paraphrase-Shot | 0.002362 | 0.160000 |
| Question Paraphrase-presence_penalty | 0.001939 | 0.194000 |
| Shot-Multi Turn | 0.001842 | 0.194000 |
| COT-temperature | 0.001708 | 0.230000 |
| Shot-COT | 0.001589 | 0.290000 |
| Language-COT | 0.001517 | 0.246000 |
| COT-presence_penalty | 0.001406 | 0.276000 |
| Question Format-max_tokens | 0.001360 | 0.306000 |
| temperature-presence_penalty | 0.001284 | 0.292000 |
| Language-Shot | 0.001276 | 0.262000 |
| Language-Question Paraphrase | 0.001188 | 0.324000 |
| presence_penalty | 0.001184 | 0.362000 |
| COT-top_p | 0.001183 | 0.300000 |
| COT-Multi Turn | 0.001037 | 0.376000 |
| Multi Turn-max_tokens | 0.000988 | 0.376000 |
| Shot-top_p | 0.000754 | 0.422000 |
| Question Paraphrase-max_tokens | 0.000703 | 0.438000 |
| Multi Turn-top_p | 0.000367 | 0.574000 |
| Question Format-presence_penalty | 0.000348 | 0.584000 |
| Question Format-Question Paraphrase | 0.000333 | 0.584000 |
| Language-presence_penalty | 0.000329 | 0.586000 |
| max_tokens-presence_penalty | 0.000324 | 0.600000 |
| top_p-presence_penalty | 0.000253 | 0.618000 |
| Shot-presence_penalty | 0.000192 | 0.702000 |
| Shot-max_tokens | 0.000164 | 0.724000 |
| Question Paraphrase | 0.000146 | 0.746000 |
| Question Format-top_p | 0.000134 | 0.728000 |
| Shot-temperature | 0.000133 | 0.724000 |
| Question Format-Multi Turn | 0.000132 | 0.718000 |
| Question Paraphrase-top_p | 0.000065 | 0.798000 |
| Multi Turn-presence_penalty | 0.000055 | 0.828000 |
| Language-top_p | 0.000023 | 0.896000 |
| Language-Question Format | 0.000022 | 0.910000 |
| Multi Turn-temperature | 0.000020 | 0.886000 |
| Question Paraphrase-temperature | 0.000010 | 0.922000 |
| Language-Multi Turn | 0.000009 | 0.906000 |
| Question Paraphrase-COT | 0.000009 | 0.922000 |
| Question Paraphrase-Multi Turn | 0.000005 | 0.918000 |
| Language | 0.000001 | 0.964000 |
| Multi Turn | 0.000000 | 0.996000 |
| Language-temperature | 0.000000 | 0.998000 |

(e) Complete ANOVA results on Qwen2.5

| Factor | Effect Size ($\eta^2$) | $p$-value |
| --- | --- | --- |
| Question Format | 0.454352 | 0.000000 |
| Question Format-COT | 0.204235 | 0.000000 |
| COT | 0.200224 | 0.000000 |
| Question Format-Shot | 0.009265 | 0.000000 |
| Shot | 0.007983 | 0.008000 |
| Shot-COT | 0.006715 | 0.002000 |
| Multi Turn | 0.003717 | 0.046000 |
| Question Format-Multi Turn | 0.003657 | 0.052000 |
| max_tokens | 0.002764 | 0.080000 |
| Language-Question Format | 0.002585 | 0.128000 |
| COT-max_tokens | 0.001865 | 0.164000 |
| Language | 0.001720 | 0.196000 |
| COT-Multi Turn | 0.001540 | 0.214000 |
| Multi Turn-max_tokens | 0.001110 | 0.322000 |
| Question Format-max_tokens | 0.000922 | 0.292000 |
| Language-COT | 0.000848 | 0.382000 |
| Question Paraphrase-presence_penalty | 0.000710 | 0.400000 |
| Language-Multi Turn | 0.000538 | 0.478000 |
| temperature | 0.000430 | 0.444000 |
| Shot-temperature | 0.000380 | 0.534000 |
| Question Format-temperature | 0.000371 | 0.550000 |
| Language-Question Paraphrase | 0.000343 | 0.550000 |
| temperature-presence_penalty | 0.000334 | 0.566000 |
| Language-temperature | 0.000290 | 0.602000 |
| top_p | 0.000257 | 0.600000 |
| max_tokens-top_p | 0.000242 | 0.618000 |
| Shot-Multi Turn | 0.000234 | 0.654000 |
| COT-temperature | 0.000219 | 0.660000 |
| Question Format-Question Paraphrase | 0.000165 | 0.674000 |
| Question Paraphrase-top_p | 0.000165 | 0.672000 |
| Language-max_tokens | 0.000165 | 0.674000 |
| Question Paraphrase | 0.000118 | 0.742000 |
| Language-top_p | 0.000107 | 0.760000 |
| temperature-top_p | 0.000107 | 0.724000 |
| Multi Turn-presence_penalty | 0.000083 | 0.762000 |
| max_tokens-presence_penalty | 0.000075 | 0.784000 |
| Shot-max_tokens | 0.000062 | 0.770000 |
| presence_penalty | 0.000051 | 0.830000 |
| Question Format-top_p | 0.000038 | 0.824000 |
| top_p-presence_penalty | 0.000038 | 0.830000 |
| Multi Turn-temperature | 0.000029 | 0.856000 |
| Question Paraphrase-temperature | 0.000027 | 0.882000 |
| COT-presence_penalty | 0.000022 | 0.846000 |
| Multi Turn-top_p | 0.000022 | 0.882000 |
| Question Format-presence_penalty | 0.000022 | 0.866000 |
| Language-presence_penalty | 0.000012 | 0.912000 |
| temperature-max_tokens | 0.000012 | 0.918000 |
| Question Paraphrase-Multi Turn | 0.000009 | 0.942000 |
| Language-Shot | 0.000007 | 0.926000 |
| Question Paraphrase-Shot | 0.000005 | 0.928000 |
| COT-top_p | 0.000003 | 0.954000 |
| Shot-presence_penalty | 0.000002 | 0.960000 |
| Shot-top_p | 0.000001 | 0.982000 |
| Question Paraphrase-COT | 0.000000 | 0.984000 |
| Question Paraphrase-max_tokens | 0.000000 | 0.990000 |

(f) Complete ANOVA results on Qwen Plus

| Factor | Effect Size ($\eta^2$) | $p$-value |
| --- | --- | --- |
| Question Format | 0.302717 | 0.000000 |
| Question Format-COT | 0.259678 | 0.000000 |
| COT | 0.098747 | 0.000000 |
| Shot | 0.047447 | 0.000000 |
| Question Format-Shot | 0.042448 | 0.000000 |
| Shot-COT | 0.024956 | 0.000000 |
| COT-max_tokens | 0.016596 | 0.000000 |
| max_tokens | 0.016280 | 0.000000 |
| Language | 0.005019 | 0.024000 |
| Language-Question Format | 0.003272 | 0.060000 |
| temperature-top_p | 0.002324 | 0.112000 |
| Multi Turn-max_tokens | 0.001979 | 0.154000 |
| Question Format-temperature | 0.001662 | 0.202000 |
| top_p | 0.001282 | 0.250000 |
| Language-Shot | 0.000991 | 0.296000 |
| Question Paraphrase-Multi Turn | 0.000877 | 0.320000 |
| COT-temperature | 0.000822 | 0.368000 |
| Multi Turn | 0.000736 | 0.428000 |
| Question Format-Question Paraphrase | 0.000654 | 0.458000 |
| temperature | 0.000608 | 0.412000 |
| Language-COT | 0.000519 | 0.472000 |
| Shot-Multi Turn | 0.000438 | 0.512000 |
| Multi Turn-presence_penalty | 0.000400 | 0.512000 |
| Shot-presence_penalty | 0.000375 | 0.496000 |
| Question Format-top_p | 0.000275 | 0.604000 |
| COT-Multi Turn | 0.000245 | 0.612000 |
| Question Format-max_tokens | 0.000226 | 0.648000 |
| Question Format-Multi Turn | 0.000166 | 0.708000 |
| Multi Turn-temperature | 0.000135 | 0.724000 |
| max_tokens-presence_penalty | 0.000128 | 0.690000 |
| top_p-presence_penalty | 0.000102 | 0.752000 |
| Question Paraphrase-Shot | 0.000101 | 0.746000 |
| COT-top_p | 0.000090 | 0.758000 |
| Language-presence_penalty | 0.000084 | 0.792000 |
| Language-max_tokens | 0.000073 | 0.762000 |
| temperature-presence_penalty | 0.000058 | 0.786000 |
| Language-Multi Turn | 0.000058 | 0.806000 |
| Question Paraphrase-top_p | 0.000058 | 0.814000 |
| Language-temperature | 0.000049 | 0.816000 |
| Shot-top_p | 0.000037 | 0.838000 |
| Question Paraphrase | 0.000033 | 0.864000 |
| Shot-temperature | 0.000033 | 0.860000 |
| Language-top_p | 0.000029 | 0.850000 |
| Question Paraphrase-max_tokens | 0.000023 | 0.862000 |
| Question Paraphrase-presence_penalty | 0.000015 | 0.902000 |
| Question Paraphrase-temperature | 0.000015 | 0.904000 |
| Multi Turn-top_p | 0.000015 | 0.924000 |
| Shot-max_tokens | 0.000013 | 0.898000 |
| COT-presence_penalty | 0.000013 | 0.918000 |
| Question Format-presence_penalty | 0.000005 | 0.946000 |
| temperature-max_tokens | 0.000002 | 0.970000 |
| presence_penalty | 0.000002 | 0.960000 |
| max_tokens-top_p | 0.000002 | 0.970000 |
| Question Paraphrase-COT | 0.000001 | 0.978000 |
| Language-Question Paraphrase | 0.000000 | 0.994000 |

(g) Complete ANOVA results on Mistral Large

| Factor | Effect Size ($\eta^2$) | $p$-value |
| --- | --- | --- |
| Question Format | 0.406873 | 0.000000 |
| Question Format-COT | 0.268919 | 0.000000 |
| COT | 0.075852 | 0.000000 |
| COT-max_tokens | 0.026017 | 0.000000 |
| max_tokens | 0.025372 | 0.000000 |
| Question Format-Multi Turn | 0.004796 | 0.024000 |
| Multi Turn | 0.003609 | 0.058000 |
| Question Format-Shot | 0.003338 | 0.068000 |
| COT-Multi Turn | 0.002708 | 0.100000 |
| Question Format-max_tokens | 0.001379 | 0.230000 |
| Shot | 0.001023 | 0.320000 |
| Language-Question Format | 0.001004 | 0.280000 |
| Multi Turn-max_tokens | 0.000881 | 0.310000 |
| Shot-Multi Turn | 0.000830 | 0.368000 |
| Language-Multi Turn | 0.000766 | 0.372000 |
| Language-COT | 0.000765 | 0.356000 |
| Question Format-Question Paraphrase | 0.000644 | 0.414000 |
| Multi Turn-presence_penalty | 0.000456 | 0.458000 |
| Question Paraphrase | 0.000364 | 0.550000 |
| Question Paraphrase-presence_penalty | 0.000353 | 0.486000 |
| max_tokens-presence_penalty | 0.000272 | 0.620000 |
| Shot-top_p | 0.000244 | 0.638000 |
| Multi Turn-top_p | 0.000244 | 0.570000 |
| Language-Question Paraphrase | 0.000235 | 0.618000 |
| Question Format-presence_penalty | 0.000227 | 0.604000 |
| COT-temperature | 0.000227 | 0.628000 |
| Question Paraphrase-Shot | 0.000193 | 0.658000 |
| temperature | 0.000185 | 0.644000 |
| Shot-presence_penalty | 0.000178 | 0.650000 |
| Question Format-temperature | 0.000178 | 0.672000 |
| Question Paraphrase-top_p | 0.000163 | 0.686000 |
| Shot-max_tokens | 0.000163 | 0.682000 |
| top_p | 0.000142 | 0.690000 |
| Question Paraphrase-max_tokens | 0.000135 | 0.688000 |
| Question Paraphrase-Multi Turn | 0.000128 | 0.772000 |
| max_tokens-top_p | 0.000110 | 0.712000 |
| Language | 0.000109 | 0.736000 |
| COT-presence_penalty | 0.000087 | 0.780000 |
| Shot-temperature | 0.000087 | 0.810000 |
| Question Paraphrase-COT | 0.000049 | 0.818000 |
| top_p-presence_penalty | 0.000049 | 0.824000 |
| presence_penalty | 0.000038 | 0.836000 |
| Language-temperature | 0.000035 | 0.840000 |
| Language-top_p | 0.000028 | 0.864000 |
| temperature-max_tokens | 0.000017 | 0.884000 |
| temperature-presence_penalty | 0.000015 | 0.884000 |
| Multi Turn-temperature | 0.000007 | 0.928000 |
| temperature-top_p | 0.000006 | 0.928000 |
| Question Paraphrase-temperature | 0.000005 | 0.950000 |
| COT-top_p | 0.000003 | 0.952000 |
| Language-Shot | 0.000003 | 0.944000 |
| Question Format-top_p | 0.000002 | 0.952000 |
| Language-max_tokens | 0.000002 | 0.950000 |
| Language-presence_penalty | 0.000002 | 0.960000 |
| Shot-COT | 0.000000 | 0.998000 |

(h) Complete ANOVA results on Mistral Medium

| Factor | Effect Size ($\eta^2$) | $p$-value |
| --- | --- | --- |
| Question Format | 0.430500 | 0.000000 |
| Question Format-COT | 0.274248 | 0.000000 |
| COT | 0.105919 | 0.000000 |
| COT-max_tokens | 0.013038 | 0.002000 |
| max_tokens | 0.012064 | 0.000000 |
| Question Format-Multi Turn | 0.007865 | 0.004000 |
| Multi Turn | 0.005334 | 0.026000 |
| Shot | 0.003097 | 0.064000 |
| COT-Multi Turn | 0.003001 | 0.090000 |
| Question Format-Shot | 0.002782 | 0.086000 |
| Multi Turn-max_tokens | 0.001354 | 0.252000 |
| Shot-Multi Turn | 0.000908 | 0.336000 |
| Language-Question Paraphrase | 0.000891 | 0.378000 |
| Shot-COT | 0.000636 | 0.434000 |
| Question Paraphrase-top_p | 0.000366 | 0.476000 |
| Question Paraphrase-presence_penalty | 0.000312 | 0.534000 |
| Question Paraphrase-Multi Turn | 0.000282 | 0.578000 |
| Language | 0.000254 | 0.626000 |
| max_tokens-presence_penalty | 0.000227 | 0.650000 |
| Language-Multi Turn | 0.000219 | 0.656000 |
| Multi Turn-presence_penalty | 0.000218 | 0.642000 |
| Question Format-temperature | 0.000178 | 0.682000 |
| Multi Turn-top_p | 0.000163 | 0.672000 |
| Language-COT | 0.000163 | 0.690000 |
| temperature-top_p | 0.000148 | 0.698000 |
| Question Format-max_tokens | 0.000128 | 0.672000 |
| Question Format-Question Paraphrase | 0.000109 | 0.746000 |
| Question Paraphrase-max_tokens | 0.000103 | 0.754000 |
| COT-top_p | 0.000091 | 0.750000 |
| COT-temperature | 0.000081 | 0.764000 |
| Shot-temperature | 0.000076 | 0.806000 |
| top_p | 0.000076 | 0.784000 |
| Question Paraphrase-temperature | 0.000076 | 0.758000 |
| Language-temperature | 0.000061 | 0.798000 |
| Question Format-presence_penalty | 0.000053 | 0.806000 |
| Shot-top_p | 0.000048 | 0.808000 |
| Language-Shot | 0.000044 | 0.832000 |
| Shot-max_tokens | 0.000044 | 0.814000 |
| Question Paraphrase-COT | 0.000037 | 0.854000 |
| presence_penalty | 0.000037 | 0.844000 |
| Question Paraphrase | 0.000034 | 0.850000 |
| Language-top_p | 0.000024 | 0.870000 |
| Language-presence_penalty | 0.000022 | 0.874000 |
| Shot-presence_penalty | 0.000019 | 0.888000 |
| temperature-presence_penalty | 0.000014 | 0.910000 |
| Language-max_tokens | 0.000008 | 0.920000 |
| Language-Question Format | 0.000007 | 0.938000 |
| Question Format-top_p | 0.000005 | 0.942000 |
| temperature | 0.000005 | 0.948000 |
| max_tokens-top_p | 0.000004 | 0.958000 |
| Question Paraphrase-Shot | 0.000003 | 0.956000 |
| Multi Turn-temperature | 0.000001 | 0.972000 |
| temperature-max_tokens | 0.000001 | 0.968000 |
| COT-presence_penalty | 0.000000 | 0.986000 |
| top_p-presence_penalty | 0.000000 | 0.994000 |

### (i) Complete ANOVA results on Moonshot-v1

| Factor | Effect Size ($\eta^2$) | $p$-value |
|---|---|---|
| Question Format | 0.297180 | 0.000000 |
| Question Format-COT | 0.147641 | 0.000000 |
| COT | 0.130758 | 0.000000 |
| Shot-COT | 0.077565 | 0.000000 |
| Question Format-Shot | 0.042631 | 0.000000 |
| Shot | 0.038198 | 0.000000 |
| COT-Multi Turn | 0.018078 | 0.000000 |
| Language-Shot | 0.005572 | 0.016000 |
| COT-max_tokens | 0.002887 | 0.090000 |
| Language-Multi Turn | 0.002824 | 0.084000 |
| Language | 0.002702 | 0.110000 |
| Question Format-Multi Turn | 0.002700 | 0.106000 |
| max_tokens | 0.002409 | 0.118000 |
| Language-Question Format | 0.002297 | 0.108000 |
| Multi Turn | 0.002185 | 0.124000 |
| Multi Turn-max_tokens | 0.001677 | 0.166000 |
| Question Format-max_tokens | 0.000966 | 0.300000 |
| Language-max_tokens | 0.000930 | 0.356000 |
| Shot-max_tokens | 0.000861 | 0.380000 |
| Language-Question Paraphrase | 0.000793 | 0.360000 |
| COT-presence_penalty | 0.000402 | 0.494000 |
| top_p-presence_penalty | 0.000314 | 0.554000 |
| Multi Turn-temperature | 0.000274 | 0.578000 |
| max_tokens-top_p | 0.000219 | 0.662000 |
| Shot-presence_penalty | 0.000203 | 0.620000 |
| top_p | 0.000142 | 0.694000 |
| Question Format-top_p | 0.000128 | 0.742000 |
| Question Paraphrase-max_tokens | 0.000128 | 0.724000 |
| Question Paraphrase-COT | 0.000115 | 0.756000 |
| Shot-temperature | 0.000092 | 0.786000 |
| max_tokens-presence_penalty | 0.000081 | 0.770000 |
| Shot-top_p | 0.000081 | 0.792000 |
| temperature-top_p | 0.000081 | 0.776000 |
| Question Paraphrase-Shot | 0.000081 | 0.804000 |
| Language-temperature | 0.000081 | 0.782000 |
| temperature-presence_penalty | 0.000053 | 0.860000 |
| Question Paraphrase-Multi Turn | 0.000053 | 0.858000 |
| Shot-Multi Turn | 0.000045 | 0.812000 |
| Question Format-temperature | 0.000037 | 0.856000 |
| COT-top_p | 0.000037 | 0.866000 |
| Question Paraphrase-presence_penalty | 0.000037 | 0.882000 |
| Multi Turn-presence_penalty | 0.000037 | 0.872000 |
| Question Format-presence_penalty | 0.000030 | 0.878000 |
| Question Paraphrase | 0.000024 | 0.864000 |
| Language-presence_penalty | 0.000019 | 0.904000 |
| Language-COT | 0.000019 | 0.884000 |
| Multi Turn-top_p | 0.000014 | 0.922000 |
| temperature-max_tokens | 0.000002 | 0.958000 |
| COT-temperature | 0.000002 | 0.954000 |
| Question Format-Question Paraphrase | 0.000002 | 0.968000 |
| Question Paraphrase-temperature | 0.000001 | 0.966000 |
| temperature | 0.000001 | 0.970000 |
| Language-top_p | 0.000001 | 0.968000 |
| presence_penalty | 0.000000 | 0.992000 |
| Question Paraphrase-top_p | 0.000000 | 0.998000 |

## A.6 Latency and Accuracy Analysis of LLMs

In this section, we analyze the relationships between the latency and accuracy of LLMs, as well as between latency and hardware architectures, based on online evaluation. On one hand, we conduct online testing to assess the accuracy of LLMs under different configuration spaces in terms of the "tail to quality" (Yang et al., 2022) metric. Here, "tail to quality" refers to the ratio of the number of tasks correctly completed within a specified threshold to the total number of tasks. Figure 8a illustrates the performance of various LLMs under the "Tail to Quality" metric, showing how their quality scores evolve across different threshold values. Among the models, deepseek (green curve) consistently demonstrates the highest quality across all thresholds, outperforming the others. Doubao (blue curve) and qwen (gray curve) follow, with doubao approaching deepseek's performance at higher thresholds. Kimi (brown curve) and qwen25 (cyan curve) exhibit relatively lower quality, though qwen25 shows rapid improvement at lower thresholds before plateauing. Overall, the chart highlights deepseek's superior capability in handling tail data, while qwen25's growth in quality becomes limited at higher thresholds.

On the other hand, following a similar approach as for accuracy, we obtain the 95% and 99% confidence intervals for latency, as shown in Figure 8b. It can be seen that, for most models, latency and accuracy on AIME'2024 are positively correlated. Notably, doubao-1.5-pro and qwen2.5 achieve relatively low latency while maintaining high accuracy. In contrast, gpt-4.1 and qwen-plus exhibit the opposite trend: they achieve lower accuracy despite higher latency.

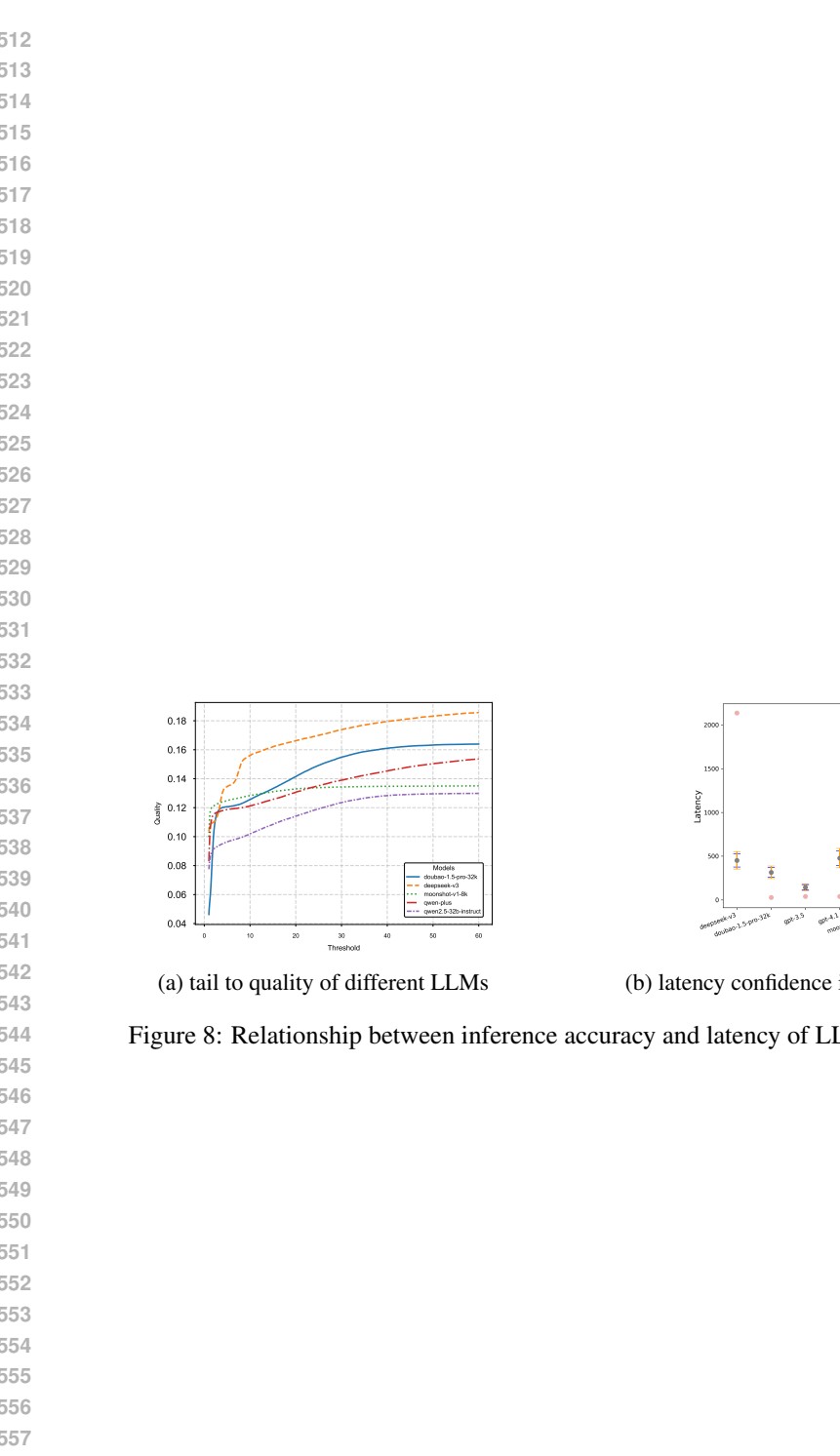

(a) tail to quality of different LLMs      (b) latency confidence intervals of different LLMs

Figure 8: Relationship between inference accuracy and latency of LLMs in online testing

