# OpenReview forum: "Beyond Benchmarks: Toward Causally Faithful Evaluation of Large Language Models"
_ICLR.cc/2026/Conference — Submitted to ICLR 2026_

### Official Review · Reviewer_6AVH · 2025-10-25

**Soundness:** 1
**Presentation:** 2
**Contribution:** 2
**Rating:** 2
**Confidence:** 4

**Summary:**

This paper proposes a framework called LLM Evaluatology that aims to disentangle a model’s intrinsic capability from the confounding effects of evaluation configurations such as decoding parameters and prompt settings. To this end, the authors introduce the concept of a Minimal Evaluation System (MES) for in-distribution testing and an augmented variant (A-MES) designed to assess robustness under distributional shifts. They further perform an ANOVA analysis to quantify how individual components (e.g., temperature, top-p, max tokens) contribute to performance variance across large language models.

**Strengths:**

This paper starts from a very interesting motivation: benchmark scores are not inherent properties of a model but the outcome of a specified evaluation system. The authors aim to disentangle a model’s intrinsic ability from other influencing factors, such as hyperparameter settings and prompting strategies. This is an important and realistic observation. In related work, we indeed see that different hyperparameters can dramatically change model performance, and that various prompting methods (e.g., Chain-of-Thought) can affect different models in a different way. A systematic evaluation framework would therefore allow for fairer comparisons between models. By introducing the MES/A-MES framework, the authors make a useful attempt to formalize an evaluation system with systemetically varying configurations. This effort contributes to the ongoing discussions about reproducibility and standardization in LLM evaluation.

**Weaknesses:**

1. I think the experimental design has major issues in how component value ranges are selected. For instance, the paper uses a temperature range of {0.0,1.0,2.0}, which is overly discrete and unrealistic. Most models are tuned for values between 0 and 1. Similarly, the smallest max_tokens value is set to 10, which would truncate nearly all responses. These settings would almost never be used in real deployment. Such unrealistic experimental configurations seem to have caused major issues in the results. If the parameter discretization and value ranges were reasonable, I would expect to observe a roughly unimodal distribution of model accuracies in Figure 4(a). However, all models show a violin shape with large mass near 0. I think these near-zero accuracies likely arise from extreme settings (e.g., temperature = 2.0 causing random outputs, or max_tokens = 10 cutting off answers). Such mixtures of valid and invalid runs inflate variance and render the reported evaluation unreliable.

2. The motivation and construction of A-MES are confusing. The authors state their motivation as “performance on a single workload can be misleading, as models may succeed on problems they have effectively memorized yet fail when the same reasoning must be applied under slightly altered conditions” (with an example shown in Line 171). However, I don’t think this example is appropriate. Adding an irrelevant distractor sentence is more like testing to robustness to adversarial perturbation rather than an analogical generation test for new use cases. This design choice also makes the subsequent definition of A-MES less convincing.
Also, the definition of out-of-distribution workloads in Line 250 does not align with the common understanding of OOD evaluation as well. It seems more related to removing petential data contamination during training rather than testing true distributional shift. Moreover, I believe it would make the A-MES results more credible to disclose more detailed documentation of the construction of the perturbed datasets.

3. This paper still has much room for improvement in presentation. I’ve listed several issues in the following Questions section. In addition, the term *LLM workload* appears to be borrowed from CPU benchmarking. Since the authors intentionally distinguish between task and workload, this concept should have been introduced earlier in the paper. As LLM workload is not a conventional term in the LLM community, a clearer definition would help readers better understand the authors’ intention.

**Questions:**

1. In figure 4(a), I couldn’t locate the reported accuracy line. I guess it should not appear in the label?

2. In Table 3 and Figure 5(a), it appears that models without reported AIME'24 scores are displayed as 0, which could be misleading to readers. Could the authors clarify how missing data is represented?

3. Regarding the "ground truth" in Figure 5(a), could the authors elaborate on why this serves as a meaningful baseline? By restrict the configuration space to the 5 most influential components, you retained the factors that dominate the variance of accuracy and fixed factors that contribute negligible variance, then the confidence interval computed on the unrestricted space will naturally cover this restricted mean. But I don’t think this indicates any robustness of the evaluation system, it merely reflects the reduced variance caused by restricting the configuration space. Also, considering the concerns mentioned earlier, I doubt if this can be called a ground truth value that reflects the intrinsic ability of an LLM.

4. I also find several parts of the paper unclear. For example, Table 1 and Table 2 are referred to as “evaluation settings” and “evaluation conditions,” and the row names in Table 2 appear to correspond to the column names in Table 1, while their listed values are very different. I carefully read corresponding sections but still could not determine under which configuration setting the experiments were actually conducted. I currently assume it corresponds to Table 2 as it was referred to in the evaluation section, but please correct me if this interpretation is mistaken.

5. If I understand correctly, your method essentially computes the mean accuracy over a predefined configuration space. However, in practical use, users typically operate under a fixed configuration. Moreover, we often observe that different models and tasks require different optimal hyperparameters, at least evident for temperature. I wonder whether a benchmark should evaluate models by averaging across all possible ones instead of under a representative configuration. Let’s assume one model achieves excellent performance within a narrow temperature range on a specific task, while another model performs slightly worse but maintains decent accuracy across a wider range. Which model should be considered better in that case? How can you justify that averaging performance over configurations is a fairer comparison than evaluating models under a representative or optimized fixed configuration?

---

> ### Author Response · Authors · 2025-11-25
> **Response to Reviewer 6AVH**
>
> Dear Reviewer 6AVH, thank you for your careful and thoughtful review. We are glad that you found our core motivation—the idea that benchmark scores depend on the evaluation system rather than being intrinsic to a model—both important and interesting, and that you see value in formalizing evaluation through the MES/A‑MES framework for more systematic, reproducible, and fair comparisons. We also appreciate your substantive critiques regarding experimental design, definitions of analogical and out-of-distribution workloads, and clarity of presentation. Your questions on interpreting the "ground truth" baseline and the choice between single-configuration versus averaged evaluation touch on the core of our framework and will help us refine both methodology and justification. Below, we respond point by point and will incorporate clarifications in the revised version.

---

> ### Author Response · Authors · 2025-11-25
> **Response to Weakness 1: Selection of component value ranges and realism of MES configurations (1/2)**
>
> Thank you very much for this careful and constructive comment. We fully agree that unrealistic evaluation settings can easily distort conclusions, so we take this concern seriously. Below we clarify our design choices and the corresponding validation we have performed.
>
> ## 1. Parameter Range Selection
>
> In our actual experiments, we first explored temperature with more fine-grained values between 0 and 1, but when constructing the final MES configuration space we deliberately coarsened this dimension to the three values reported in the paper. The reason is combinatorial: with 10 variables, each additional level per variable multiplies the total number of configurations, quickly making systematic exploration intractable. We therefore treated {0.0, 1.0, 2.0} as a compressed parametrization of a richer underlying search, and verified on the finer-grained runs that the qualitative trends we report are consistent with those obtained at higher resolution. The value ranges of the other parameters were similarly compressed for the same reason.
>
> ## 2. `max_tokens = 10` does not truncate answers in our setup
>
> Your concern notes that `max_tokens = 10` could truncate answers in typical conversational settings. However, in our non‑CoT evaluation setup, we ask the LLM to use a very compact answer format:
>
> - For multiple-choice questions:
>   `####A####`
> - For numeric fill-in-the-blank:
>   `####342####`
>
> Here, `####` is used as a special delimiter and corresponds to *one token* in our tokenizer; thus a complete answer like `####A####` typically consumes only a **small handful of tokens** (about 3 tokens).
>
> Under this constrained format:
> - `max_tokens = 10` is more than sufficient to generate complete answers for both multiple-choice and fill-in-the-blank questions.
> - In our logs, we do not observe systematic truncation of answers at `max_tokens = 10` for these tasks.
>
> ## 3. `temperature = 2.0` does not universally cause "random" outputs
>
> We agree that for some models, pushing temperature to 2.0 can lead to unstable behavior. However, this is model-dependent and not universally true. For example, consider the following configuration for DeepSeek:
>
> ```text
> {
>   'Language': 'yy',
>   'Question Format': 0,
>   'Question Paraphrase': 0,
>   'Shot': 1,
>   'COT': 0,
>   'Multi Turn': 0,
>   'temperature': 2.0,
>   'top_p': 0.6,
>   'presence_penalty': 0.5,
>   'max_tokens': 100
> }
> ```
>
> Under this configuration, the observed accuracy is 56.67%, which is clearly far above random guessing for AIME problems. This indicates that including `temperature = 2.0` in the evaluation space is not equivalent to injecting invalid runs. Rather, `temperature = 2.0` is challenging but not pathological for the model, and it provides useful information about robustness under aggressive decoding settings.
>
> ## 4. Realistic deployment scenarios for `max_tokens = 10` and `temperature = 2.0`
>
> `max_tokens = 10` does have practical use cases in real deployments, especially when users only need very short responses from the model. By constraining `max_tokens`, applications can cap output length to save time (avoiding long generations when only a brief signal is required) and reduce cost (since pricing is typically proportional to the number of output tokens). Consequently, although `max_tokens = 10` may appear "extreme" from the perspective of open-ended chat, it is a realistic and meaningful setting for short-answer tasks like ours, as well as for many production scenarios that prioritize brevity and efficiency.
>
> Similarly, `temperature = 2.0` is not purely an academic extreme. It is used in creative-generation scenarios such as poetry, fiction, and brainstorming, where diversity and novelty are prioritized. It is also employed for generating unusual phrasing or surprising ideas in exploratory ideation tools, where users explicitly trade reliability for creativity. In such applications, practitioners intentionally set a high temperature to push the model away from generic responses. Therefore, we view `temperature = 2.0` as a realistic configuration for specific use cases, even if it is not ideal for strict QA-type benchmarks. We include such settings to study how sensitive model performance is to decoding extremes, since real deployments often explore a wide range of temperatures across tasks.

---

> ### Author Response · Authors · 2025-11-25
> **Response to Weakness 1: Selection of component value ranges and realism of MES configurations (2/2)**
>
> ## 5. Why near-zero accuracies are not pathological on AIME'24
>
> AIME problems are extremely difficult, even for human experts. In our author group:
> - With strong mathematical backgrounds, we could solve only about one third of the problems without looking at the official solutions.
> - Even after carefully studying the solution strategies, we still found about 10% of the problems for which we could not fully understand the solution idea.
>
> Given this difficulty, the concentration of mass near zero in the violin plots (Figure 4(a)) primarily reflects the intrinsic hardness of AIME’24, rather than the use of extreme decoding settings.
>
>
> ## 6. Direct test: removing all `max_tokens = 10` and `temperature = 2.0` runs
>
> To address your concern more directly, we ran an explicit ablation in which we removed all evaluation points with `max_tokens = 10` and all points with `temperature = 2.0`, and then recomputed and replotted the violin plots corresponding to Figure 4(a). The new plots are qualitatively very similar to the originals: the density near zero accuracy is slightly reduced, but the overall shape remains, still with substantial mass close to 0. This indicates that the mass near zero is not primarily caused by these "extreme" settings. Instead, it arises from a combination of the intrinsic difficulty of AIME'24, non‑CoT configurations, less favorable prompts and languages, as well as other factors.
>
> Therefore, excluding `max_tokens = 10` and `temperature = 2.0` does not materially change our main conclusions or the qualitative distributions of accuracies.
>
> In summary, we acknowledge the importance of using realistic and interpretable configurations, and have therefore (i) compressed parameter ranges only after initially exploring finer-grained settings, (ii) carefully designed the answer format so that `max_tokens = 10` is sufficient and non-pathological, (iii) empirically verified that `temperature = 2.0` can yield non-random performance across models, (iv) motivated both `max_tokens = 10` and `temperature = 2.0` with real deployment scenarios, and (v) confirmed via ablation that removing these "extreme" settings leaves the qualitative accuracy distributions essentially unchanged. Together, these analyses support that our MES configuration space, while intentionally broad, does not artificially inflate variance or undermine the reliability of the reported evaluation results.

---

> ### Author Response · Authors · 2025-11-25
> **Response to Weakness 2: Motivation and construction of A‑MES and definition of out‑of‑distribution workloads (1/3)**
>
> ## 1. On "distractor insertion": robustness vs. analogical generalization
>
> You are absolutely right that our current example (adding an irrelevant sentence) reads like a robustness or adversarial‑style perturbation. Our intention, however, was not to claim that this alone constitutes analogical reasoning; rather, A‑MES is designed to cover a broader family of workload shifts, including robustness to realistic "noise" in user queries.
>
> In real‑world usage, users rarely submit clean, minimal prompts; they often add story background, opinions, meta‑commentary, or partially incorrect intuitions. Users still expect the model to solve the core task correctly despite such "perturbations". From this perspective, introducing redundant or misleading sentences is not artificial but faithfully reflects actual deployment conditions. Evaluating robustness to these perturbations is therefore an essential part of "causally faithful" evaluation.
>
> More importantly, our redundancy insertion is not a single ad‑hoc edit; it is implemented via three explicit, controllable categories, and all instances are generated systematically via LLM prompting:
>
> 1. Context‑irrelevant redundancy: sentences completely unrelated to the problem.
>    *Example:*
>    > *The weather today seems quite pleasant, and it might be a great day for a picnic.* Find the number of triples of nonnegative integers $(a,b,c)$ satisfying $a + b + c = 300$ and \[a^2b + a^2c + b^2a + b^2c + c^2a + c^2b = 6,000,000.\]
>
>    Here, the weather is entirely unrelated to the math content.
>
> 2. Context‑relevant, explanatory redundancy: additional sentences that explain concepts already present in the question (semantics and solution unchanged).
>    *Example:*
>    > There exist real numbers $x$ and $y$, both greater than 1, such that $\log_x\left(y^x\right)=\log_y\left(x^{4y}\right)=10$. *A logarithm is a way to express how many times a base must be multiplied by itself to get a certain number*. Find $xy$.
>
>    The added sentence explains the notion of a logarithm while leaving the underlying problem unchanged.
>
> 3. Context‑relevant, misleading redundancy: sentences that are thematically related but subtly promote an incorrect heuristic.
>    *Example:*
>    > Alice and Bob play the following game. A stack of $n$ tokens lies before them. The players take turns with Alice going first. On each turn, the player removes either $1$ token or $4$ tokens from the stack. *Many players adopt a greedy approach here: always take $4$ whenever possible to shorten the game and restrict the opponent's replies.* Whoever removes the last token wins. Find the number of positive integers $n$ less than or equal to $2024$ for which there exists a strategy for Bob that guarantees that Bob will win the game regardless of Alice's play.
>
>     The extra sentence about the "greedy approach" is logically related to the game but suggests a flawed strategy, intentionally nudging the solver toward an incorrect line of reasoning
>
> We also analyzed these categories separately and observed an interesting pattern: context‑irrelevant redundancy tends to cause the largest drop in model accuracy, whereas even deliberately misleading context‑relevant redundancy has a smaller impact on accuracy than context‑irrelevant noise.

---

> ### Author Response · Authors · 2025-11-25
> **Response to Weakness 2: Motivation and construction of A‑MES and definition of out‑of‑distribution workloads (2/3)**
>
> Distractor insertion is only one type of analogical transformation we use (with an emphasis on robustness). In addition, we include:
>
> - **Numeric substitution:** we systematically change the numerical constants in the problem while keeping the solution method and underlying structure intact.
>     *Example:*
>     - **Original:** Find the largest possible real part of \[(75+117i)z + \frac{96+144i}{z}\] where $z$ is a complex number with $|z|=4$. A common shortcut is to take $z$ to be a positive real number, since for a fixed modulus the real part is often largest when the argument of $z$ is zero.
>     - **Numeric variant:** Find the largest possible real part of \[(100+112i)z + \frac{60+144i}{z}\] where $z$ is a complex number with $|z|=4$. A common shortcut is to take $z$ to be a positive real number, since for a fixed modulus the real part is often largest when the argument of $z$ is zero.
>
> - **Conditional recomposition:** we change which quantity is treated as the "target" and which appears as a given condition, while preserving the same geometric or algebraic relationships.
>     *Example:*
>     - **Original:**
>     > Rectangles $ABCD$ and $EFGH$ are drawn such that $D,E,C,F$ are collinear. Also, $A,D,H,G$ all lie on a circle. If $BC=16$,$AB=107$,$FG=17$, and $EF=184$, what is the length of $CE$?
>     - **Conditional recomposition:**
>     > Rectangles $ABCD$ and $EFGH$ are drawn such that $D,E,C,F$ are collinear. Also, $A,D,H,G$ all lie on a circle. If $BC=16$,$AB=107$,$CE=104$, and $EF=184$, what is the length of $FG$?
>
> To summarize, in our framework "analogical transformations" are intended to capture a class of reasoning‑preserving modifications: they may change the numerical values or which variable is queried, and may introduce realistic redundancy, but they keep the underlying solution strategy essentially unchanged. This is exactly what we aim to probe—whether the model has internalized the reasoning pattern rather than merely memorized a specific benchmark instance. If the term analogical suggests a narrower notion to you and has contributed to this confusion, we would be happy to adopt a more precise term in the revised version.

---

> ### Author Response · Authors · 2025-11-25
> **Response to Weakness 2: Motivation and construction of A‑MES and definition of out‑of‑distribution workloads (3/3)**
>
> ## 2. On the use of "out‑of‑distribution" (OOD)
>
> We apologize for the confusion caused by our terminology. Our intention was not to redefine OOD in a way that conflicts with its standard usage in the ML literature. In our setting, we conceptually distinguish three categories of workloads:
>
> 1. Seen / potentially memorized: tasks that may have been present (or nearly present) in training.
> 2. Unseen but structurally similar: tasks not memorized verbatim, but solvable by applying previously learned patterns or transformations.
> 3. Completely novel: tasks that are unlikely to appear in the training corpus and thus probe more genuine generalization.
>
> In the paper, we loosely referred to category (3) as "out‑of‑distribution". Thanks to your comment, we now see that calling this "OOD" can be misleading, since OOD in the general literature often refers to a well‑characterized distributional shift (e.g., different domain, style, or covariates), rather than simply "unseen exam problems". We will adopt a more precise term for this category in the revised version.

---

> ### Author Response · Authors · 2025-11-25
> **Response to Weakness 3: Lack of a clear definition for the term "LLM workload"**
>
> Thank you for pointing out the potential confusion caused by our use of the term "workload". Our use of "workload" is indeed inspired by the computer architecture community.
>
> In our setting, a "workload" corresponds to a question within a benchmark that needs to be solved by the LLM, while a "task" has a closely related but slightly more general meaning, referring for example to a broader type of question or capability. However, we agree that the term "workload" is not yet standardized in the LLM community, and that both the term itself and its connection to CPU benchmarking are not introduced early enough in the paper, so the current presentation will be made clearer in the revision.

---

> ### Author Response · Authors · 2025-11-25
> **Response to Question 1: Redundant reported-accuracy line in figure 4(a)**
>
> Thank you for carefully checking Figure 4(a) and for pointing out the issue with the legend.
>
> You are absolutely right: in Figure 4(a) there should not be a separate legend entry for a "reported accuracy line". That item in the legend is a plotting mistake on our side and does not correspond to an actual curve that should be interpreted by the reader. In the revised version, we will remove the incorrect legend entry from Figure 4(a).

---

> ### Author Response · Authors · 2025-11-25
> **Response to Question 2: Zero values in Table 3 and Figure 5(a)**
>
> Thank you for pointing out the potential confusion regarding the 0 accuracies in Table 3 and Figure 5(a). We would like to clarify the following:
>
> The 0 accuracies are not placeholders for missing data. For all models listed in Table 3 and Figure 5(a), whenever a model shows 0 accuracy, it means that on our evaluated configuration, the model did not solve any of the AIME'24 problems correctly, not that we lacked data or simply treated "not reported" as zero.
>
> In fact, under some evaluation settings, 0% accuracy on AIME'24 is not only possible but even expected. AIME is an extremely challenging U.S. mathematics invitational competition: among the authors, even with strong mathematical backgrounds, we can correctly solve only about one third of the problems unaided, and even after carefully studying official solutions there remains roughly 10% of the problems whose step‑by‑step reasoning is still difficult to follow. Given this intrinsic difficulty, it is therefore not surprising that for some models and some configurations—especially those without chain‑of‑thought or suitable prompting—the measured accuracy is 0%, as our experiments indeed show that such configurations often lead to systematic failure on these items.
>
> To avoid misleading, in the revision we will explicitly state that a 0 accuracy value means that no questions are solved correctly under the evaluated configurations, rather than indicating missing data. We will also clarify that, given the intrinsic difficulty of AIME and the absence of chain‑of‑thought in some configurations, a 0% accuracy is in fact a realistic and interpretable outcome for certain models and settings.

---

> ### Author Response · Authors · 2025-11-25
> **Response to Question 3: Interpretation of the "ground truth" baseline in Figure 5(a)**
>
> Thank you for raising this – our earlier wording was not precise enough and indeed made it easy to misunderstand how the "ground truth" in Figure 5(a) is constructed. In fact, the configuration space defined by the 5 most influential components is not a subset of the random samples. Therefore, there is no mathematical guarantee that the confidence interval estimated on the sampled space will naturally cover the restricted mean.
>
> More concretely, for the ANOVA and the "ground truth" in Figure 5(a), we construct a factorial subspace over the 10 components (Language, Question Format, Question Paraphrase, Shot, COT, Multi Turn, temperature, top_p, presence_penalty, max_tokens). For each component, we partition its original value range into a "low" and a "high" level by selecting representative low‑ and high‑value settings, and then take the full factorial combination of these two levels across all 10 components. This yields a 2¹⁰ = 1024-point design that is balanced across all 10 factors.
>
> From this 1024‑point design we run ANOVA and then select the 5 most influential components (e.g., Question Format, COT, max tokens, Shot, Multi Turn for a given model). For these 5 components we then take all of their original value ranges (as listed in Table 2), and for the remaining 5 components we keep them at fixed values. The Cartesian product of "all values of the 5 important components × fixed values of the other 5" defines a new configuration space, on which we perform exhaustive evaluation and average the accuracy. This average is what we call the "restricted‑space ground truth".
>
> In contrast, the random sampling used to construct the confidence intervals in Figure 5(a) is performed over the full 10‑dimensional configuration space defined in Table 2, whose size is 15,552.
>
> We sample configurations uniformly from this full space, evaluate them, and continue sampling until both the sample mean and its confidence interval have converged within the predefined thresholds.
>
> Importantly, many of the configurations in the restricted space do not appear in the randomly sampled configurations, and vice versa. The restricted space is therefore not a subset of the randomly sampled configuration set.
>
> Consequently, the fact that the confidence intervals derived from random sampling over the full 15,552‑point space consistently cover the mean obtained from exhaustive testing on the separate restricted space is not guaranteed by construction. It is an empirical check of consistency between two different approximations of the "global" performance.

---

> ### Author Response · Authors · 2025-11-25
> **Response to Question 4: The actual experimental configuration**
>
> Thank you for raising this point — it helped us realize that the distinction between Table 1 and Table 2 is not yet sufficiently explicit in the current version.
>
> In Section 3 we wrote:
>
> > "To make this issue concrete, we systematically reviewed major benchmarks and compiled a taxonomy of which components are explicitly defined and which are left open (Table 1)."
>
> This means that Table 1 is only used to summarize and compare how existing benchmarks and model technical reports typically set (or leave unspecified) the 10 key components. It serves to illustrate the current evaluation practice and its lack of control over many indispensable variables, thereby motivating our methodology. Concretely:
>
> - The rows of Table 1 are different benchmarks (e.g., MMLU, AIME, GPQA, etc.);
> - The columns are the 10 key components (Language, Question Format, Shot, COT, temperature, etc.);
> - The values come from official benchmark documentation and model technical reports and are used to show:
>     * which components are explicitly fixed,
>    * which are left open/underspecified or have multiple possible choices,
>     * and how different models are often evaluated on the same benchmark under different configurations.
>
> Thus, Table 1 is a "status-quo taxonomy" of how others evaluate models; it is not the configuration we used in our experiments.
>
> By contrast, Table 2 appears in Section 4 when we introduce our LLM evaluatology methodology and define the Minimal Evaluation System (MES). Table 2 specifies the evaluation conditions (EC) that we actively control and on which our experiments are actually run.
>
> All of our MES and A-MES experiments (including the results in Figure 4, Table 3, and the ANOVA analysis) are conducted under the configuration space specified in Table 2, not Table 1.
>
> In the revised version, we will clarify more explicitly under which configurations our experiments are conducted.

---

> ### Author Response · Authors · 2025-11-25
> **Response to Question 5: Appropriate metric for comparing models**
>
> Thank you for this thoughtful question; it touches exactly the core design choice behind our methodology. Our position is not that every benchmark must always average over a large configuration space, but that:
>
> 1. Different stakeholders have very different, sometimes conflicting, configuration needs, and
> 2. If we only report performance under a single (even "optimized") configuration, we systematically bias the benchmark toward some users and against others, and we make causal attribution to the model itself unreliable.
>
> ## 1. Why a single "representative" configuration is problematic
>
> In practice, users do not share a single fixed configuration, even for something as simple as temperature:
>
> - Creative writing / brainstorming / ideation.
>   Users typically prefer *higher* temperature (e.g., 0.8–1.0 or even above) to obtain diverse, exploratory outputs.
>
> - Safety‑critical or correctness‑critical use cases (e.g., medical triage support, compliance checks, financial calculations, math competitions).
>   Users usually choose *low* temperature (e.g., 0.0–0.2) and often strict decoding to minimize randomness and hallucinations.
>
> LLMs are deployed to a broad, heterogeneous user base across many domains, not to a homogeneous user who always uses exactly one temperature. If a benchmark fixes the temperature at a single "representative" choice, it implicitly represents some subset of applications at the expense of others. Any single fixed configuration is, unavoidably, a choice of whose use case "counts" as the benchmark.
>
> ## 2. Why averaging over a configuration space is useful and what it means
>
> Our method does not claim that "the true user setting is the uniform distribution over all configurations". Instead, we do two things:
>
> 1. We explicitly define an evaluation condition space EC (Table 2) over indispensable components (Language, Question Format, Shot, COT, temperature, top‑p, etc.), and
> 2. We estimate the expected accuracy under that space by controlled sampling (with convergence checks on the mean and confidence intervals).
>
> This has several motivations:
>
> - Fairness across users.
>     By averaging over multiple plausible configurations, we approximate how a model behaves across a spectrum of realistic use patterns, rather than privileging a single usage style.
>
> - Causal attributions to the model itself.
>     When different models are evaluated under different, partially specified configurations, observed performance differences can conflate intrinsic model quality with arbitrary choices of prompts and decoding hyperparameters. By defining a shared EC space and averaging over it in a controlled manner, we reduce this confounding and obtain measurements that better reflect the effect of the model itself. In addition, by combining this with ANOVA over the EC space, we can quantitatively decompose variance into contributions from the model and from specific components (e.g., Question Format, COT, max tokens), making the causal structure of the evaluation more transparent.
>
> Conceptually, our average is a model of an "evaluation user population": instead of assuming a single fixed user behavior, we define an explicit configuration distribution and report the expected performance under that distribution.
>
> ## 3. Your example: narrow strong band vs. broad moderate performance
>
> You raise a key trade-off:
> > Suppose Model A is excellent within a narrow temperature range on a task, while Model B is slightly worse but maintains decent accuracy across a wider range. Which model is "better"?
>
> If stakeholders care about robustness across settings, then they should prefer B when the mean over a broad configuration space is higher.
>
> If they care about peak performance under a tightly controlled configuration, they might prefer A — provided they are willing and able to enforce that precise configuration in deployment.
>
> In other words, there is no universally correct answer to "which is better?"; the evaluation outcome depends on individual preferences, and our methodology is compatible with this fact:
>
> - In our experiments, we use relatively broad ranges for each component in EC. However, in the MES framework and the accompanying tooling we propose, the value ranges of all components in EC are user‑configurable rather than fixed.
> - If a user cares about a **narrow** operating regime (e.g., a tight temperature band, fixed language, fixed format), they can set correspondingly narrow ranges and evaluate models under that restricted EC.
> - If a user instead cares about **broader** behavior across diverse configurations, they can define wider ranges for the same components.
>
> In both cases, the same evaluatology pipeline is applied within the user‑specified EC space, yielding results tailored to their particular preferences.

---

> > ### Comment · Reviewer_6AVH · 2025-11-28
> >
> > I thank the authors for providing the additional explanations, which effectively addressed most of my concerns, and I would be very happy to increase my score. Unfortunately, due to recent technical issues on OpenReview and their consequences, the system no longer allows me to modify my score. I sincerely regret this situation, but I would be glad to update my score accordingly if an opportunity becomes available later.

---

> > > ### Author Response · Authors · 2025-11-28
> > >
> > > Thank you very much for your thoughtful follow-up and for letting us know. We truly appreciate that our rebuttal addressed your concerns and that you would be willing to increase your score. We completely understand the current technical limitations on OpenReview, and we are already very grateful for the time, effort, and care you have invested in reviewing our work.

---

### Official Review · Reviewer_5j6t · 2025-10-31

**Soundness:** 3
**Presentation:** 2
**Contribution:** 3
**Rating:** 4
**Confidence:** 2

**Summary:**

The paper shows that LLM performance depends strongly on evaluation settings like prompt format and decoding, meaning current benchmarks often measure the evaluation setup rather than the model. It proposes a structured evaluation framework (MES / A-MES) to control and vary these components, revealing that model rankings can change significantly under different conditions. The goal is to make LLM evaluations more reliable and attributionally faithful.

**Strengths:**

- clear and compelling problem framing: the paper condignly argues that most current LLM evaluations conflate model ability with configuration choices
- conceptual contribution is novel and original: MES/A-MES provide a structured way to explicitly specify and vary evaluation conditions
- empirical evidence of variance strengthens the need for an evaluation framework
- ANOVA to quantify the contribution of prompt format, COT, decoding parameters is useful

**Weaknesses:**

- method is conceptual rather than algorithmic
- unclear if this generalizes beyond academic reasoning benchmarks
- complexity and practical burden on practitioners might be too high; unclear scalability
- limited human evaluations --> maybe human ablation could be added
- causal language might be overstated

**Questions:**

- Which parts of A-MES generation are automated and which require expert authoring? How would scalability be possible?
- DO you expect ANOVA factor importance to generalize across models and tasks, or is it benchmark-specific?
- How should practitioners choose a single evaluations core after MES/A-MES exploration? What is the summary statistic?

---

> ### Author Response · Authors · 2025-11-25
> **Response to Reviewer 5j6t**
>
> Dear Reviewer 5j6t, thank you for carefully reviewing our submission and providing thoughtful feedback. We appreciate your recognition of our problem framing, the novelty of the MES/A‑MES framework, and our ANOVA-based factor attribution.We also value your constructive critiques regarding the conceptual nature of our method, generalization beyond academic benchmarks, practical complexity, limited human evaluation, and the use of causal language. We address each point below and will incorporate corresponding clarifications in the revised version.

---

> ### Author Response · Authors · 2025-11-25
> **Response to Weakness 1: "method is conceptual rather than algorithmic" (1/8)**
>
> Due to space limitations, we did not fully explain some aspects of LLM evaluatology in the paper, especially the details of A‑MES. This omission may have made our approach appear more conceptual than algorithmic.
>
> In our implementation, A‑MES is instantiated as five systematically defined, script‑driven transformation pipelines including: three analogical (distractor insertion, numeric substitution, conditional recomposition) and two novel (recent‑source adaptation and conceptual synthesis). For each pipeline, we fix general prompts and scaffolding, and then run the entire process automatically through LLM API calls (e.g., GPT‑5), lightweight verification scripts and other auxiliary tooling. This setup scales to large workloads and produces diverse variants, without any per‑item manual rewriting or hand‑crafting of individual problems. Moreover, we are actively exploring additional automated AIME‑style transformation pipelines to further enrich A‑MES.
>
> Overall, these five transformation pipelines realize seven concrete augmentation mechanisms for each workload (here, a workload means a question within the benchmark):
> 1. Distractor Insertion
>    - Context‑irrelevant distractor insertion
>    - Context‑relevant explanatory distractor insertion
>    - Context‑relevant misleading distractor insertion
> 2. Numeric substitutions
> 3. Conditional recomposition
> 4. Recent‑source adaptation
> 5. Conceptual synthesis
>
> These mechanisms show that A‑MES is not a collection of ad‑hoc, hand‑edited examples, but a unified, automatable framework that systematically augments existing benchmarks to more comprehensively evaluate model capabilities. For each question within a benchmark, we first apply all seven mechanisms to construct the full space of augmented variants, filtering out transformation attempts that fail (e.g., numeric substitutions or conditional recomposition without a stable solver). Evaluation then samples directly from this augmentation space, ensuring that no invalid transformations are ever selected. The framework enumerates the entire augmentation space upfront, and the small number of discarded variants has negligible impact on overall coverage or robustness. For several major benchmarks, we have already constructed the corresponding augmentation spaces and plan to release them publicly in future work. Below, we provide concrete descriptions of the seven mechanisms.

---

> > ### Author Response · Authors · 2025-11-25
> > **Response to Weakness 1: "method is conceptual rather than algorithmic" (4/8)**
> >
> > ## 2. Analogical‑2: numeric substitutions via code‑based solution extraction
> >
> > For numeric substitutions, we use a uniform pipeline built around LLM‑generated Python solvers and automatic verification scripts, rather than manually changing a few numbers:
> >
> > 1. We first call an LLM to extract the primary knowledge points tested by the original problem, and query a pre‑constructed formula library indexed by knowledge point to retrieve potentially relevant formulas.
> > 2. We feed the original problem, its official answer, the retrieved formulas, and (where available) multiple correct solution sketches into the LLM. The official answer and solution sketches are provided by the user, and providing solution sketches is optional.
> > 3. The LLM is prompted to:
> >    - analyze the problem's solution strategy, using the provided solution sketches when available
> >    - write a Python solution program where problem‑specific numbers are extracted as explicit variables with reasonable value ranges.
> > 4. We then ask another LLM to inspect the generated Python code and verify that it implements a general computational procedure for solving the problem, rather than relying on hard‑coded instance‑specific outputs or trivial pattern matching.
> > 5. We import the LLM‑generated Python code as a local module and call its solve() function with the original numeric values as inputs, checking whether the resulting output matches the official answer.
> > 6. If the code fails (wrong answer or runtime error), we return the error message and incorrect output to the LLM, asking it to refine the code; we repeat this refinement–verification loop for up to five attempts and keep the Python code if it passes on the original instance.
> >
> > After obtaining a correct solver, we automatically sample new numeric configurations within the validated ranges to generate analogical variants of the same underlying problem. For example:
> >
> > - **Original:** Find the largest possible real part of \[(75+117i)z + \frac{96+144i}{z}\] where $z$ is a complex number with $|z|=4$. A common shortcut is to take $z$ to be a positive real number, since for a fixed modulus the real part is often largest when the argument of $z$ is zero.
> > - **Numeric variant:** Find the largest possible real part of \[(100+112i)z + \frac{60+144i}{z}\] where $z$ is a complex number with $|z|=4$. A common shortcut is to take $z$ to be a positive real number, since for a fixed modulus the real part is often largest when the argument of $z$ is zero.
> >
> > This "knowledge‑point extraction → formula retrieval → analyze → code → verify → resample" pipeline is identical across all problems.
> >
> > Algorithm 4: Numeric Substitutions
> >
> > ```pseudo
> > FUNCTION BuildNumericSolver(problem_text, answer_gold, solution_sketches, max_iter=5, max_refine=5):
> >   knowledge_points ← LLM_ExtractKnowledgePoints(problem_text)
> >   retrieved_formulas ← RETRIEVE_FORMULAS(knowledge_points)
> >   history ← EMPTY_LIST
> >   FOR iter IN 1..max_iter:
> >     PROMPT ← BuildPromptCodeGen(problem_text, answer_gold, solution_sketches, retrieved_formulas)
> >     history.APPEND((PROMPT, None))
> >     (CODE, param, value_ranges)   ← LLM_CALL(PROMPT)
> >     is_hard_code ← LLM_HARD_CODE_CHECK(CODE)
> >     IF is_hard_code:
> >       CONTINUE
> >     FOR refine_step IN 0..max_refine:
> >       (output, error) ← RUN_PYTHON(
> >                             CODE,
> >                             input = param.value
> >                           )
> >       history.APPEND((CODE, (output, error)))
> >       IF error == NONE AND VERIFY(output, answer_gold):
> >         RETURN (CODE, param, value_ranges)
> >       IF refine_step == max_refine:
> >         BREAK
> >       PROMPT_refine ← BuildPromptCodeRefine(
> >                               problem_text = problem_text,
> >                               answer_gold  = answer_gold,
> >                               history      = history
> >                         )
> >       (CODE, param, value_ranges) ← LLM_CALL(PROMPT_refine)
> >   RETURN FAILURE
> >
> > FUNCTION GenerateNumericVariants(problem_text, answer_gold, solution_sketches):
> >   (solver_code, param, param_ranges) ← BuildNumericSolver(problem_text, answer_gold, solution_sketches)
> >   IF solver_code == FAILURE:
> >     RETURN NONE
> >   new_param ← SAMPLE(param_ranges)
> >   new_problem_text ← InstantiateNumericProblemText(problem_text, param, new_param)
> >   (output, error) ← RUN_PYTHON(solver_code, input = new_params)
> >   new_answer_gold ← output
> >   RETURN (new_problem_text, new_answer_gold)
> > ```

---

> ### Author Response · Authors · 2025-11-25
> **Response to Weakness 1: "method is conceptual rather than algorithmic" (2/8)**
>
> ## 1. Analogical‑1: distractor insertion with three well-defined categories
>
> For distractor insertion, we define three explicit, controllable categories of redundancy and implement all instances via LLM prompting. To ensure that the inserted distractors strictly follow our predefined specifications, we empirically test several candidate LLMs and choose the one that most consistently adheres to these constraints (GPT-5). This selection is made solely to guarantee transformation fidelity rather than to compare model capabilities. For each item to be transformed, the chosen LLM is invoked through an API and, guided by our structured prompts, automatically produces and inserts the required redundant content. The prompts are provided below.
>
> - **Context‑irrelevant redundancy**
>   - Provide the LLM with an example containing an original question and a version with added context‑irrelevant redundancy.
>   - Instruct the LLM to insert one sentence at a random position that is completely unrelated to the target question.
>
>   Algorithm 1: Context‑Irrelevant Redundancy Insertion
>
>   ```pseudo
>   FUNCTION InsertContextIrrelevantDistractor(problem_text):
>     EXAMPLE_PAIR ← (orig_example, example_with_irrelevant_context)
>     PROMPT ← BuildPromptIrrelevant(EXAMPLE_PAIR, problem_text)
>     RESPONSE ← LLM_CALL(PROMPT)
>     transformed_text ← ParseTransformedProblem(RESPONSE)
>     IF NOT BasicSanityCheck(problem_text, transformed_text):
>       RETURN FAILURE
>     RETURN transformed_text
>   ```
>
> - **Context‑relevant, explanatory redundancy**
>   - Provide the LLM with an example of an original question and a version with added explanatory redundancy.
>   - Instruct the LLM to insert a redundant sentence at a random position in each target question that explains a concept already appearing in the target question.
>
>   Algorithm 2: Context‑relevant Explanatory Redundancy Insertion
>
>   ```pseudo
>   FUNCTION InsertExplanatoryDistractor(problem_text):
>     EXAMPLE_PAIR ← (orig_example, example_with_explanatory_sentence)
>     PROMPT ← BuildPromptExplanatory(EXAMPLE_PAIR, problem_text)
>     RESPONSE ← LLM_CALL(PROMPT)
>     transformed_text ← ParseTransformedProblem(RESPONSE)
>     IF NOT BasicSanityCheck(problem_text, transformed_text):
>       RETURN FAILURE
>     RETURN transformed_text
>   ```
>
> - **Context‑relevant, misleading redundancy**
>   - Provide the LLM with an example containing an original question and a version with added misleading but logically related redundancy.
>   - Supply the model with the correct answer and several correct solution approaches, and instruct it to avoid directly hinting at these correct strategies when crafting the misleading cue. The official answer and solution approaches are provided by the user, and providing solution approaches is optional.
>   - Instruct the model to insert a redundant sentence that nudges the reader toward an incorrect strategy or line of reasoning, without explicitly revealing that it is "misleading" or "distracting".
>
>   Algorithm 3: Context‑relevant Misleading Redundancy Insertion
>
>   ```pseudo
>   FUNCTION InsertMisleadingDistractor(problem_text, answer_gold, solution_sketches):
>     EXAMPLE_PAIR ← (orig_example, example_with_misleading_sentence)
>     PROMPT ← BuildPromptMisleading(
>                 example_pair      = EXAMPLE_PAIR,
>                 target_problem    = problem_text,
>                 answer_gold       = answer_gold,
>                 solution_sketches = solution_sketches
>                )
>     RESPONSE ← LLM_CALL(PROMPT)
>     transformed_text ← ParseTransformedProblem(RESPONSE)
>     IF NOT BasicSanityCheck(problem_text, transformed_text):
>       RETURN FAILURE
>     RETURN transformed_text
>   ```

---

> ### Author Response · Authors · 2025-11-25
> **Response to Weakness 1: "method is conceptual rather than algorithmic" (3/8)**
>
> In practice, the selected LLM produces variations that are more diverse and linguistically natural than manual editing. In particular, its context‑relevant misleading redundancies tend to hint at incorrect heuristics in a more subtle way than hand‑written versions, while still strictly adhering to the predefined category constraints. The entire process involves no per‑item manual editing. The three examples of redundancy for three types generated by the above procedure are illustrated as follows:
>
> 1. Context‑irrelevant redundancy example:
>    > *The weather today seems quite pleasant, and it might be a great day for a picnic.* Find the number of triples of nonnegative integers $(a,b,c)$ satisfying $a + b + c = 300$ and \[a^2b + a^2c + b^2a + b^2c + c^2a + c^2b = 6,000,000.\]
>
>    Here, weather is entirely unrelated to the math content.
>
> 2. Context‑relevant, explanatory redundancy example:
>    > There exist real numbers $x$ and $y$, both greater than 1, such that $\log_x\left(y^x\right)=\log_y\left(x^{4y}\right)=10$. *A logarithm is a way to express how many times a base must be multiplied by itself to get a certain number*. Find $xy$.
>
>    The added sentence explains the notion of a logarithm while leaving the underlying problem unchanged.
>
> 3. Context‑relevant, misleading redundancy example:
>    > Alice and Bob play the following game. A stack of $n$ tokens lies before them. The players take turns with Alice going first. On each turn, the player removes either $1$ token or $4$ tokens from the stack. *Many players adopt a greedy approach here: always take $4$ whenever possible to shorten the game and restrict the opponent's replies.* Whoever removes the last token wins. Find the number of positive integers $n$ less than or equal to $2024$ for which there exists a strategy for Bob that guarantees that Bob will win the game regardless of Alice's play.
>
>     The extra sentence about the "greedy approach" is logically related to the game but suggests a flawed strategy, intentionally nudging the solver toward an incorrect line of reasoning

---

> ### Author Response · Authors · 2025-11-25
> **Response to Weakness 1: "method is conceptual rather than algorithmic" (5/8)**
>
> ## 3. Analogical‑3: conditional recomposition via invertible‑condition analysis
>
> For conditional recompositions, we again adopt a general and automatable pipeline built around LLM‑generated Python solvers and automatic verification scripts rather than manually rewriting statements:
>
> 1. We first call an LLM to extract the primary knowledge points tested by the original question, and query a pre‑constructed formula library indexed by knowledge point to retrieve potentially relevant formulas.
> 2. We feed the original problem, its official answer, the retrieved formulas, and (where available) multiple correct solution sketches into the LLM. The official answer and solution sketches are provided by the user, and providing solution sketches is optional.
> 3. The LLM is prompted to:
>    - identify the key conditions and the target quantity;
>    - determine whether some of these conditions and the target can be interchanged—i.e., whether knowing the original answer allows us to infer some of the original conditions (an invertible relationship).
> 4. When such an invertible relationship exists, the LLM is asked to write a Python solution program for the recomposed problem, where the original target now appears as an input condition and (a subset of) the original conditions become the new target.
> 5. We then ask another LLM to inspect the generated Python code and verify that it implements a general computational procedure for solving the problem, rather than relying on hard‑coded instance‑specific outputs or trivial pattern matching.
> 6. We import the LLM‑generated Python code as a local module and call its solve() function, plugging the original answer value into the new "condition" slot and checking whether the returned output correctly recovers the original condition values.
> 7. Any discrepancy or runtime error is fed back to the LLM for iterative refinement, just as in the numeric substitutions pipeline. we repeat this refinement–verification loop for up to five attempts and keep the Python code if it passes on the instance.
>
> Once a correct solver for the recomposed version is obtained, we can further vary the new input variables within reasonable ranges to generate additional condition‑recomposed variants.
>
> - **Original:**
> > Rectangles $ABCD$ and $EFGH$ are drawn such that $D,E,C,F$ are collinear. Also, $A,D,H,G$ all lie on a circle. If $BC=16$,$AB=107$,$FG=17$, and $EF=184$, what is the length of $CE$?
> - **Conditional recomposition:**
> > Rectangles $ABCD$ and $EFGH$ are drawn such that $D,E,C,F$ are collinear. Also, $A,D,H,G$ all lie on a circle. If $BC=16$,$AB=107$,$CE=104$, and $EF=184$, what is the length of $FG$?
>
> These conditional recompositions are therefore produced by a uniform "knowledge‑point extraction → formula retrieval → analyze → code → verify → resample" pipeline, not by hand‑crafting each rephrased problem.

---

> ### Author Response · Authors · 2025-11-25
> **Response to Weakness 1: "method is conceptual rather than algorithmic" (6/8)**
>
> Algorithm 5: Conditional Recomposition
>
> ```pseudo
> FUNCTION BuildRecomposedSolver(problem_text, answer_gold, solution_sketches, max_iter=5, max_refine=5):
>   knowledge_points ← LLM_ExtractKnowledgePoints(problem_text)
>   retrieved_formulas ← RETRIEVE_FORMULAS(knowledge_points)
>   history ← EMPTY_LIST
>   ANALYSIS_PROMPT ← BuildPromptInvertibleAnalysis(
>                            problem_text,
>                            answer_gold,
>                            solution_sketches,
>                            retrieved_formulas
>                       )
>   ANALYSIS_RESPONSE ← LLM_CALL(ANALYSIS_PROMPT)
>   (invertible, cond_as_unknown, target_as_given, recomposed_problem_text) ← ParseInvertibleStructure(ANALYSIS_RESPONSE)
>   IF NOT invertible:
>     RETURN FAILURE
>   FOR iter IN 1..max_iter:
>     PROMPT ← BuildPromptRecomposedCodeGen(
>                       origin_problem_text = problem_text
>                       new_problem_text = recomposed_problem_text,
>                       origin_answer_gold = answer_gold,
>                       new_answer_gold = cond_as_unknown.original_values,
>                       solution_sketches = solution_sketches,
>                       retrieved_formulas = retrieved_formulas
>                 )
>     history.APPEND((PROMPT, None))
>     (CODE, value_ranges)   ← LLM_CALL(PROMPT)
>     is_hard_code ← LLM_HARD_CODE_CHECK(CODE)
>     IF is_hard_code:
>       CONTINUE
>     FOR refine_step IN 0..max_refine:
>       (output, error) ← RUN_PYTHON(
>                             CODE,
>                             input = answer_gold
>                           )
>       history.APPEND((CODE, (output, error)))
>       IF error == NONE AND VERIFY(output, cond_as_unknown.original_values):
>         RETURN (CODE, cond_as_unknown, target_as_given, value_ranges, recomposed_problem_text)
>       IF refine_step == max_refine:
>         BREAK
>       PROMPT ← BuildPromptRecomposedCodeRefine(
>                            problem_text = recomposed_problem_text,
>                            answer_gold  = cond_as_unknown.original_values,
>                            history      = history
>                         )
>       (CODE, value_ranges) ← LLM_CALL(PROMPT)
>   RETURN FAILURE
>
> FUNCTION GenerateRecomposedVariants(problem_text, answer_gold, solution_sketches):
>   (solver_code, cond_as_unknown, target_as_given, value_ranges, recomposed_problem_text) ← BuildRecomposedSolver(problem_text, answer_gold, solution_sketches)
>   IF solver_code == FAILURE:
>     RETURN NONE
>   new_param ← SAMPLE(value_ranges)
>   new_problem_text ← InstantiateRecomposedProblemText(recomposed_problem_text, target_as_given, new_param)
>   (output, error) ← RUN_PYTHON(solver_code, input = new_param)
>   new_answer_gold ← output
>   RETURN (new_problem_text, new_answer_gold)
> ```

---

> ### Author Response · Authors · 2025-11-25
> **Response to Weakness 1: "method is conceptual rather than algorithmic" (7/8)**
>
> ## 4. Novel‑1: recent‑source adaptation via structured retrieval and paraphrasing
>
> In the "novel" branch, the first mechanism is recent‑source adaptation, which is also fully scriptable:
>
> 1. We first use an LLM to extract the primary knowledge points tested by a given source problem.
> 2. We query open‑access repositories of centralized exam questions that index items by region, year, subject, and knowledge point, and crawl the most recent 2025 exam problems matching the extracted knowledge points.
> 3. The retrieved problems are paraphrased by the LLM and can be further transformed using the three analogical methods (redundancy insertion, numeric substitution, and conditional recomposition).
>
> This yields a set of new, recent‑source problems that are structurally aligned at the knowledge level but clearly distinct in surface form and provenance. The entire workflow is driven by scripts and general prompts, without hand‑curating individual items.
>
> Algorithm 6: Recent‑Source Adaptation
>
> ```pseudo
> FUNCTION RecentSourceAdaptation(problem_text, metadata, K):
>   KP_PROMPT   ← BuildPromptExtractKnowledgePoints(problem_text, metadata)
>   KP_RESPONSE ← LLM_CALL(KP_PROMPT)
>   KPs         ← ParseKnowledgePoints(KP_RESPONSE)
>   IF KPs == NONE:
>     RETURN NONE
>   year_range ← {2025}
>   candidate_item ← RETRIEVE_EXAMS(
>                            knowledge_points = KPs,
>                            year_range       = year_range,
>                            subject          = metadata.subject
>                        )
>   IF candidate_item == NONE:
>     RETURN NONE
>   PARA_PROMPT ← BuildPromptParaphrase(
>                            source_text   = candidate_item.text,
>                            answer_gold   = candidate_item.answer_gold,
>                            preserve_KPs  = KPs
>                          )
>   PARA_RESPONSE    ← LLM_CALL(PARA_PROMPT)
>   paraphrased_item ← ParseParaphrasedProblem(PARA_RESPONSE)
>   adapted_item = {
>             text : paraphrased_item.text,
>             answer : candidate_item.answer_gold,
>             KPs : KPs,
>             provenance : {
>                 source_exam : candidate_item.metadata,
>                 transform   : "paraphrase"
>             }
>           }
>   RETURN adapted_item
> ```

---

> ### Author Response · Authors · 2025-11-25
> **Response to Weakness 1: "method is conceptual rather than algorithmic" (8/8)**
>
> ## 5. Novel‑2: textbook‑based conceptual synthesis via a parsed knowledge base
>
> The second "novel" mechanism is conceptual synthesis from authoritative textbooks. We first crawl a large collection of authoritative textbooks across different subjects from the web, and then use the LLM API's built‑in functionality for parsing local PDF files to extract their content. Based on the extracted content, we build a structured knowledge base in which each concept is associated with definitions, properties, theorems, phenomena, and canonical examples extracted from the textbooks.
>
> 1. Given a problem to be augmented, we use an LLM to identify its main knowledge points, and then retrieve the corresponding entries from the structured knowledge base. If the subject‑specific knowledge base is missing, we trigger the textbook crawling and parsing step to expand the knowledge base, and then retrieve the corresponding entries from it.
> 2. Conditioned on these entries, the LLM is prompted to generate new conceptual questions targeting the underlying knowledge points, rather than copying any existing problem.
>
> For example, one generated question associated with the concept of *logarithms* is:
> > What kind of mathematical idea/method turns exponentiation and multiplication into multiplication and addition?
>
> This pipeline turns textbook content into fresh conceptual questions that align with the original topic but are novel in form and focus.
>
> Algorithm 7: Conceptual Synthesis
>
> ```pseudo
> FUNCTION ConceptualSynthesis(problem_text, metadata):
>   KP_PROMPT   ← BuildPromptExtractKnowledgePoints(problem_text, metadata)
>   KP_RESPONSE ← LLM_CALL(KP_PROMPT)
>   KPs         ← ParseKnowledgePoints(KP_RESPONSE)
>   IF KPs == NONE:
>     RETURN NONE
>   kb_entry ← RETRIEVE_KB_ENTRY(
>                  knowkedge_points = KPs
>                  subject          = metadata.subject,
>                )
>   IF kb_entry == NONE:
>     CRAWL_AND_PARSE_TEXTBOOKS(knowkedge_points = KPs, subject = metadata.subject)
>     kb_entry ← RETRIEVE_KB_ENTRY(
>                    knowkedge_points = KPs
>                    subject          = metadata.subject,
>                  )
>     IF kb_entry == NONE:
>       RETURN NONE
>   GEN_PROMPT   ← BuildPromptConceptualQuestionGeneration(
>                       kb_entry = kb_entry,
>                       KPs        = KPs,
>                  )
>   GEN_RESPONSE ← LLM_CALL(GEN_PROMPT)
>   raw_item     ← ParseGeneratedConceptualQuestions(GEN_RESPONSE)
>   conceptual_item = {
>             text       : raw_item.text,
>             answer     : raw_item.answer,
>             KPs        : KPs
>           }
>   RETURN conceptual_item
> ```
>
> ## Summary
>
> These mechanisms show that our method is not merely a high‑level conceptual proposal. A‑MES is realized as a concrete, fully automated workflow with fixed prompts, scripted control logic, programmatic verification, and data‑driven retrieval and synthesis.
>
> In the revised version, we will add a concise algorithmic overview summarizing this implementation, so that the algorithmic nature and reproducibility of LLM evaluatology are made explicit.

---

> ### Author Response · Authors · 2025-11-25
> **Response to Weakness 2: "unclear if this generalizes beyond academic reasoning benchmarks"**
>
> Our methodology is not limited to public academic benchmarks and naturally extends to non‑benchmark settings. As long as there exists a seed workload—such as an enterprise’s internal question bank, a school or institutional exam repository, or other custom task collections—our MES/A‑MES pipeline can be instantiated on top of it without any conceptual change. In this view, the internal or proprietary workload simply serves as the workload component in our framework; our tooling then automatically constructs the corresponding configuration space, performs workload augmentation, and runs the same sampling and attribution procedures. This enables organizations to obtain causally interpretable, system‑level evaluations tailored to their own private or domain‑specific tasks, rather than being restricted to public academic reasoning benchmarks.
>
> Moreover, existing reasoning benchmarks are typically defined as a fixed workload (a set of questions) plus scoring rules, with many indispensable components left underspecified, such as prompting, decoding, and system configuration.  In contrast, our LLM evaluatology explicitly treats a benchmark not as "a dataset with a number", but as a full evaluation system: we jointly consider (i) the evaluation object (an LLM or an LLM-based service), and (ii) a bundle of evaluation conditions, including the workload, the prompting method, and the decoding parameters. From this perspective, an academic reasoning benchmark like AIME, under a single default configuration, is just one point in a much larger configuration space (EC). Our contribution is to make this space explicit and controlled, systematically augment it in a semantically meaningful way, and attribute performance differences to specific components, rather than reporting a single score. Thus, our proposal is not "one more reasoning benchmark" but a general recipe for turning LLM evaluation into a causally interpretable system-level benchmark.
>
> Furthermore, our method is in fact fully generalizable. Specifically, we would like to clarify that our LLM evaluatology framework is task‑agnostic: once we define the set of evaluation variables (C1–C10) and their value ranges, the MES configuration space and the corresponding A‑MES workflows are automatically generated by our tooling. The procedures for random sampling, convergence‑based stopping, and ANOVA remain unchanged regardless of whether the benchmark is reasoning(AIME), knowledge (MMLU), science (GPQA), coding, or multi‑turn, and in our current experiments we apply this pipeline to AIME, MMLU, and GPQA with no benchmark‑specific adaptation or custom engineering, demonstrating that the framework already works out‑of‑the‑box across three different benchmarks. We are actively extending the framework to additional benchmarks, including coding, tool‑use, long‑context, and more complex multi‑turn interactions, following exactly the same methodology.

---

> ### Author Response · Authors · 2025-11-25
> **Response to Weakness 3: "complexity and practical burden on practitioners might be too high; unclear scalability"**
>
> We understand the concern that our MES/A‑MES framework may appear complex or costly. While the initial evaluation exceeds a single-number benchmark, this cost is intrinsic to rigorous attribution: standard evaluations cannot disentangle the effects of question format, prompting, decoding parameters, or other factors.
>
> Controlling cost and ensuring scalability. We do not exhaustively explore all MES configurations; instead, we sample configurations grounded in convergence and law of large numbers stopping criterion, keeping evaluation bounded. Tooling automates configuration space construction and sampling, A‑MES transformations, and ANOVA analysis, so users only need to specify which C‑variables to include, their value ranges, and their evaluation budget. Experiments show that a moderate number of sampled configurations suffices to obtain stable variable-importance rankings.
>
> Practical insights for future use. The initial full evaluatology run identifies high-impact variables (e.g., Question Format, max_tokens) and low-impact ones (e.g., presence_penalty). Subsequent users can focus on the key factors, holding others constant, to capture most insights at a fraction of the cost. In this way, the first run, though more expensive, produces a prioritized, actionable view of the evaluation space, enabling lighter-weight, task-tailored evaluatology afterward.

---

> ### Author Response · Authors · 2025-11-25
> **Response to Weakness 4: "limited human evaluations --> maybe human ablation could be added"**
>
> Our current experiments focus on tasks with objective ground-truth answers (e.g., AIME, MMLU, GPQA). Even after A‑MES transformations, we explicitly ensure that each item still has a well-defined, automatically checkable correct answer. Under this setting, human judgments are, in principle, expected to align closely with the ground truth.
>
> We fully agree that human ablation studies are very valuable, especially for open‑ended or subjective tasks, where human evaluation is essential for assessing output quality, error types, and nuanced behavior that cannot be reduced to a single correct answer. In ongoing follow‑up work on open‑ended question evaluation, we are planning to adopt exactly this strategy: sampling outputs under different MES/A‑MES configurations and having human annotators (experts or crowd workers) provide detailed judgments, then comparing these with automatic metrics and ANOVA‑based importance analyses.

---

> ### Author Response · Authors · 2025-11-25
> **Response to Weakness 5: "causal language might be overstated"**
>
> Thank you for raising this point. Our "causal" contribution lies in  contributing a systematic approach to uncovering causal relationships in complex LLM evaluation systems. Specifically, we construct a minimal evaluation system (MES and A‑MES) that explicitly defines the essential components of evaluation and systematically controls them. This allows us to attribute observed performance differences to specific factors and their interactions, addressing the core challenge that in complex systems “everyone is a stakeholder.”
>
> Existing LLM evaluations do not emphasize the notion of attribution and tend to overlook the impact of confounding factors. In practice, the observed accuracy depends on workload format, prompt methods, decoding parameters, and even system‑level factors. However, current benchmarks typically do not ask whether the measured accuracy is due to the model itself or to these confounders. We make this attribution question explicit and place it at the center of our evaluation design.
>
> To our knowledge, we are the first to define a minimal evaluation system covering 10 key factors and to systematically analyze their contributions to LLM output variance. While our work does not introduce conventional causal inference methods, our work makes a foundational contribution by providing a systematic approach to characterize true causal effects and accurately attribute performance differences in complex LLM evaluation systems. Beyond technical implementation, this framework establishes a principled methodology that moves LLM evaluation beyond single-setting snapshots, enabling more interpretable, reproducible, and comprehensive assessments across diverse conditions. we will clarify this distinction in the revised manuscript to avoid overstating the use of causal language.

---

> ### Author Response · Authors · 2025-11-25
> **Response to Question 1: "Which parts of A-MES generation are automated and which require expert authoring? How would scalability be possible?"**
>
> Regarding "Which parts of A‑MES generation are automated and which require expert authoring? How would scalability be possible?", the detailed automation pipeline of A‑MES has already been described in our response to Weakness 1, and its scalability is discussed in our response to Weakness 2.
>
> In brief:
>
> - Automation
>
>   As detailed under Weakness 1, A‑MES is instantiated as a set of script‑driven pipelines (three analogical and two novel branches), all of which are executed automatically via:
>   - fixed, reusable prompts and scaffolding,
>   - LLM API calls,
>   - programmatic verification,
>   - and retrieval from external repositories or parsed textbooks.
>   There is no per‑item manual authoring or hand‑crafting of individual problems. Once the original workload is given, the augmentation is "one‑click": the scripts generate all A‑MES instances end‑to‑end.
>
> - Scalability
>
>   As described under Weakness 2, the framework is fully generalizable in the sense that:
>   - The MES variables (C1–C10) and their ranges are defined once, and the MES/A‑MES configuration space is generated automatically.
>   - The sampling, convergence‑based stopping rule, and ANOVA‑based analysis are identical across benchmarks.
>   - In our experiments, the same implementation runs unchanged on AIME (math reasoning), MMLU (broad knowledge), and GPQA (scientific reasoning), without any benchmark‑specific engineering.
>   We are currently extending the same pipelines to coding, tool‑use, multi-turn and long‑context settings, again without changing the core methodology.

---

> ### Author Response · Authors · 2025-11-25
> **Response to Question 2: "DO you expect ANOVA factor importance to generalize across models and tasks, or is it benchmark-specific?"**
>
> Our empirical results indicate that ANOVA‑based factor importance exhibits both cross‑benchmark regularities and benchmark‑specific characteristics.
>
> On the regularity side, some factors consistently emerge as highly influential across models and benchmarks. For example, in all the benchmarks we analyzed, both Question Format and COT have large and statistically significant effects on accuracy. This suggests that these factors are not idiosyncratic to a single dataset, but rather reflect general sensitivities of current LLMs to how questions are formatted and whether reasoning is explicitly elicited.
>
> On the benchmark‑specific side, we also observe clear differences. A salient example is Question Paraphrase: on AIME, Question Paraphrase has negligible effect, indicating that paraphrasing difficult math problems does not reliably change model accuracy; in contrast, on MMLU, paraphrasing often becomes a key factor— for several models, Question Paraphrase reach statistical significance and explain a non‑trivial share of variance.
>
> In ongoing work, we are extending the ANOVA analysis to more tasks and models to more systematically characterize which factors generalize and which are dataset‑dependent. We will update the paper with these expanded results as soon as they are ready.

---

> ### Author Response · Authors · 2025-11-25
> **Response to Question 3: "How should practitioners choose a single evaluations core after MES/A-MES exploration? What is the summary statistic?"**
>
> This is a very valuable question, and we acknowledge that this part was not fully explicit in the paper. Ultimately, practitioners need a single evaluation score per model. In our framework, this is addressed statistically and guided by the intended evaluation goals.
>
> First, the selection of a representative score can be aligned with the stakeholders’ priorities—for example, focusing on specific evaluation dimensions or practical usage scenarios that matter most.
>
> Second, the A‑MES sampling space naturally subsumes the MES space, as each original MES instance is included as one of the possible augmentations. By sampling from the broader A‑MES space, we effectively cover a wider range of configurations and potential use cases, while still maintaining a non-zero probability of evaluating the original MES settings.
>
> The final reported evaluation score for each model is then the mean performance over the sampled instances, together with 95% and 99% confidence intervals, providing a stable summary metric that balances comprehensiveness with practical efficiency.

---

### Official Review · Reviewer_MRzL · 2025-11-01

**Soundness:** 2
**Presentation:** 4
**Contribution:** 3
**Rating:** 6
**Confidence:** 3

**Summary:**

This paper points out that using existing benchmark to provide a single evaluation of LLM may be reliable, as the authors find out that the same LLM’s performance can vary significantly in a benchmark given different configuration. Therefore, the authors uses Minimal Evaluation System for LLM evaluation and specify 10 dimensions, covering the question posted to LLM, method of prompting as well as parameters like temperature. The authors further augment existing benchmarks by generating transformed questions or entirely new questions to test the robustness of LLM in novel settings. The authors also evaluate the importance of the 10 dimensions for each LLM.

**Strengths:**

Originality: this paper argues that the configuration of a evaluation settings is important and define 10 important dimensions for the evaluation settings, which is a unique contribution to make the evaluation more systematic.

Quality: this paper applies extensive experiments on recent LLMs, which shows rich insights of robustness of models.

Clarity: this paper presents the idea in a relatively clear way, especially the literature review part, which systematically categorizes and covers the major approaches of LLM evaluation.

Significance: this paper tries to solve a very fundamental problem, which is building a systematic framework to evaluate the LLM, rather than testing LLM in a single setting, which could lead to evaluation bias.

**Weaknesses:**

1.	This paper doesn’t reveal enough details of implementation of the workload augmentation part. For instance, in “distractor insertion” in Transformed part, the authors didn’t mention how to insert distractor that is relevant to the question context. Note that we can always insert random sentences into questions, but if those sentences are totally out of the context of question, those sentences may mimic how LLM users’ additional comments distracts LLM in real life usage. More importantly, for Out-of-distribution workloads, the authors didn’t specify how to generate those new workloads. The authors didn’t list the algorithm but just mention that they rely on “textbook/academic statements”. However, generating new questions are inherently challenging and non-trivial, which is a critical gap that needs to fill if the authors would like to claim this contribution of A-MES.

2.	This paper’s attribution method themselves are value because it is essentially randomly controlled experiment, but I’m not sure whether the authors highlight them as “high dimensional causal attribution”. The word “causal” in title and abstract makes readers think that authors make some unique contributions in applying some novel causal method (say, treatment effect estimation) in observed data, which is really challenging. But in fact, the contribution to the causal method itself is not really there, as authors just apply ANOVA method on randomly controlled experiment data, which is correct but not inherently challenging in causal.

**Questions:**

1.	In line 242 to 249 about Analogical transformation (Transformed) workloads, do you have a systematic and scalable way to generate “distractors”? I would assume those distractors  should be still relevant to the questions’ context to actually serve as a distractor, is it true (just as your example in line 178 to 181 , you add that “symmetric” sentence, which still seems to be relevant to the topic of geometry)? If so, what are the ways to generate distractors that are relevant to this topic? Did you use a separate LLM to generate high quality distractors?  Also, how do you “swapping problem statements and conditions”? In some math problems, if you put the question as your condition, and your condition as question, I don’t think it is a valid math problem anymore. Or by “swapping”, do you mean just swap their positions in sentences?

2.	In line 250 to 255, first, the method (a) of recent-source adaptation (harvesting new questions such as new college entrance examination questions) itself is sound, but how does this method differ from those Dynamic or continuously refreshed benchmarks such as LiveCodeBench?  In method (b), how do you make sure the questions you generated from textbook/academic statements are valid questions themselves? Also, how do you generate the correct answers of those questions. If you generate those new (question, answer) pairs from LLM, then it doesn’t seem to make sense, because those questions are inherently new in nature and there is no guarantee the generating-LLM can generate the 100% correct answers.

3.	In Table 2, why would the value of Question Paraphrase be binary (Yes and No)? I thought to paraphrase a question, there should be multiple ways to do it (rather than just yes or no).

4.	In line 324, is 500 random sampling enough? I think in Section 4.1, the total combination of all configuration is more than 7,000. Therefore, is 500 random sample enough to cover all configuration?

---

> ### Author Response · Authors · 2025-11-25
> **Response to Reviewer MRzL**
>
> Dear Reviewer MRzL, thank you for carefully reviewing our submission and providing detailed, constructive feedback. We appreciate your recognition of our work’s originality, particularly our definition of ten evaluation dimensions, as well as your positive comments on our experiments, presentation, and the importance of moving beyond single-setting evaluations. We also value your critiques, including requests for more details on workload augmentation, concerns about the use of ANOVA in supporting causal claims, and questions on specific design choices. We address each point below and will incorporate the corresponding clarifications in the revised version.

---

> ### Author Response · Authors · 2025-11-25
> **Response to weakness 1: Limited details on the implementation of workload augmentation (1/5)**
>
> Due to space limitations, we only briefly outlined the A-MES part in the submission. Below, we provide a more detailed description of this implementation and will incorporate these clarifications and examples into the revised version.
>
> In our implementation, A‑MES is instantiated as five systematically defined, script‑driven transformation pipelines including: three analogical (distractor insertion, numeric substitution, conditional recomposition) and two novel (recent‑source adaptation and conceptual synthesis). For each pipeline, we fix general prompts and scaffolding, and then run the entire process automatically through LLM API calls (e.g., GPT‑5), lightweight verification scripts and other auxiliary tooling. This setup scales to large workloads and produces diverse variants, without any per‑item manual rewriting or hand‑crafting of individual problems. Moreover, we are actively exploring additional automated AIME‑style transformation pipelines to further enrich A‑MES.
>
> Overall, these five transformation pipelines realize seven concrete augmentation mechanisms for each workload (here, a workload means a question within the benchmark):
> 1. Distractor Insertion
>    - Context‑irrelevant distractor insertion
>    - Context‑relevant explanatory distractor insertion
>    - Context‑relevant misleading distractor insertion
> 2. Numeric substitutions
> 3. Conditional recomposition
> 4. Recent‑source adaptation
> 5. Conceptual synthesis
>
> These mechanisms show that A‑MES is not a collection of ad‑hoc, hand‑edited examples, but a unified, automatable framework that systematically augments existing benchmarks to more comprehensively evaluate model capabilities. For each question within a benchmark, we first apply all seven mechanisms to construct the full space of augmented variants, filtering out transformation attempts that fail (e.g., numeric substitutions or conditional recomposition without a stable solver). Evaluation then samples directly from this augmentation space, ensuring that no invalid transformations are ever selected. The framework enumerates the entire augmentation space upfront, and the small number of discarded variants has negligible impact on overall coverage or robustness. For several major benchmarks, we have already constructed the corresponding augmentation spaces and plan to release them publicly in future work. Below, we provide concrete descriptions of the seven mechanisms.

---

> ### Author Response · Authors · 2025-11-25
> **Response to weakness 1: Limited details on the implementation of workload augmentation (2/5)**
>
> ## 1. Analogical‑1: distractor insertion with three well-defined categories
>
> For distractor insertion, we define three explicit, controllable categories of redundancy and implement all instances via LLM prompting. To ensure that the inserted distractors strictly follow our predefined specifications, we empirically test several candidate LLMs and choose the one that most consistently adheres to these constraints (GPT-5). This selection is made solely to guarantee transformation fidelity rather than to compare model capabilities. For each item to be transformed, the chosen LLM is invoked through an API and, guided by our structured prompts, automatically produces and inserts the required redundant content. The prompts are provided below.
>
> - **Context‑irrelevant redundancy**
>   - Provide the LLM with an example containing an original question and a version with added context‑irrelevant redundancy.
>   - Instruct the LLM to insert one sentence at a random position that is completely unrelated to the target question.
>
> - **Context‑relevant, explanatory redundancy**
>   - Provide the LLM with an example of an original question and a version with added explanatory redundancy.
>   - Instruct the LLM to insert a redundant sentence at a random position in each target question that explains a concept already appearing in the target question.
>
> - **Context‑relevant, misleading redundancy**
>   - Provide the LLM with an example containing an original question and a version with added misleading but logically related redundancy.
>   - Supply the model with the correct answer and several correct solution approaches, and instruct it to avoid directly hinting at these correct strategies when crafting the misleading cue. The official answer and solution approaches are provided by the user, and providing solution approaches is optional.
>   - Instruct the model to insert a redundant sentence that nudges the reader toward an incorrect strategy or line of reasoning, without explicitly revealing that it is "misleading" or "distracting".
>
> In practice, the selected LLM produces variations that are more diverse and linguistically natural than manual editing. In particular, its context‑relevant misleading redundancies tend to hint at incorrect heuristics in a more subtle way than hand‑written versions, while still strictly adhering to the predefined category constraints. The entire process involves no per‑item manual editing. The three examples of redundancy for three types generated by the above procedure are illustrated as follows:
>
> 1. Context‑irrelevant redundancy example:
>    > *The weather today seems quite pleasant, and it might be a great day for a picnic.* Find the number of triples of nonnegative integers $(a,b,c)$ satisfying $a + b + c = 300$ and \[a^2b + a^2c + b^2a + b^2c + c^2a + c^2b = 6,000,000.\]
>
>    Here, weather is entirely unrelated to the math content.
>
> 2. Context‑relevant, explanatory redundancy example:
>    > There exist real numbers $x$ and $y$, both greater than 1, such that $\log_x\left(y^x\right)=\log_y\left(x^{4y}\right)=10$. *A logarithm is a way to express how many times a base must be multiplied by itself to get a certain number*. Find $xy$.
>
>    The added sentence explains the notion of a logarithm while leaving the underlying problem unchanged.
>
> 3. Context‑relevant, misleading redundancy example:
>    > Alice and Bob play the following game. A stack of $n$ tokens lies before them. The players take turns with Alice going first. On each turn, the player removes either $1$ token or $4$ tokens from the stack. *Many players adopt a greedy approach here: always take $4$ whenever possible to shorten the game and restrict the opponent's replies.* Whoever removes the last token wins. Find the number of positive integers $n$ less than or equal to $2024$ for which there exists a strategy for Bob that guarantees that Bob will win the game regardless of Alice's play.
>
>     The extra sentence about the "greedy approach" is logically related to the game but suggests a flawed strategy, intentionally nudging the solver toward an incorrect line of reasoning

---

> ### Author Response · Authors · 2025-11-25
> **Response to weakness 1: Limited details on the implementation of workload augmentation (3/5)**
>
> ## 2. Analogical‑2: numeric substitutions via code‑based solution extraction
>
> For numeric substitutions, we use a uniform pipeline built around LLM‑generated Python solvers and automatic verification scripts, rather than manually changing a few numbers:
>
> 1. We first call an LLM to extract the primary knowledge points tested by the original problem, and query a pre‑constructed formula library indexed by knowledge point to retrieve potentially relevant formulas.
> 2. We feed the original problem, its official answer, the retrieved formulas, and (where available) multiple correct solution sketches into the LLM. The official answer and solution sketches are provided by the user, and providing solution sketches is optional.
> 3. The LLM is prompted to:
>    - analyze the problem's solution strategy, using the provided solution sketches when available
>    - write a Python solution program where problem‑specific numbers are extracted as explicit variables with reasonable value ranges.
> 4. We then ask another LLM to inspect the generated Python code and verify that it implements a general computational procedure for solving the problem, rather than relying on hard‑coded instance‑specific outputs or trivial pattern matching.
> 5. We import the LLM‑generated Python code as a local module and call its solve() function with the original numeric values as inputs, checking whether the resulting output matches the official answer.
> 6. If the code fails (wrong answer or runtime error), we return the error message and incorrect output to the LLM, asking it to refine the code; we repeat this refinement–verification loop for up to five attempts and keep the Python code if it passes on the original instance.
>
> After obtaining a correct solver, we automatically sample new numeric configurations within the validated ranges to generate analogical variants of the same underlying problem. For example:
>
> - **Original:** Find the largest possible real part of \[(75+117i)z + \frac{96+144i}{z}\] where $z$ is a complex number with $|z|=4$. A common shortcut is to take $z$ to be a positive real number, since for a fixed modulus the real part is often largest when the argument of $z$ is zero.
> - **Numeric variant:** Find the largest possible real part of \[(100+112i)z + \frac{60+144i}{z}\] where $z$ is a complex number with $|z|=4$. A common shortcut is to take $z$ to be a positive real number, since for a fixed modulus the real part is often largest when the argument of $z$ is zero.
>
> This "knowledge‑point extraction → formula retrieval → analyze → code → verify → resample" pipeline is identical across all problems.

---

> ### Author Response · Authors · 2025-11-25
> **Response to weakness 1: Limited details on the implementation of workload augmentation (4/5)**
>
> ## 3. Analogical‑3: conditional recomposition via invertible‑condition analysis
>
> For conditional recompositions, we again adopt a general and automatable pipeline built around LLM‑generated Python solvers and automatic verification scripts rather than manually rewriting statements:
>
> 1. We first call an LLM to extract the primary knowledge points tested by the original question, and query a pre‑constructed formula library indexed by knowledge point to retrieve potentially relevant formulas.
> 2. We feed the original problem, its official answer, the retrieved formulas, and (where available) multiple correct solution sketches into the LLM. The official answer and solution sketches are provided by the user, and providing solution sketches is optional.
> 3. The LLM is prompted to:
>    - identify the key conditions and the target quantity;
>    - determine whether some of these conditions and the target can be interchanged—i.e., whether knowing the original answer allows us to infer some of the original conditions (an invertible relationship).
> 4. When such an invertible relationship exists, the LLM is asked to write a Python solution program for the recomposed problem, where the original target now appears as an input condition and (a subset of) the original conditions become the new target.
> 5. We then ask another LLM to inspect the generated Python code and verify that it implements a general computational procedure for solving the problem, rather than relying on hard‑coded instance‑specific outputs or trivial pattern matching.
> 6. We import the LLM‑generated Python code as a local module and call its solve() function, plugging the original answer value into the new "condition" slot and checking whether the returned output correctly recovers the original condition values.
> 7. Any discrepancy or runtime error is fed back to the LLM for iterative refinement, just as in the numeric substitutions pipeline. we repeat this refinement–verification loop for up to five attempts and keep the Python code if it passes on the instance.
>
> Once a correct solver for the recomposed version is obtained, we can further vary the new input variables within reasonable ranges to generate additional condition‑recomposed variants.
>
> - **Original:**
> > Rectangles $ABCD$ and $EFGH$ are drawn such that $D,E,C,F$ are collinear. Also, $A,D,H,G$ all lie on a circle. If $BC=16$,$AB=107$,$FG=17$, and $EF=184$, what is the length of $CE$?
> - **Conditional recomposition:**
> > Rectangles $ABCD$ and $EFGH$ are drawn such that $D,E,C,F$ are collinear. Also, $A,D,H,G$ all lie on a circle. If $BC=16$,$AB=107$,$CE=104$, and $EF=184$, what is the length of $FG$?
>
> These conditional recompositions are therefore produced by a uniform "knowledge‑point extraction → formula retrieval → analyze → code → verify → resample" pipeline, not by hand‑crafting each rephrased problem.

---

> ### Author Response · Authors · 2025-11-25
> **Response to weakness 1: Limited details on the implementation of workload augmentation (5/5)**
>
> ## 4. Novel‑1: recent‑source adaptation via structured retrieval and paraphrasing
>
> In the "novel" branch, the first mechanism is recent‑source adaptation, which is also fully scriptable:
>
> 1. We first use an LLM to extract the primary knowledge points tested by a given source problem.
> 2. We query open‑access repositories of centralized exam questions that index items by region, year, subject, and knowledge point, and crawl the most recent 2025 exam problems matching the extracted knowledge points.
> 3. The retrieved problems are paraphrased by the LLM and can be further transformed using the three analogical methods (redundancy insertion, numeric substitution, and conditional recomposition).
>
> This yields a set of new, recent‑source problems that are structurally aligned at the knowledge level but clearly distinct in surface form and provenance. The entire workflow is driven by scripts and general prompts, without hand‑curating individual items.
>
> ## 5. Novel‑2: textbook‑based conceptual synthesis via a parsed knowledge base
>
> The second "novel" mechanism is conceptual synthesis from authoritative textbooks. We first crawl a large collection of authoritative textbooks across different subjects from the web, and then use the LLM API's built‑in functionality for parsing local PDF files to extract their content. Based on the extracted content, we build a structured knowledge base in which each concept is associated with definitions, properties, theorems, phenomena, and canonical examples extracted from the textbooks.
>
> 1. Given a problem to be augmented, we use an LLM to identify its main knowledge points, and then retrieve the corresponding entries from the structured knowledge base. If the subject‑specific knowledge base is missing, we trigger the textbook crawling and parsing step to expand the knowledge base, and then retrieve the corresponding entries from it.
> 2. Conditioned on these entries, the LLM is prompted to generate new conceptual questions targeting the underlying knowledge points, rather than copying any existing problem.
>
> For example, one generated question associated with the concept of *logarithms* is:
> > What kind of mathematical idea/method turns exponentiation and multiplication into multiplication and addition?
>
> This pipeline turns textbook content into fresh conceptual questions that align with the original topic but are novel in form and focus.

---

> ### Author Response · Authors · 2025-11-25
> **Response to weakness 2: Use of the term "causal" for our attribution method**
>
> Thank you for raising this point. While our paper does not introduce new causal inference methods, it contributes a systematic approach to uncovering causal relationships in complex LLM evaluation systems. Specifically, we construct a minimal evaluation system (MES and A‑MES) that explicitly defines the essential components of evaluation and systematically controls them. This allows us to attribute observed performance differences to specific factors and their interactions, addressing the core challenge that in complex systems “everyone is a stakeholder.”
>
> Existing LLM evaluations rarely focus on attribution and often overlook confounding factors such as workload format, prompt design, decoding parameters, or system-level effects. By explicitly incorporating these factors and systematically sampling the resulting configuration space, our approach quantifies how much variance each factor explains, making the attribution question central to the evaluation.
>
> To our knowledge, we are the first to define a minimal evaluation system covering 10 key factors and to systematically analyze their contributions to LLM output variance. While the development of new causal inference methods remains an important direction, our work makes a foundational contribution by providing a systematic approach to characterize true causal effects and accurately attribute performance differences in complex LLM evaluation systems. Beyond technical implementation, this framework establishes a principled methodology that moves LLM evaluation beyond single-setting snapshots, enabling more interpretable, reproducible, and comprehensive assessments across diverse conditions.

---

> ### Author Response · Authors · 2025-11-25
> **Response to question 1: Implementation of analogical transformations in A‑MES**
>
> Thank you for this detailed question. We provided a detailed description of the implementation of the analogical transformation workloads in our response to Weakness 1. Please kindly refer to that section for more details.

---

> ### Author Response · Authors · 2025-11-25
> **Response to question 2: Implementation of novel workloads in A‑MES**
>
> Thank you for this question. The concrete procedures for both recent‑source adaptation and conceptual synthesis are detailed in our response to Weakness 1, where we describe how they fit into the broader A‑MES construction pipeline.
>
> Regarding your concerns about recent‑source adaptation, we clarify as follows. First, recent‑source adaptation is only one branch of our augmentation design; it is complemented by conceptual synthesis and the analogical transformations. Together, these seven mechanisms form a systematic augmentation methodology. In addition, the MES framework itself also provides a rich configuration space. This structured design goes beyond simply refreshing items over time, and is not present in existing dynamic benchmarks such as LiveCodeBench.
>
> Second, for recent‑source adaptation, our goal is not only to "harvest new questions", but to systematically align them to specific knowledge points of our base workloads, and then optionally apply the analogical pipelines on top. This gives us knowledge‑controlled novelty, rather than a generic continuously updated pool.
>
> Regarding conceptual synthesis, we do not ask the LLM to generate arbitrary new questions and answers. Instead, the generation process is grounded in authoritative textbooks: we parse standard textbooks into a structured knowledge base, retrieve the relevant entries for a knowledge points corresponding to a given workload item, and strictly constrain the LLM's generation to these vetted materials when producing conceptual questions. This grounding in authoritative sources significantly improves the reliability of the generated (question, answer) pairs compared with unconstrained LLM generation. In other words, our conceptual synthesis is not free‑form generation "out of thin air", but a controlled transformation of trusted textbook content, which is precisely how we ensure that the resulting questions are well‑posed and their answers correct.

---

> ### Author Response · Authors · 2025-11-25
> **Response to question 3: Binary coding of "Question Paraphrase" in Table 2**
>
> The "Question Paraphrase" column in Table 2 is binary by design, which does not imply that there is only a single way to paraphrase a question. The binary flag distinguishes two regimes:
>
> - No (original): the original question text is used, which may appear in a model's training data.
> - Yes (paraphrased): the question is restated in different wording while preserving all underlying semantics, answers, numerical values, and natural language constraints, avoiding potential data‑contamination issues.
>
> To construct paraphrases, we first compare several candidate LLMs on a small validation set and select the one that reliably follows these constraints. This model is then instructed to restate each question, ensuring semantic equivalence while changing surface wording. The purpose of this binary indicator is to diagnose memorization and data‑contamination effects: comparing model performance on original versus paraphrased items reveals whether behavior is robust to rewording or overly tied to specific training instances. The specific paraphrasing strategy or number of rewrites is irrelevant for this goal.
>
> Within MES, all workload transformations—including Question Paraphrase—preserve intrinsic semantics, providing alternative surface realizations of the same task. By contrast, A‑MES analogical transformations (distractor insertion, numeric substitution, condition recomposition) alter problem structure while maintaining core reasoning patterns, producing a family of structurally varied, more challenging variants.Therefore, keeping Question Paraphrase as a binary indicator cleanly separates same‑semantics rewording from the structural, changed‑semantics regime in A‑MES, aligning the design with our evaluation objectives.

---

> ### Author Response · Authors · 2025-11-25
> **Response to question 4: Sufficiency of 500 random samples**
>
> Thank you for raising this concern. For each benchmark, we first construct a configuration space of 15,552 distinct settings defined by ten controllable variables (e.g., Language, Question Format, Question Paraphrase, Shot, Chain-of-Thought, Multi-Turn, temperature, top_p, presence_penalty, and max_tokens). We then generate workload-level variants for each setting using the seven augmentation mechanisms, producing a ~100,000-point configuration space. To implement random sampling while ensuring reproducibility and consistency across model evaluations, we randomly shuffle the diverse settings using a random seed to form a list and select the first <sample_size> configurations.
>
> This sample size choice is not arbitrary: we first determine per‑model sample sizes using an explicit convergence procedure, and then additionally validate them with an LLN‑based calculation. The final "500" threshold is thus a conservative upper cap backed by two independent criteria rather than a heuristic guess.
>
> Concretely, when testing each model on each workload, we process configurations in batches of 10. After every 10 samples, we compute the running mean accuracy and its 95% confidence interval, and we stop sampling once both of the following conditions are satisfied:
>
> 1. The absolute changes in the running mean accuracy over the last three consecutive updates are all smaller than 0.002; that is, the absolute differences between the last and second‑last, the second‑last and third‑last, and the third‑last and fourth‑last running means are each smaller than 0.002.
> 2. The length of the 95% confidence interval is smaller than 0.06.
>
> Using this rule, the sample sizes at which different models converged were:
>
> - GPT‑4.1: 260
> - GPT‑3.5: 220
> - Mistral Medium: 260
> - Mistral Large: 290
> - Qwen Plus: 260
> - Qwen2.5: 400
> - DeepSeek‑V3: 320
> - Doubao‑1.5‑pro‑32k: 480
> - Moonshot‑v1: 220
>
> To further validate the above convergence-based sample sizes, we used a Law of Large Numbers (LLN)–based estimation. For each model, we first used the observed accuracies on the sampled configurations to estimated the variance of accuracy, and then computed the minimal sample size required to ensure that the sample mean is within a small error tolerance of the true mean with high probability. This gave the following estimates:
>
> - GPT‑4.1: 170
> - GPT‑3.5: 88
> - Mistral Medium: 229
> - Mistral Large: 253
> - Qwen Plus: 218
> - Qwen2.5: 329
> - DeepSeek‑V3: 285
> - Doubao‑1.5‑pro‑32k: 421
> - Moonshot‑v1: 63
>
> For every model, the LLN‑estimated minimal sample size is smaller than the sample size returned by the convergence method above. Therefore, we take the largest convergence‑based sample size across models as a baseline and add an additional safety margin, setting a global cap of 500 configurations for MES in Section 5.1, which also helps maintain cross‑model comparability in our evaluation.

---

### Official Review · Reviewer_fLKa · 2025-11-03

**Soundness:** 3
**Presentation:** 3
**Contribution:** 2
**Rating:** 4
**Confidence:** 2

**Summary:**

The paper argues that benchmark scores conflate model ability with many “indispensable” evaluation components (workloads, prompts, decoding, software/hardware). It proposes LLM evaluatology: define a Minimal Evaluation System (MES) and an Augmented MES (A-MES) that expands workloads (seen/analogical/novel) and then attribute performance via controlled sampling and ANOVA.

**Strengths:**

- The evaluation of LLM is of great importance in the research communities of LLM. Makes a compelling case that current single-config benchmarks are causally unfaithful and often misleading.
- Shows big accuracy variation across configs and non-transitive rank changes
- Quantifies main effects/interactions (Question Format, CoT, max tokens among top factors), lending interpretability to what actually matters.

**Weaknesses:**

- The proposed solutions seems trivial and handicrafted.
- The stopping rule (mean/CI thresholds) is reasonable, but guidance on sample sizes per workload/model and variance across random seeds would help practitioners replicate cost-accuracy trade-offs.
- Table 3 appears to list GPT-3.5 accuracy as 1.10, which must be a typo (accuracy >1). A thorough proofread for such inconsistencies would improve credibility.
- The main narrative leans heavily on AIME (plus MMLU/GPQA in appendix). Including code, tool-use, long-context, and multi-turn interactive settings in-depth would strengthen generality.

**Questions:**

See cons.

---

> ### Author Response · Authors · 2025-11-25
> **Response to Reviewer fLKa**
>
> Dear Reviewer fLKa,Thank you very much for your thoughtful and constructive review. We appreciate your recognition of our contributions, as well as your helpful feedback regarding the simplicity of our proposed methods, the need for clearer sampling guidance, the table typos, and the limited scope beyond AIME. We address these points in detail below and will incorporate the corresponding improvements in the revised version.

---

> ### Author Response · Authors · 2025-11-25
> **Response to weakness 1: "The proposed solution seems trivial and handcrafted." (1/5)**
>
> Thank you for raising this concern. Due to space constraints, the main text focused on the high‑level design of A‑MES and may have made the concrete pipeline appear more "handcrafted" than it actually is.
>
> In our implementation, A‑MES is instantiated as five systematically defined, script‑driven transformation pipelines including: three analogical (distractor insertion, numeric substitution, conditional recomposition) and two novel (recent‑source adaptation and conceptual synthesis). For each pipeline, we fix general prompts and scaffolding, and then run the entire process automatically through LLM API calls (e.g., GPT‑5), lightweight verification scripts and other auxiliary tooling. This setup scales to large workloads and produces diverse variants, without any per‑item manual rewriting or hand‑crafting of individual problems. Moreover, we are actively exploring additional automated AIME‑style transformation pipelines to further enrich A‑MES.
>
> Overall, these five transformation pipelines realize seven concrete augmentation mechanisms for each workload (here, a workload means a question within the benchmark):
>
> 1. Distractor Insertion
>    - Context‑irrelevant distractor insertion
>    - Context‑relevant explanatory distractor insertion
>    - Context‑relevant misleading distractor insertion
> 2. Numeric substitutions
> 3. Conditional recomposition
> 4. Recent‑source adaptation
> 5. Conceptual synthesis
>
> These mechanisms show that A‑MES is not a collection of ad‑hoc, hand‑edited examples, but a unified, automatable framework that systematically augments existing benchmarks to more comprehensively evaluate model capabilities. For each question within a benchmark, we first apply all seven mechanisms to construct the full space of augmented variants, filtering out transformation attempts that fail (e.g., numeric substitutions without a stable solver). Evaluation then samples directly from this augmentation space, ensuring that no invalid transformations are ever selected. The framework enumerates the entire augmentation space upfront, and the small number of discarded variants has negligible impact on overall coverage or robustness. For several major benchmarks, we have already constructed the corresponding augmentation spaces and plan to release them publicly in future work. Below, we provide concrete descriptions of the seven mechanisms.

---

> > ### Author Response · Authors · 2025-11-25
> > **Response to weakness 3: "GPT‑3.5 accuracy appears as 1.10 in Table 3 (typo)." (1/1)**
> >
> > Thank you for carefully catching this typo; we agree that such inconsistencies can harm the credibility of the results. The correct value for GPT‑3.5's accuracy in Table 3 is 0.1, not 1.10. In addition, we have re-checked all other reported numerical results in the paper and confirmed that there are no further inconsistencies. We will fix this typo in the revised manuscript and also perform a thorough proofread to eliminate similar issues.

---

> ### Author Response · Authors · 2025-11-25
> **Response to weakness 1: "The proposed solution seems trivial and handcrafted." (2/5)**
>
> ## 1. Analogical‑1: distractor insertion with three well-defined categories
>
> For distractor insertion, we define three explicit, controllable categories of redundancy and implement all instances via LLM prompting. To ensure that the inserted distractors strictly follow our predefined specifications, we empirically test several candidate LLMs and choose the one that most consistently adheres to these constraints (GPT-5). This selection is made solely to guarantee transformation fidelity rather than to compare model capabilities. For each item to be transformed, the chosen LLM is invoked through an API and, guided by our structured prompts, automatically produces and inserts the required redundant content. The prompts are provided below.
>
> - **Context‑irrelevant redundancy**
>   - Provide the LLM with an example containing an original question and a version with added context‑irrelevant redundancy.
>   - Instruct the LLM to insert one sentence at a random position that is completely unrelated to the target question.
>
> - **Context‑relevant, explanatory redundancy**
>   - Provide the LLM with an example of an original question and a version with added explanatory redundancy.
>   - Instruct the LLM to insert a redundant sentence at a random position in each target question that explains a concept already appearing in the target question.
>
> - **Context‑relevant, misleading redundancy**
>   - Provide the LLM with an example containing an original question and a version with added misleading but logically related redundancy.
>   - Supply the model with the correct answer and several correct solution approaches, and instruct it to avoid directly hinting at these correct strategies when crafting the misleading cue. The official answer and solution sketches are provided by the user, and providing solution sketches is optional.
>   - Instruct the model to insert a redundant sentence that nudges the reader toward an incorrect strategy or line of reasoning, without explicitly revealing that it is "misleading" or "distracting".
>
> In practice, the selected LLM produces variations that are more diverse and linguistically natural than manual editing. In particular, its context‑relevant misleading redundancies tend to hint at incorrect heuristics in a more subtle way than hand‑written versions, while still strictly adhering to the predefined category constraints. The entire process involves no per‑item manual editing. The three examples of redundancy for three types generated by the above procedure are illustrated as follows:
>
> 1. Context‑irrelevant redundancy example:
>    > *The weather today seems quite pleasant, and it might be a great day for a picnic.* Find the number of triples of nonnegative integers $(a,b,c)$ satisfying $a + b + c = 300$ and \[a^2b + a^2c + b^2a + b^2c + c^2a + c^2b = 6,000,000.\]
>
>    Here, weather is entirely unrelated to the math content.
>
> 2. Context‑relevant, explanatory redundancy example:
>    > There exist real numbers $x$ and $y$, both greater than 1, such that $\log_x\left(y^x\right)=\log_y\left(x^{4y}\right)=10$. *A logarithm is a way to express how many times a base must be multiplied by itself to get a certain number*. Find $xy$.
>
>    The added sentence explains the notion of a logarithm while leaving the underlying problem unchanged.
>
> 3. Context‑relevant, misleading redundancy example:
>    > Alice and Bob play the following game. A stack of $n$ tokens lies before them. The players take turns with Alice going first. On each turn, the player removes either $1$ token or $4$ tokens from the stack. *Many players adopt a greedy approach here: always take $4$ whenever possible to shorten the game and restrict the opponent's replies.* Whoever removes the last token wins. Find the number of positive integers $n$ less than or equal to $2024$ for which there exists a strategy for Bob that guarantees that Bob will win the game regardless of Alice's play.
>
>     The extra sentence about the "greedy approach" is logically related to the game but suggests a flawed strategy, intentionally nudging the solver toward an incorrect line of reasoning

---

> ### Author Response · Authors · 2025-11-25
> **Response to weakness 1: "The proposed solution seems trivial and handcrafted." (3/5)**
>
> ## 2. Analogical‑2: numeric substitutions via code‑based solution extraction
>
> For numeric substitutions, we use a uniform pipeline built around LLM‑generated Python solvers and automatic verification scripts, rather than manually changing a few numbers:
>
> 1. We first call an LLM to extract the primary knowledge points tested by the original problem, and query a pre‑constructed formula library indexed by knowledge point to retrieve potentially relevant formulas.
> 2. We feed the original problem, its official answer, the retrieved formulas, and (where available) multiple correct solution sketches into the LLM. The official answer and solution sketches are provided by the user, and providing solution sketches is optional.
> 3. The LLM is prompted to:
>    - analyze the problem's solution strategy, using the provided solution sketches when available
>    - write a Python solution program where problem‑specific numbers are extracted as explicit variables with reasonable value ranges.
> 4. We then ask another LLM to inspect the generated Python code and verify that it implements a general computational procedure for solving the problem, rather than relying on hard‑coded instance‑specific outputs or trivial pattern matching.
> 5. We import the LLM‑generated Python code as a local module and call its solve() function with the original numeric values as inputs, checking whether the resulting output matches the official answer.
> 6. If the code fails (wrong answer or runtime error), we return the error message and incorrect output to the LLM, asking it to refine the code; we repeat this refinement–verification loop for up to five attempts and keep the Python code if it passes on the original instance.
>
> After obtaining a correct solver, we automatically sample new numeric configurations within the validated ranges to generate analogical variants of the same underlying problem. For example:
>
> - **Original:** Find the largest possible real part of \[(75+117i)z + \frac{96+144i}{z}\] where $z$ is a complex number with $|z|=4$. A common shortcut is to take $z$ to be a positive real number, since for a fixed modulus the real part is often largest when the argument of $z$ is zero.
> - **Numeric variant:** Find the largest possible real part of \[(100+112i)z + \frac{60+144i}{z}\] where $z$ is a complex number with $|z|=4$. A common shortcut is to take $z$ to be a positive real number, since for a fixed modulus the real part is often largest when the argument of $z$ is zero.
>
> This "knowledge‑point extraction → formula retrieval → analyze → code → verify → resample" pipeline is identical across all problems.

---

> ### Author Response · Authors · 2025-11-25
> **Response to weakness 1: "The proposed solution seems trivial and handcrafted." (4/5)**
>
> ## 3. Analogical‑3: conditional recomposition via invertible‑condition analysis
>
> For conditional recompositions, we again adopt a general and automatable pipeline built around LLM‑generated Python solvers and automatic verification scripts rather than manually rewriting statements:
>
> 1. We first call an LLM to extract the primary knowledge points tested by the original question, and query a pre‑constructed formula library indexed by knowledge point to retrieve potentially relevant formulas.
> 2. We feed the original problem, its official answer, the retrieved formulas, and (where available) multiple correct solution sketches into the LLM. The official answer and solution sketches are provided by the user, and providing solution sketches is optional.
> 3. The LLM is prompted to:
>    - identify the key conditions and the target quantity;
>    - determine whether some of these conditions and the target can be interchanged—i.e., whether knowing the original answer allows us to infer some of the original conditions (an invertible relationship).
> 4. When such an invertible relationship exists, the LLM is asked to write a Python solution program for the recomposed problem, where the original target now appears as an input condition and (a subset of) the original conditions become the new target.
> 5. We then ask another LLM to inspect the generated Python code and verify that it implements a general computational procedure for solving the problem, rather than relying on hard‑coded instance‑specific outputs or trivial pattern matching.
> 6. We import the LLM‑generated Python code as a local module and call its solve() function, plugging the original answer value into the new "condition" slot and checking whether the returned output correctly recovers the original condition values.
> 7. Any discrepancy or runtime error is fed back to the LLM for iterative refinement, just as in the numeric substitutions pipeline. we repeat this refinement–verification loop for up to five attempts and keep the Python code if it passes on the instance.
>
> Once a correct solver for the recomposed version is obtained, we can further vary the new input variables within reasonable ranges to generate additional condition‑recomposed variants.
>
> - **Original:**
> > Rectangles $ABCD$ and $EFGH$ are drawn such that $D,E,C,F$ are collinear. Also, $A,D,H,G$ all lie on a circle. If $BC=16$,$AB=107$,$FG=17$, and $EF=184$, what is the length of $CE$?
> - **Conditional recomposition:**
> > Rectangles $ABCD$ and $EFGH$ are drawn such that $D,E,C,F$ are collinear. Also, $A,D,H,G$ all lie on a circle. If $BC=16$,$AB=107$,$CE=104$, and $EF=184$, what is the length of $FG$?
>
> These conditional recompositions are therefore produced by a uniform "knowledge‑point extraction → formula retrieval → analyze → code → verify → resample" pipeline, not by hand‑crafting each rephrased problem.

---

> ### Author Response · Authors · 2025-11-25
> **Response to weakness 1: "The proposed solution seems trivial and handcrafted." (5/5)**
>
> ## 4. Novel‑1: recent‑source adaptation via structured retrieval and paraphrasing
>
> In the "novel" branch, the first mechanism is recent‑source adaptation, which is also fully scriptable:
>
> 1. We first use an LLM to extract the primary knowledge points tested by a given source problem.
> 2. We query open‑access repositories of centralized exam questions that index items by region, year, subject, and knowledge point, and crawl the most recent 2025 exam problems matching the extracted knowledge points.
> 3. The retrieved problems are paraphrased by the LLM and can be further transformed using the three analogical methods (redundancy insertion, numeric substitution, and conditional recomposition).
>
> This yields a set of new, recent‑source problems that are structurally aligned at the knowledge level but clearly distinct in surface form and provenance. The entire workflow is driven by scripts and general prompts, without hand‑curating individual items.
>
> ## 5. Novel‑2: textbook‑based conceptual synthesis via a parsed knowledge base
>
> The second "novel" mechanism is conceptual synthesis from authoritative textbooks. We first crawl a large collection of authoritative textbooks across different subjects from the web, and then use the LLM API's built‑in functionality for parsing local PDF files to extract their content. Based on the extracted content, we build a structured knowledge base in which each concept is associated with definitions, properties, theorems, phenomena, and canonical examples extracted from the textbooks.
>
> 1. Given a problem to be augmented, we use an LLM to identify its main knowledge points, and then retrieve the corresponding entries from the structured knowledge base. If the subject‑specific knowledge base is missing, we trigger the textbook crawling and parsing step to expand the knowledge base, and then retrieve the corresponding entries from it.
> 2. Conditioned on these entries, the LLM is prompted to generate new conceptual questions targeting the underlying knowledge points, rather than copying any existing problem.
>
> For example, one generated question associated with the concept of *logarithms* is:
> > What kind of mathematical idea/method turns exponentiation and multiplication into multiplication and addition?
>
> This pipeline turns textbook content into fresh conceptual questions that align with the original topic but are novel in form and focus.

---

> ### Author Response · Authors · 2025-11-25
> **Response to weakness 2: "More guidance on sample sizes per workload/model and variance across random seeds." (1/2)**
>
> We greatly appreciate this suggestion and agree that providing concrete guidance is important. In our experiments, for each benchmark, we first construct a configuration space of 15,552 distinct settings defined by ten controllable variables (e.g., Language, Question Format, Question Paraphrase, Shot, Chain-of-Thought, Multi-Turn, temperature, top_p, presence_penalty, and max_tokens). We then generate workload-level variants for each setting using the seven augmentation mechanisms, producing a ~100,000-point configuration space. This forms the full configuration space for the benchmark. From this space, we determine the required sample size using two stopping criteria. To implement random sampling while ensuring reproducibility and consistency across model evaluations, we randomly shuffle the  diverse settings using a random seed to form a list and select the first <sample_size> configurations. This procedure, which was only briefly mentioned in the main text, will be described in detail in the revised version.
>
> ## Convergence-based stopping rule
>
> For each workload, when testing each model, we evaluate configurations sequentially in batches of 10 from the shuffled list. After every 10 samples, we compute the running mean accuracy and its 95% confidence interval. We stop sampling when both of the following conditions are satisfied:
>
> 1. The absolute changes in the running mean accuracy over the last three consecutive updates are all smaller than 0.002; that is, the absolute differences between the last and second‑last, the second‑last and third‑last, and the third‑last and fourth‑last running means are each smaller than 0.002.
> 2. The length of the 95% confidence interval is smaller than 0.06.
>
> Using this rule, the sample sizes at which different models converged were:
>
> - GPT‑4.1: 260
> - GPT‑3.5: 220
> - Mistral Medium: 260
> - Mistral Large: 290
> - Qwen Plus: 260
> - Qwen2.5: 400
> - DeepSeek‑V3: 320
> - Doubao‑1.5‑pro‑32k: 480
> - Moonshot‑v1: 220
>
> ## LLN-based stopping rule
>
> To further validate the above convergence-based sample sizes, we used a Law of Large Numbers (LLN)–based estimation. For each model, we first used the observed accuracies on the sampled configurations to estimated the variance of accuracy, and then computed the minimal sample size required to ensure that the sample mean is within a small error tolerance of the true mean with high probability. This gave the following estimates:
>
> - GPT‑4.1: 170
> - GPT‑3.5: 88
> - Mistral Medium: 229
> - Mistral Large: 253
> - Qwen Plus: 218
> - Qwen2.5: 329
> - DeepSeek‑V3: 285
> - Doubao‑1.5‑pro‑32k: 421
> - Moonshot‑v1: 63
>
> Importantly, for every model, the LLN-estimated minimal sample size is smaller than the sample size returned by the convergence method above. Therefore, we conservatively adopt the larger convergence-based sample sizes in our experiments.

---

> ### Author Response · Authors · 2025-11-25
> **Response to weakness 2: "More guidance on sample sizes per workload/model and variance across random seeds." (2/2)**
>
> ## Practical guidance for practitioners
>
> To make this concrete for other practitioners, we summarize how one can determine sample sizes and stopping criteria in a new setting, even with a different configuration space or benchmark:
>
> 1. **Define and enumerate the configuration space.**
>    - Identify the evaluation variables (analogous to our C1–C10 in Table 2: workload, prompting, decoding, etc.) and specify a finite value range for each.
>    - Enumerate the full Cartesian product of these values to obtain a discrete configuration space.
>
> 2. **Fix a global shuffled order and share it across models.**
>    - Use a fixed random seed to randomly permute the full list of configurations once.
>    - Reuse this single shuffled order for all models and workloads.
>    - Always sample configurations as the first N entries of this global list, rather than re‑shuffling per model or per run.
>
> 3. **Joint stopping rule based on convergence and the Law of Large Numbers**
>    - First fix a batch size (e.g., 10), and follow a globally shuffled configuration order; advance by one batch at a time (i.e., evaluate `batch size` configurations per step).
>    - After each batch, recompute, based on all configurations evaluated so far, the model’s mean accuracy on the given benchmark and the corresponding 95% confidence interval.
>    - When the following conditions are met:
>      1. The mean accuracy is effectively stable (e.g., the differences between the last four mean accuracies are all smaller than 0.002), and
>      2. The 95% confidence interval is sufficiently narrow (e.g., interval length < 0.06),
>      the estimate can be regarded as converged at the current sample size N_conv, yielding a converged sample size and mean accuracy.
>    - After convergence, apply a simple Law of Large Numbers–based estimation, using the variance of the current results to compute how many samples are theoretically required to achieve the desired error tolerance and confidence level, giving an "LLN-based sample size" N_LLN.
>    - If N_LLN is larger than the current N_conv, continue sampling along the same shuffled order until at least N_LLN configurations have been evaluated; otherwise, stop sampling.
>
> The entire evaluation workflow described above is implemented in an automation scripts. After publication, these scripts will be released as open source, so that practitioners can perform evaluation following the guidance with minimal effort.

---

> ### Author Response · Authors · 2025-11-25
> **Response to weakness 4: "The main narrative leans heavily on AIME; more in-depth code/long-context/multi-turn settings would strengthen generality." (1/1)**
>
> We fully agree that including more task types would make the empirical section stronger. At the same time, we would like to clarify that our LLM evaluatology framework is task-agnostic: once we define the set of evaluation variables (C1–C10) and their value ranges, the MES configuration space and the corresponding A‑MES workflows are automatically generated by our tooling. The procedures for random sampling, convergence-based stopping, and ANOVA remain unchanged regardless of whether the benchmark is math (AIME), knowledge (MMLU), science (GPQA), coding, or multi-turn, and in our current experiments we apply this pipeline to AIME, MMLU, and GPQA with no benchmark-specific adaptation or custom engineering, demonstrating that the framework already works out-of-the-box across three different benchmarks. We are actively extending the framework to additional benchmarks, including coding, tool-use, long-context, and more complex multi-turn interactions, following exactly the same methodology. The full automation tooling will be released as open source upon publication.

---

### Author Response · Authors · 2025-12-03
**Summary of rebuttal and revisions for AC**

Dear Area Chair,
We sincerely thank you for your time and effort in handling our submission, especially given the complications introduced by the system issues.

Across the reviews, there is clear consensus on the key strengths of our work:
1. a systematic evaluation framework, where the 10 clearly defined dimensions make LLM evaluation more structured and principled;
2. compelling evidence that single-configuration benchmarks often produce causally unfaithful inferences by conflating model ability with configuration choices; and
3. interpretable attribution analysis, enabled by ANOVA, that reveals the main effects and interactions driving performance differences.

The reviewers also raised several constructive concerns, all of which we have thoroughly addressed in the revision and rebuttal. We added missing implementation details, clarified the generalizability of A-MES, refined the sampling and stopping criteria, moderated the causal claims, and provided both cost analyses and justification for the configuration ranges.

Notably, Reviewer 6AVH (rating 2, confidence 4) explicitly stated that our rebuttal resolved most of their concerns and that they are willing to raise their score, which we believe indicates the effectiveness of our clarifications.

A detailed point-by-point response is provided below, and we respectfully submit these updates for the AC's consideration in the final assessment.

---

> ### Author Response · Authors · 2025-12-03
> **# Issues 1: Lack of sufficient implementation details of A-MES.(Reviewer fLKa weakness 1; Reviewer MRzL weakness 1, question 1,2; Reviewer 5j6t question 1; Reviewer 5j6t weakness 1) (1/5)**
>
> In our implementation, A‑MES is instantiated as five systematically defined, script‑driven transformation pipelines including: three analogical (distractor insertion, numeric substitution, conditional recomposition) and two novel (recent‑source adaptation and conceptual synthesis). For each pipeline, we fix general prompts and scaffolding, and then run the entire process automatically through LLM API calls (e.g., GPT‑5), lightweight verification scripts and other auxiliary tooling. This setup scales to large workloads and produces diverse variants, without any per‑item manual rewriting or hand‑crafting of individual problems. Moreover, we are actively exploring additional automated AIME‑style transformation pipelines to further enrich A‑MES.
>
> Overall, these five transformation pipelines realize seven concrete augmentation mechanisms for each workload (here, a workload means a question within the benchmark):
>
> 1. Distractor Insertion
>    - Context‑irrelevant distractor insertion
>    - Context‑relevant explanatory distractor insertion
>    - Context‑relevant misleading distractor insertion
> 2. Numeric substitutions
> 3. Conditional recomposition
> 4. Recent‑source adaptation
> 5. Conceptual synthesis
>
> These mechanisms show that A‑MES is not a collection of ad‑hoc, hand‑edited examples, but a unified, automatable framework that systematically augments existing benchmarks to more comprehensively evaluate model capabilities. For each question within a benchmark, we first apply all seven mechanisms to construct the full space of augmented variants, filtering out transformation attempts that fail (e.g., numeric substitutions or conditional recomposition without a stable solver). Evaluation then samples directly from this augmentation space, ensuring that no invalid transformations are ever selected. The framework enumerates the entire augmentation space upfront, and the small number of discarded variants has negligible impact on overall coverage or robustness. For several major benchmarks, we have already constructed the corresponding augmentation spaces and plan to release them publicly in future work. Below, we provide concrete descriptions of the seven mechanisms.

---

> ### Author Response · Authors · 2025-12-03
> **# Issues 1: Lack of sufficient implementation details of A-MES.(Reviewer fLKa weakness 1; Reviewer MRzL weakness 1, question 1,2; Reviewer 5j6t question 1; Reviewer 5j6t weakness 1) (2/5)**
>
> ## 1. Analogical‑1: distractor insertion with three well-defined categories
>
> For distractor insertion, we define three explicit, controllable categories of redundancy and implement all instances via LLM prompting. To ensure that the inserted distractors strictly follow our predefined specifications, we empirically test several candidate LLMs and choose the one that most consistently adheres to these constraints (GPT-5). This selection is made solely to guarantee transformation fidelity rather than to compare model capabilities. For each item to be transformed, the chosen LLM is invoked through an API and, guided by our structured prompts, automatically produces and inserts the required redundant content. The prompts are provided below.
>
> - **Context‑irrelevant redundancy**
>   - Provide the LLM with an example containing an original question and a version with added context‑irrelevant redundancy.
>   - Instruct the LLM to insert one sentence at a random position that is completely unrelated to the target question.
>
> - **Context‑relevant, explanatory redundancy**
>   - Provide the LLM with an example of an original question and a version with added explanatory redundancy.
>   - Instruct the LLM to insert a redundant sentence at a random position in each target question that explains a concept already appearing in the target question.
>
> - **Context‑relevant, misleading redundancy**
>   - Provide the LLM with an example containing an original question and a version with added misleading but logically related redundancy.
>   - Supply the model with the correct answer and several correct solution approaches, and instruct it to avoid directly hinting at these correct strategies when crafting the misleading cue. The official answer and solution approaches are provided by the user, and providing solution approaches is optional.
>   - Instruct the model to insert a redundant sentence that nudges the reader toward an incorrect strategy or line of reasoning, without explicitly revealing that it is "misleading" or "distracting".
>
> In practice, the selected LLM produces variations that are more diverse and linguistically natural than manual editing. In particular, its context‑relevant misleading redundancies tend to hint at incorrect heuristics in a more subtle way than hand‑written versions, while still strictly adhering to the predefined category constraints. The entire process involves no per‑item manual editing. The three examples of redundancy for three types generated by the above procedure are illustrated as follows:
>
> 1. Context‑irrelevant redundancy example:
>    > *The weather today seems quite pleasant, and it might be a great day for a picnic.* Find the number of triples of nonnegative integers $(a,b,c)$ satisfying $a + b + c = 300$ and \[a^2b + a^2c + b^2a + b^2c + c^2a + c^2b = 6,000,000.\]
>
>    Here, weather is entirely unrelated to the math content.
>
> 2. Context‑relevant, explanatory redundancy example:
>    > There exist real numbers $x$ and $y$, both greater than 1, such that $\log_x\left(y^x\right)=\log_y\left(x^{4y}\right)=10$. *A logarithm is a way to express how many times a base must be multiplied by itself to get a certain number*. Find $xy$.
>
>    The added sentence explains the notion of a logarithm while leaving the underlying problem unchanged.
>
> 3. Context‑relevant, misleading redundancy example:
>    > Alice and Bob play the following game. A stack of $n$ tokens lies before them. The players take turns with Alice going first. On each turn, the player removes either $1$ token or $4$ tokens from the stack. *Many players adopt a greedy approach here: always take $4$ whenever possible to shorten the game and restrict the opponent's replies.* Whoever removes the last token wins. Find the number of positive integers $n$ less than or equal to $2024$ for which there exists a strategy for Bob that guarantees that Bob will win the game regardless of Alice's play.
>
>     The extra sentence about the "greedy approach" is logically related to the game but suggests a flawed strategy, intentionally nudging the solver toward an incorrect line of reasoning

---

> ### Author Response · Authors · 2025-12-03
> **# Issues 1: Lack of sufficient implementation details of A-MES.(Reviewer fLKa weakness 1; Reviewer MRzL weakness 1, question 1,2; Reviewer 5j6t question 1; Reviewer 5j6t weakness 1) (3/5)**
>
> ## 2. Analogical‑2: numeric substitutions via code‑based solution extraction
>
> For numeric substitutions, we use a uniform pipeline built around LLM‑generated Python solvers and automatic verification scripts, rather than manually changing a few numbers:
>
> 1. We first call an LLM to extract the primary knowledge points tested by the original problem, and query a pre‑constructed formula library indexed by knowledge point to retrieve potentially relevant formulas.
> 2. We feed the original problem, its official answer, the retrieved formulas, and (where available) multiple correct solution sketches into the LLM. The official answer and solution sketches are provided by the user, and providing solution sketches is optional.
> 3. The LLM is prompted to:
>    - analyze the problem's solution strategy, using the provided solution sketches when available
>    - write a Python solution program where problem‑specific numbers are extracted as explicit variables with reasonable value ranges.
> 4. We then ask another LLM to inspect the generated Python code and verify that it implements a general computational procedure for solving the problem, rather than relying on hard‑coded instance‑specific outputs or trivial pattern matching.
> 5. We import the LLM‑generated Python code as a local module and call its solve() function with the original numeric values as inputs, checking whether the resulting output matches the official answer.
> 6. If the code fails (wrong answer or runtime error), we return the error message and incorrect output to the LLM, asking it to refine the code; we repeat this refinement–verification loop for up to five attempts and keep the Python code if it passes on the original instance.
>
> After obtaining a correct solver, we automatically sample new numeric configurations within the validated ranges to generate analogical variants of the same underlying problem. For example:
>
> - **Original:** Find the largest possible real part of \[(75+117i)z + \frac{96+144i}{z}\] where $z$ is a complex number with $|z|=4$. A common shortcut is to take $z$ to be a positive real number, since for a fixed modulus the real part is often largest when the argument of $z$ is zero.
> - **Numeric variant:** Find the largest possible real part of \[(100+112i)z + \frac{60+144i}{z}\] where $z$ is a complex number with $|z|=4$. A common shortcut is to take $z$ to be a positive real number, since for a fixed modulus the real part is often largest when the argument of $z$ is zero.
>
> This "knowledge‑point extraction → formula retrieval → analyze → code → verify → resample" pipeline is identical across all problems.

---

> ### Author Response · Authors · 2025-12-03
> **# Issues 1: Lack of sufficient implementation details of A-MES.(Reviewer fLKa weakness 1; Reviewer MRzL weakness 1, question 1,2; Reviewer 5j6t question 1; Reviewer 5j6t weakness 1) (4/5)**
>
> ## 3. Analogical‑3: conditional recomposition via invertible‑condition analysis
>
> For conditional recompositions, we again adopt a general and automatable pipeline built around LLM‑generated Python solvers and automatic verification scripts rather than manually rewriting statements:
>
> 1. We first call an LLM to extract the primary knowledge points tested by the original question, and query a pre‑constructed formula library indexed by knowledge point to retrieve potentially relevant formulas.
> 2. We feed the original problem, its official answer, the retrieved formulas, and (where available) multiple correct solution sketches into the LLM. The official answer and solution sketches are provided by the user, and providing solution sketches is optional.
> 3. The LLM is prompted to:
>    - identify the key conditions and the target quantity;
>    - determine whether some of these conditions and the target can be interchanged—i.e., whether knowing the original answer allows us to infer some of the original conditions (an invertible relationship).
> 4. When such an invertible relationship exists, the LLM is asked to write a Python solution program for the recomposed problem, where the original target now appears as an input condition and (a subset of) the original conditions become the new target.
> 5. We then ask another LLM to inspect the generated Python code and verify that it implements a general computational procedure for solving the problem, rather than relying on hard‑coded instance‑specific outputs or trivial pattern matching.
> 6. We import the LLM‑generated Python code as a local module and call its solve() function, plugging the original answer value into the new "condition" slot and checking whether the returned output correctly recovers the original condition values.
> 7. Any discrepancy or runtime error is fed back to the LLM for iterative refinement, just as in the numeric substitutions pipeline. we repeat this refinement–verification loop for up to five attempts and keep the Python code if it passes on the instance.
>
> Once a correct solver for the recomposed version is obtained, we can further vary the new input variables within reasonable ranges to generate additional condition‑recomposed variants.
>
> - **Original:**
> > Rectangles $ABCD$ and $EFGH$ are drawn such that $D,E,C,F$ are collinear. Also, $A,D,H,G$ all lie on a circle. If $BC=16$,$AB=107$,$FG=17$, and $EF=184$, what is the length of $CE$?
> - **Conditional recomposition:**
> > Rectangles $ABCD$ and $EFGH$ are drawn such that $D,E,C,F$ are collinear. Also, $A,D,H,G$ all lie on a circle. If $BC=16$,$AB=107$,$CE=104$, and $EF=184$, what is the length of $FG$?
>
> These conditional recompositions are therefore produced by a uniform "knowledge‑point extraction → formula retrieval → analyze → code → verify → resample" pipeline, not by hand‑crafting each rephrased problem.

---

> ### Author Response · Authors · 2025-12-03
> **# Issues 1: Lack of sufficient implementation details of A-MES.(Reviewer fLKa weakness 1; Reviewer MRzL weakness 1, question 1,2; Reviewer 5j6t question 1; Reviewer 5j6t weakness 1) (5/5)**
>
> ## 4. Novel‑1: recent‑source adaptation via structured retrieval and paraphrasing
>
> In the "novel" branch, the first mechanism is recent‑source adaptation, which is also fully scriptable:
>
> 1. We first use an LLM to extract the primary knowledge points tested by a given source problem.
> 2. We query open‑access repositories of centralized exam questions that index items by region, year, subject, and knowledge point, and crawl the most recent 2025 exam problems matching the extracted knowledge points.
> 3. The retrieved problems are paraphrased by the LLM and can be further transformed using the three analogical methods (redundancy insertion, numeric substitution, and conditional recomposition).
>
> For example, one recent-source adaptation question generated for the concept of *logarithms* is:
> > Given $2^{\log_{2} a} = 3$ and $\log_{5} 5^{b} = 2$, find $a - b$
>
> This yields a set of new, recent‑source problems that are structurally aligned at the knowledge level but clearly distinct in surface form and provenance. The entire workflow is driven by scripts and general prompts, without hand‑curating individual items.
>
> ## 5. Novel‑2: textbook‑based conceptual synthesis via a parsed knowledge base
>
> The second "novel" mechanism is conceptual synthesis from authoritative textbooks. We first crawl a large collection of authoritative textbooks across different subjects from the web, and then use the LLM API's built‑in functionality for parsing local PDF files to extract their content. Based on the extracted content, we build a structured knowledge base in which each concept is associated with definitions, properties, theorems, phenomena, and canonical examples extracted from the textbooks.
>
> 1. Given a problem to be augmented, we use an LLM to identify its main knowledge points, and then retrieve the corresponding entries from the structured knowledge base. If the subject‑specific knowledge base is missing, we trigger the textbook crawling and parsing step to expand the knowledge base, and then retrieve the corresponding entries from it.
> 2. Conditioned on these entries, the LLM is prompted to generate new conceptual questions targeting the underlying knowledge points, rather than copying any existing problem.
>
> For example, one generated question associated with the concept of *logarithms* is:
> > What kind of mathematical idea/method turns exponentiation and multiplication into multiplication and addition?
>
> This pipeline turns textbook content into fresh conceptual questions that align with the original topic but are novel in form and focus.

---

> ### Author Response · Authors · 2025-12-03
> **# Issue 2: Unclear whether the method can generalize. (Reviewer fLKa weakness 4; Reviewer 5j6t weakness 2)**
>
> Our methodology is not limited to public academic benchmarks and naturally extends to non‑benchmark settings. As long as there exists a seed workload—such as an enterprise’s internal question bank, a school or institutional exam repository, or other custom task collections—our MES/A‑MES pipeline can be instantiated on top of it without any conceptual change. In this view, the internal or proprietary workload simply serves as the workload component in our framework; our tooling then automatically constructs the corresponding configuration space, performs workload augmentation, and runs the same sampling and attribution procedures. This enables organizations to obtain causally interpretable, system‑level evaluations tailored to their own private or domain‑specific tasks, rather than being restricted to public academic reasoning benchmarks.
>
> Furthermore, our method is in fact fully generalizable. Specifically, we would like to clarify that our LLM evaluatology framework is task‑agnostic: once we define the set of evaluation variables (C1–C10) and their value ranges, the MES configuration space and the corresponding A‑MES workflows are automatically generated by our tooling. The procedures for random sampling, convergence‑based stopping, and ANOVA remain unchanged regardless of whether the benchmark is reasoning(AIME), knowledge (MMLU), science (GPQA), coding, or multi‑turn, and in our current experiments we apply this pipeline to AIME, MMLU, and GPQA with no benchmark‑specific adaptation or custom engineering, demonstrating that the framework already works out‑of‑the‑box across three different benchmarks. We are actively extending the framework to additional benchmarks, including coding, tool‑use, long‑context, and more complex multi‑turn interactions, following exactly the same methodology.

---

> ### Author Response · Authors · 2025-12-03
> **# Issue 3: Lack of guidance on sample size and variance (Reviewer fLKa weakness 2) (1/2)**
>
> In our experiments, for each benchmark, we first construct a configuration space of 15,552 distinct settings defined by ten controllable variables (e.g., Language, Question Format, Question Paraphrase, Shot, Chain-of-Thought, Multi-Turn, temperature, top_p, presence_penalty, and max_tokens). We then generate workload-level variants for each setting using the seven augmentation mechanisms, producing a ~100,000-point configuration space. This forms the full configuration space for the benchmark. From this space, we determine the required sample size using two stopping criteria. To implement random sampling while ensuring reproducibility and consistency across model evaluations, we randomly shuffle the  diverse settings using a random seed to form a list and select the first <sample_size> configurations. This procedure, which was only briefly mentioned in the main text, will be described in detail in the revised version.
>
> ## Convergence-based stopping rule
>
> For each workload, when testing each model, we evaluate configurations sequentially in batches of 10 from the shuffled list. After every 10 samples, we compute the running mean accuracy and its 95% confidence interval. We stop sampling when both of the following conditions are satisfied:
>
> 1. The absolute changes in the running mean accuracy over the last three consecutive updates are all smaller than 0.002; that is, the absolute differences between the last and second‑last, the second‑last and third‑last, and the third‑last and fourth‑last running means are each smaller than 0.002.
> 2. The length of the 95% confidence interval is smaller than 0.06.
>
> Using this rule, the sample sizes at which different models converged were:
>
> - GPT‑4.1: 260
> - GPT‑3.5: 220
> - Mistral Medium: 260
> - Mistral Large: 290
> - Qwen Plus: 260
> - Qwen2.5: 400
> - DeepSeek‑V3: 320
> - Doubao‑1.5‑pro‑32k: 480
> - Moonshot‑v1: 220
>
> ## LLN-based stopping rule
>
> To further validate the above convergence-based sample sizes, we used a Law of Large Numbers (LLN)–based estimation. For each model, we first used the observed accuracies on the sampled configurations to estimated the variance of accuracy, and then computed the minimal sample size required to ensure that the sample mean is within a small error tolerance of the true mean with high probability. This gave the following estimates:
>
> - GPT‑4.1: 170
> - GPT‑3.5: 88
> - Mistral Medium: 229
> - Mistral Large: 253
> - Qwen Plus: 218
> - Qwen2.5: 329
> - DeepSeek‑V3: 285
> - Doubao‑1.5‑pro‑32k: 421
> - Moonshot‑v1: 63
>
> Importantly, for every model, the LLN-estimated minimal sample size is smaller than the sample size returned by the convergence method above. Therefore, we conservatively adopt the larger convergence-based sample sizes in our experiments.

---

> ### Author Response · Authors · 2025-12-03
> **# Issue 3: Lack of guidance on sample size and variance (Reviewer fLKa weakness 2) (2/2)**
>
> ## Practical guidance for practitioners
>
> To make this concrete for other practitioners, we summarize how one can determine sample sizes and stopping criteria in a new setting, even with a different configuration space or benchmark:
>
> 1. **Define and enumerate the configuration space.**
>    - Identify the evaluation variables (analogous to our C1–C10 in Table 2: workload, prompting, decoding, etc.) and specify a finite value range for each.
>    - Enumerate the full Cartesian product of these values to obtain a discrete configuration space.
>
> 2. **Fix a global shuffled order and share it across models.**
>    - Use a fixed random seed to randomly permute the full list of configurations once.
>    - Reuse this single shuffled order for all models and workloads.
>    - Always sample configurations as the first N entries of this global list, rather than re‑shuffling per model or per run.
>
> 3. **Joint stopping rule based on convergence and the Law of Large Numbers**
>    - First fix a batch size (e.g., 10), and follow a globally shuffled configuration order; advance by one batch at a time (i.e., evaluate `batch size` configurations per step).
>    - After each batch, recompute, based on all configurations evaluated so far, the model’s mean accuracy on the given benchmark and the corresponding 95% confidence interval.
>    - When the following conditions are met:
>      1. The mean accuracy is effectively stable (e.g., the differences between the last four mean accuracies are all smaller than 0.002), and
>      2. The 95% confidence interval is sufficiently narrow (e.g., interval length < 0.06),
>      the estimate can be regarded as converged at the current sample size N_conv, yielding a converged sample size and mean accuracy.
>    - After convergence, apply a simple Law of Large Numbers–based estimation, using the variance of the current results to compute how many samples are theoretically required to achieve the desired error tolerance and confidence level, giving an "LLN-based sample size" \(N_{\text{LLN}}\).
>    - If \(N_{\text{LLN}}\) is larger than the current N_conv, continue sampling along the same shuffled order until at least \(N_{\text{LLN}}\) configurations have been evaluated; otherwise, stop sampling.
>
> The entire evaluation workflow described above is implemented in an automation scripts. After publication, these scripts will be released as open source, so that practitioners can perform evaluation following the guidance with minimal effort.

---

> ### Author Response · Authors · 2025-12-03
> **# Issue 4: The contribution to causality may be overstated. (Reviewer MRzL weakness 2; Reviewer 5j6t weakness 5)**
>
> Our "causal" contribution lies in  contributing a systematic approach to uncovering causal relationships in complex LLM evaluation systems. Specifically, we construct a minimal evaluation system (MES and A‑MES) that explicitly defines the essential components of evaluation and systematically controls them. This allows us to attribute observed performance differences to specific factors and their interactions, addressing the core challenge that in complex systems “everyone is a stakeholder.”
>
> Existing LLM evaluations do not emphasize the notion of attribution and tend to overlook the impact of confounding factors. In practice, the observed accuracy depends on workload format, prompt methods, decoding parameters, and even system‑level factors. However, current benchmarks typically do not ask whether the measured accuracy is due to the model itself or to these confounders. We make this attribution question explicit and place it at the center of our evaluation design.
>
> To our knowledge, we are the first to define a minimal evaluation system covering 10 key factors and to systematically analyze their contributions to LLM output variance. While our work does not introduce conventional causal inference methods, our work makes a foundational contribution by providing a systematic approach to characterize true causal effects and accurately attribute performance differences in complex LLM evaluation systems. Beyond technical implementation, this framework establishes a principled methodology that moves LLM evaluation beyond single-setting snapshots, enabling more interpretable, reproducible, and comprehensive assessments across diverse conditions. we will clarify this distinction in the revised manuscript to avoid overstating the use of causal language.

---

> ### Author Response · Authors · 2025-12-03
> **# Issue 5: The experimental design has issues in the choice of component value ranges; these settings are rarely used in real deployments, and the near-zero accuracies are likely driven by extreme configurations. (Reviewer 6AVH weakness 1) (1/2)**
>
> We fully agree that unrealistic evaluation settings can easily distort conclusions, so we take this concern seriously. Below we clarify our design choices and the corresponding validation we have performed.
>
> ## 1. Parameter Range Selection
>
> In our actual experiments, we first explored temperature with more fine-grained values between 0 and 1, but when constructing the final MES configuration space we deliberately coarsened this dimension to the three values reported in the paper. The reason is combinatorial: with 10 variables, each additional level per variable multiplies the total number of configurations, quickly making systematic exploration intractable. We therefore treated {0.0, 1.0, 2.0} as a compressed parametrization of a richer underlying search, and verified on the finer-grained runs that the qualitative trends we report are consistent with those obtained at higher resolution. The value ranges of the other parameters were similarly compressed for the same reason.
>
> ## 2. `max_tokens = 10` does not truncate answers in our setup
>
> The reviewer's concern notes that `max_tokens = 10` could truncate answers in typical conversational settings. However, in our non‑CoT evaluation setup, we ask the LLM to use a very compact answer format:
>
> - For multiple-choice questions:
>   `####A####`
> - For numeric fill-in-the-blank:
>   `####342####`
>
> Here, `####` is used as a special delimiter and corresponds to *one token* in our tokenizer; thus a complete answer like `####A####` typically consumes only a **small handful of tokens** (about 3 tokens).
>
> Under this constrained format:
> - `max_tokens = 10` is more than sufficient to generate complete answers for both multiple-choice and fill-in-the-blank questions.
> - In our logs, we do not observe systematic truncation of answers at `max_tokens = 10` for these tasks.
>
> ## 3. `temperature = 2.0` does not universally cause "random" outputs
>
> We agree that for some models, pushing temperature to 2.0 can lead to unstable behavior. However, this is model-dependent and not universally true. For example, consider the following configuration for DeepSeek:
>
> ```text
> {
>   'Language': 'yy',
>   'Question Format': 0,
>   'Question Paraphrase': 0,
>   'Shot': 1,
>   'COT': 0,
>   'Multi Turn': 0,
>   'temperature': 2.0,
>   'top_p': 0.6,
>   'presence_penalty': 0.5,
>   'max_tokens': 100
> }
> ```
>
> Under this configuration, the observed accuracy is 56.67%, which is clearly far above random guessing for AIME problems. This indicates that including `temperature = 2.0` in the evaluation space is not equivalent to injecting invalid runs. Rather, `temperature = 2.0` is challenging but not pathological for the model, and it provides useful information about robustness under aggressive decoding settings.

---

> ### Author Response · Authors · 2025-12-03
> **# Issue 5: The experimental design has issues in the choice of component value ranges; these settings are rarely used in real deployments, and the near-zero accuracies are likely driven by extreme configurations. (Reviewer 6AVH weakness 1) (2/2)**
>
> ## 4. Realistic deployment scenarios for `max_tokens = 10` and `temperature = 2.0`
>
> `max_tokens = 10` does have practical use cases in real deployments, especially when users only need very short responses from the model. By constraining `max_tokens`, applications can cap output length to save time (avoiding long generations when only a brief signal is required) and reduce cost (since pricing is typically proportional to the number of output tokens). Consequently, although `max_tokens = 10` may appear "extreme" from the perspective of open-ended chat, it is a realistic and meaningful setting for short-answer tasks like ours, as well as for many production scenarios that prioritize brevity and efficiency.
>
> Similarly, `temperature = 2.0` is not purely an academic extreme. It is used in creative-generation scenarios such as poetry, fiction, and brainstorming, where diversity and novelty are prioritized. It is also employed for generating unusual phrasing or surprising ideas in exploratory ideation tools, where users explicitly trade reliability for creativity. In such applications, practitioners intentionally set a high temperature to push the model away from generic responses. Therefore, we view `temperature = 2.0` as a realistic configuration for specific use cases, even if it is not ideal for strict QA-type benchmarks. We include such settings to study how sensitive model performance is to decoding extremes, since real deployments often explore a wide range of temperatures across tasks.
>
> ## 5. Why near-zero accuracies are not pathological on AIME'24
>
> AIME problems are extremely difficult, even for human experts. In our author group:
> - With strong mathematical backgrounds, we could solve only about one third of the problems without looking at the official solutions.
> - Even after carefully studying the solution strategies, we still found about 10% of the problems for which we could not fully understand the solution idea.
>
> Given this difficulty, the concentration of mass near zero in the violin plots (Figure 4(a)) primarily reflects the intrinsic hardness of AIME’24, rather than the use of extreme decoding settings.
>
> ## 6. Direct test: removing all `max_tokens = 10` and `temperature = 2.0` runs
>
> To address his concern more directly, we ran an explicit ablation in which we removed all evaluation points with `max_tokens = 10` and all points with `temperature = 2.0`, and then recomputed and replotted the violin plots corresponding to Figure 4(a). The new plots are qualitatively very similar to the originals: the density near zero accuracy is slightly reduced, but the overall shape remains, still with substantial mass close to 0. This indicates that the mass near zero is not primarily caused by these "extreme" settings. Instead, it arises from a combination of the intrinsic difficulty of AIME'24, non‑CoT configurations, less favorable prompts and languages, as well as other factors.
>
> Therefore, excluding `max_tokens = 10` and `temperature = 2.0` does not materially change our main conclusions or the qualitative distributions of accuracies.
>
> In summary, we acknowledge the importance of using realistic and interpretable configurations, and have therefore (i) compressed parameter ranges only after initially exploring finer-grained settings, (ii) carefully designed the answer format so that `max_tokens = 10` is sufficient and non-pathological, (iii) empirically verified that `temperature = 2.0` can yield non-random performance across models, (iv) motivated both `max_tokens = 10` and `temperature = 2.0` with real deployment scenarios, and (v) confirmed via ablation that removing these "extreme" settings leaves the qualitative accuracy distributions essentially unchanged. Together, these analyses support that our MES configuration space, while intentionally broad, does not artificially inflate variance or undermine the reliability of the reported evaluation results.

---

> ### Author Response · Authors · 2025-12-03
> **# Issue 6: Complexity and practical burden on practitioners might be too high (Reviewer 5j6t weakness 3)**
>
> We understand the concern that our MES/A‑MES framework may appear complex or costly. While the initial evaluation exceeds a single-number benchmark, this cost is intrinsic to rigorous attribution: standard evaluations cannot disentangle the effects of question format, prompting, decoding parameters, or other factors.
>
> Controlling cost and ensuring scalability. We do not exhaustively explore all MES configurations; instead, we sample configurations grounded in convergence and law of large numbers stopping criterion, keeping evaluation bounded. Tooling automates configuration space construction and sampling, A‑MES transformations, and ANOVA analysis, so users only need to specify which C‑variables to include, their value ranges, and their evaluation budget. Experiments show that a moderate number of sampled configurations suffices to obtain stable variable-importance rankings.
>
> Practical insights for future use. The initial full evaluatology run identifies high-impact variables (e.g., Question Format, max_tokens) and low-impact ones (e.g., presence_penalty). Subsequent users can focus on the key factors, holding others constant, to capture most insights at a fraction of the cost. In this way, the first run, though more expensive, produces a prioritized, actionable view of the evaluation space, enabling lighter-weight, task-tailored evaluatology afterward.

---

> ### Author Response · Authors · 2025-12-03
> **# Issue 7: Can ANOVA factor importance generalize? (Reviewer 5j6t question2)**
>
> Our empirical results indicate that ANOVA‑based factor importance exhibits both cross‑benchmark regularities and benchmark‑specific characteristics.
>
> On the regularity side, some factors consistently emerge as highly influential across models and benchmarks. For example, in all the benchmarks we analyzed, both Question Format and COT have large and statistically significant effects on accuracy. This suggests that these factors are not idiosyncratic to a single dataset, but rather reflect general sensitivities of current LLMs to how questions are formatted and whether reasoning is explicitly elicited.
>
> On the benchmark‑specific side, we also observe clear differences. A salient example is Question Paraphrase: on AIME, Question Paraphrase has negligible effect, indicating that paraphrasing difficult math problems does not reliably change model accuracy; in contrast, on MMLU, paraphrasing often becomes a key factor— for several models, Question Paraphrase reach statistical significance and explain a non‑trivial share of variance.
>
> In ongoing work, we are extending the ANOVA analysis to more tasks and models to more systematically characterize which factors generalize and which are dataset‑dependent. We will update the paper with these expanded results as soon as they are ready.

---

> ### Author Response · Authors · 2025-12-03
> **# Issue 8: Whether a benchmark should evaluate models by averaging across all possible ones instead of under a representative configuration (Reviewer 6AVH question 5)**
>
> Our position is not that every benchmark must always average over a large configuration space, but that:
>
> 1. Different stakeholders have very different, sometimes conflicting, configuration needs, and
> 2. If we only report performance under a single (even "optimized") configuration, we systematically bias the benchmark toward some users and against others, and we make causal attribution to the model itself unreliable.
>
> ## 1. Why a single "representative" configuration is problematic
>
> In practice, users do not share a single fixed configuration, even for something as simple as temperature:
>
> - Creative writing / brainstorming / ideation.
>   Users typically prefer *higher* temperature (e.g., 0.8–1.0 or even above) to obtain diverse, exploratory outputs.
>
> - Safety‑critical or correctness‑critical use cases (e.g., medical triage support, compliance checks, financial calculations, math competitions).
>   Users usually choose *low* temperature (e.g., 0.0–0.2) and often strict decoding to minimize randomness and hallucinations.
>
> LLMs are deployed to a broad, heterogeneous user base across many domains, not to a homogeneous user who always uses exactly one temperature. If a benchmark fixes the temperature at a single "representative" choice, it implicitly represents some subset of applications at the expense of others. Any single fixed configuration is, unavoidably, a choice of whose use case "counts" as the benchmark.
>
> ## 2. Why averaging over a configuration space is useful and what it means
>
> Our method does not claim that "the true user setting is the uniform distribution over all configurations". Instead, we do two things:
>
> 1. We explicitly define an evaluation condition space EC (Table 2) over indispensable components (Language, Question Format, Shot, COT, temperature, top‑p, etc.), and
> 2. We estimate the expected accuracy under that space by controlled sampling (with convergence checks on the mean and confidence intervals).
>
> This has several motivations:
>
> - Fairness across users.
>     By averaging over multiple plausible configurations, we approximate how a model behaves across a spectrum of realistic use patterns, rather than privileging a single usage style.
>
> - Causal attributions to the model itself.
>     When different models are evaluated under different, partially specified configurations, observed performance differences can conflate intrinsic model quality with arbitrary choices of prompts and decoding hyperparameters. By defining a shared EC space and averaging over it in a controlled manner, we reduce this confounding and obtain measurements that better reflect the effect of the model itself. In addition, by combining this with ANOVA over the EC space, we can quantitatively decompose variance into contributions from the model and from specific components (e.g., Question Format, COT, max tokens), making the causal structure of the evaluation more transparent.
>
> Conceptually, our average is a model of an "evaluation user population": instead of assuming a single fixed user behavior, we define an explicit configuration distribution and report the expected performance under that distribution.
>
> ## 3. Reviewer's example: narrow strong band vs. broad moderate performance
>
> The reviewer raises a key trade-off:
> > Suppose Model A is excellent within a narrow temperature range on a task, while Model B is slightly worse but maintains decent accuracy across a wider range. Which model is "better"?
>
> If stakeholders care about robustness across settings, then they should prefer B when the mean over a broad configuration space is higher.
>
> If they care about peak performance under a tightly controlled configuration, they might prefer A — provided they are willing and able to enforce that precise configuration in deployment.
>
> In other words, there is no universally correct answer to "which is better?"; the evaluation outcome depends on individual preferences, and our methodology is compatible with this fact:
>
> - In our experiments, we use relatively broad ranges for each component in EC. However, in the MES framework and the accompanying tooling we propose, the value ranges of all components in EC are user‑configurable rather than fixed.
> - If a user cares about a **narrow** operating regime (e.g., a tight temperature band, fixed language, fixed format), they can set correspondingly narrow ranges and evaluate models under that restricted EC.
> - If a user instead cares about **broader** behavior across diverse configurations, they can define wider ranges for the same components.
>
> In both cases, the same evaluatology pipeline is applied within the user‑specified EC space, yielding results tailored to their particular preferences.

---

> ### Author Response · Authors · 2025-12-03
> **# Issue 9: What is the single evaluation score after MES/A-MES exploration? (Reviewer 5j6t question 3)**
>
> This is a very valuable question, and we acknowledge that this part was not fully explicit in the paper. Ultimately, practitioners need a single evaluation score per model. In our framework, this is addressed statistically and guided by the intended evaluation goals.
>
> First, the selection of a representative score can be aligned with the stakeholders’ priorities—for example, focusing on specific evaluation dimensions or practical usage scenarios that matter most.
>
> Second, the A‑MES sampling space naturally subsumes the MES space, as each original MES instance is included as one of the possible augmentations. By sampling from the broader A‑MES space, we effectively cover a wider range of configurations and potential use cases, while still maintaining a non-zero probability of evaluating the original MES settings.
>
> The final reported evaluation score for each model is then the mean performance over the sampled instances, together with 95% and 99% confidence intervals, providing a stable summary metric that balances comprehensiveness with practical efficiency.

---

> ### Author Response · Authors · 2025-12-03
> **# Issue 10: Why question paraphrase in Table 2 is binary (Reviewer MRzL question 3)**
>
> The "Question Paraphrase" column in Table 2 is binary by design, which does not imply that there is only a single way to paraphrase a question. The binary flag distinguishes two regimes:
>
> - No (original): the original question text is used, which may appear in a model's training data.
> - Yes (paraphrased): the question is restated in different wording while preserving all underlying semantics, answers, numerical values, and natural language constraints, avoiding potential data‑contamination issues.
>
> To construct paraphrases, we first compare several candidate LLMs on a small validation set and select the one that reliably follows these constraints. This model is then instructed to restate each question, ensuring semantic equivalence while changing surface wording. The purpose of this binary indicator is to diagnose memorization and data‑contamination effects: comparing model performance on original versus paraphrased items reveals whether behavior is robust to rewording or overly tied to specific training instances. The specific paraphrasing strategy or number of rewrites is irrelevant for this goal.
>
> Within MES, all workload transformations—including Question Paraphrase—preserve intrinsic semantics, providing alternative surface realizations of the same task. By contrast, A‑MES analogical transformations (distractor insertion, numeric substitution, condition recomposition) alter problem structure while maintaining core reasoning patterns, producing a family of structurally varied, more challenging variants.Therefore, keeping Question Paraphrase as a binary indicator cleanly separates same‑semantics rewording from the structural, changed‑semantics regime in A‑MES, aligning the design with our evaluation objectives.

---

> ### Author Response · Authors · 2025-12-03
> **# Issue 11: Motivation and construction of A‑MES and definition of out‑of‑distribution workloads (Reviewer 6AVH weakness 2) (1/3)**
>
> The reviewer was absolutely right that our current example (adding an irrelevant sentence) reads like a robustness or adversarial‑style perturbation. Our intention, however, was not to claim that this alone constitutes analogical reasoning; rather, A‑MES is designed to cover a broader family of workload shifts, including robustness to realistic "noise" in user queries.
>
> In real‑world usage, users rarely submit clean, minimal prompts; they often add story background, opinions, meta‑commentary, or partially incorrect intuitions. Users still expect the model to solve the core task correctly despite such "perturbations". From this perspective, introducing redundant or misleading sentences is not artificial but faithfully reflects actual deployment conditions. Evaluating robustness to these perturbations is therefore an essential part of "causally faithful" evaluation.
>
> More importantly, our redundancy insertion is not a single ad‑hoc edit; it is implemented via three explicit, controllable categories, and all instances are generated systematically via LLM prompting:
>
> 1. Context‑irrelevant redundancy: sentences completely unrelated to the problem.
>    *Example:*
>    > *The weather today seems quite pleasant, and it might be a great day for a picnic.* Find the number of triples of nonnegative integers $(a,b,c)$ satisfying $a + b + c = 300$ and \[a^2b + a^2c + b^2a + b^2c + c^2a + c^2b = 6,000,000.\]
>
>    Here, the weather is entirely unrelated to the math content.
>
> 2. Context‑relevant, explanatory redundancy: additional sentences that explain concepts already present in the question (semantics and solution unchanged).
>    *Example:*
>    > There exist real numbers $x$ and $y$, both greater than 1, such that $\log_x\left(y^x\right)=\log_y\left(x^{4y}\right)=10$. *A logarithm is a way to express how many times a base must be multiplied by itself to get a certain number*. Find $xy$.
>
>    The added sentence explains the notion of a logarithm while leaving the underlying problem unchanged.
>
> 3. Context‑relevant, misleading redundancy: sentences that are thematically related but subtly promote an incorrect heuristic.
>    *Example:*
>    > Alice and Bob play the following game. A stack of $n$ tokens lies before them. The players take turns with Alice going first. On each turn, the player removes either $1$ token or $4$ tokens from the stack. *Many players adopt a greedy approach here: always take $4$ whenever possible to shorten the game and restrict the opponent's replies.* Whoever removes the last token wins. Find the number of positive integers $n$ less than or equal to $2024$ for which there exists a strategy for Bob that guarantees that Bob will win the game regardless of Alice's play.
>
>     The extra sentence about the "greedy approach" is logically related to the game but suggests a flawed strategy, intentionally nudging the solver toward an incorrect line of reasoning
>
> We also analyzed these categories separately and observed an interesting pattern: context‑irrelevant redundancy tends to cause the largest drop in model accuracy, whereas even deliberately misleading context‑relevant redundancy has a smaller impact on accuracy than context‑irrelevant noise.

---

> ### Author Response · Authors · 2025-12-03
> **# Issue 11: Motivation and construction of A‑MES and definition of out‑of‑distribution workloads (Reviewer 6AVH weakness 2) (2/3)**
>
> Distractor insertion is only one type of analogical transformation we use (with an emphasis on robustness). In addition, we include:
>
> - **Numeric substitution:** we systematically change the numerical constants in the problem while keeping the solution method and underlying structure intact.
>     *Example:*
>     - **Original:** Find the largest possible real part of \[(75+117i)z + \frac{96+144i}{z}\] where $z$ is a complex number with $|z|=4$. A common shortcut is to take $z$ to be a positive real number, since for a fixed modulus the real part is often largest when the argument of $z$ is zero.
>     - **Numeric variant:** Find the largest possible real part of \[(100+112i)z + \frac{60+144i}{z}\] where $z$ is a complex number with $|z|=4$. A common shortcut is to take $z$ to be a positive real number, since for a fixed modulus the real part is often largest when the argument of $z$ is zero.
>
> - **Conditional recomposition:** we change which quantity is treated as the "target" and which appears as a given condition, while preserving the same geometric or algebraic relationships.
>     *Example:*
>     - **Original:**
>     > Rectangles $ABCD$ and $EFGH$ are drawn such that $D,E,C,F$ are collinear. Also, $A,D,H,G$ all lie on a circle. If $BC=16$,$AB=107$,$FG=17$, and $EF=184$, what is the length of $CE$?
>     - **Conditional recomposition:**
>     > Rectangles $ABCD$ and $EFGH$ are drawn such that $D,E,C,F$ are collinear. Also, $A,D,H,G$ all lie on a circle. If $BC=16$,$AB=107$,$CE=104$, and $EF=184$, what is the length of $FG$?
>
> To summarize, in our framework "analogical transformations" are intended to capture a class of reasoning‑preserving modifications: they may change the numerical values or which variable is queried, and may introduce realistic redundancy, but they keep the underlying solution strategy essentially unchanged. This is exactly what we aim to probe—whether the model has internalized the reasoning pattern rather than merely memorized a specific benchmark instance.

---

> ### Author Response · Authors · 2025-12-03
> **# Issue 11: Motivation and construction of A‑MES and definition of out‑of‑distribution workloads (Reviewer 6AVH weakness 2) (3/3)**
>
> ## 2. On the use of "out‑of‑distribution" (OOD)
>
> We apologize for the confusion caused by our terminology. Our intention was not to redefine OOD in a way that conflicts with its standard usage in the ML literature. In our setting, we conceptually distinguish three categories of workloads:
>
> 1. Seen / potentially memorized: tasks that may have been present (or nearly present) in training.
> 2. Unseen but structurally similar: tasks not memorized verbatim, but solvable by applying previously learned patterns or transformations.
> 3. Completely novel: tasks that are unlikely to appear in the training corpus and thus probe more genuine generalization.
>
> In the paper, we loosely referred to category (3) as "out‑of‑distribution". Thanks to his comment, we now see that calling this "OOD" can be misleading, since OOD in the general literature often refers to a well‑characterized distributional shift (e.g., different domain, style, or covariates), rather than simply "unseen exam problems". We will adopt a more precise term for this category in the revised version.

---

> ### Author Response · Authors · 2025-12-03
> **# Issue 12: Lack of a clear definition for the term "LLM workload" (Reviewer 6AVH question 3)**
>
> Our use of "workload" is indeed inspired by the computer architecture community. In our setting, a "workload" corresponds to a question within a benchmark that needs to be solved by the LLM, while a "task" has a closely related but slightly more general meaning, referring for example to a broader type of question or capability. However, we agree that the term "workload" is not yet standardized in the LLM community, and that both the term itself and its connection to CPU benchmarking are not introduced early enough in the paper, so the current presentation will be made clearer in the revision.

---

> ### Author Response · Authors · 2025-12-03
> **# Issue 13: Is 500 random sampling enough? (Reviewer MRzL question 4)**
>
> For each benchmark, we first construct a configuration space of 15,552 distinct settings defined by ten controllable variables (e.g., Language, Question Format, Question Paraphrase, Shot, Chain-of-Thought, Multi-Turn, temperature, top_p, presence_penalty, and max_tokens). We then generate workload-level variants for each setting using the seven augmentation mechanisms, producing a ~100,000-point configuration space. To implement random sampling while ensuring reproducibility and consistency across model evaluations, we randomly shuffle the diverse settings using a random seed to form a list and select the first <sample_size> configurations.
>
> This sample size choice is not arbitrary: we first determine per‑model sample sizes using an explicit convergence procedure, and then additionally validate them with an LLN‑based calculation. The final "500" threshold is thus a conservative upper cap backed by two independent criteria rather than a heuristic guess.
>
> Concretely, when testing each model on each workload, we process configurations in batches of 10. After every 10 samples, we compute the running mean accuracy and its 95% confidence interval, and we stop sampling once both of the following conditions are satisfied:
>
> 1. The absolute changes in the running mean accuracy over the last three consecutive updates are all smaller than 0.002; that is, the absolute differences between the last and second‑last, the second‑last and third‑last, and the third‑last and fourth‑last running means are each smaller than 0.002.
> 2. The length of the 95% confidence interval is smaller than 0.06.
>
> Using this rule, the sample sizes at which different models converged were:
>
> - GPT‑4.1: 260
> - GPT‑3.5: 220
> - Mistral Medium: 260
> - Mistral Large: 290
> - Qwen Plus: 260
> - Qwen2.5: 400
> - DeepSeek‑V3: 320
> - Doubao‑1.5‑pro‑32k: 480
> - Moonshot‑v1: 220
>
> To further validate the above convergence-based sample sizes, we used a Law of Large Numbers (LLN)–based estimation. For each model, we first used the observed accuracies on the sampled configurations to estimated the variance of accuracy, and then computed the minimal sample size required to ensure that the sample mean is within a small error tolerance of the true mean with high probability. This gave the following estimates:
>
> - GPT‑4.1: 170
> - GPT‑3.5: 88
> - Mistral Medium: 229
> - Mistral Large: 253
> - Qwen Plus: 218
> - Qwen2.5: 329
> - DeepSeek‑V3: 285
> - Doubao‑1.5‑pro‑32k: 421
> - Moonshot‑v1: 63
>
> For every model, the LLN‑estimated minimal sample size is smaller than the sample size returned by the convergence method above. Therefore, we take the largest convergence‑based sample size across models as a baseline and add an additional safety margin, setting a global cap of 500 configurations for MES in Section 5.1, which also helps maintain cross‑model comparability in our evaluation.

---

> ### Author Response · Authors · 2025-12-03
> **# Issue 14: Typographical errors in the paper (Reviewer fLKa weakness 3; Reviewer 6AVH question 1)**
>
> In Table 3, the accuracy of gpt3.5 was mistakenly written as 1.1 instead of 0.1, and Figure 4(a) contained a redundant legend. These issues were carefully identified by the reviewers; we have corrected them and re-checked the paper to ensure that similar problems do not remain.

---

> ### Author Response · Authors · 2025-12-03
>
> We hope that these clarifications and revisions make the contributions of our work clearer and demonstrate that the reviewers' concerns have been carefully considered and resolved.

---

### Meta-Review · Area_Chair_tkMw · 2026-01-07

**Summary:**

This paper addresses an important question, but the authors seem to be unaware of much of the existing literature on the same topic. Perhaps most relevant, Biderman et al., 2024 (https://arxiv.org/abs/2405.14782) highlights the importance of the evaluation setup, including specific prompts, prompt formatting (shots, COT), and hyperparameters (temperature, top_p, presence, max_tokens), most of the elements identified in the Minimum Evaluation System in this work. Biderman et al. released an evaluation harness which enables their implementation to be used which was widely adopted by the community (it backed the HuggingFace leaderboard), and the same principles were adopted into the AISI Inspect Harness which has largely replaced it (https://inspect.aisi.org.uk). Similar recommendations can be found in Reuel et al., 2025 (https://arxiv.org/abs/2411.12990). Both of these also recommend code release for truly repeatable results, which is absent here.

Most, and likely all, of the specific elements used here have also been implemented previously (e.g. distractors and number changes in Mirzadeh et al., 2024 https://arxiv.org/html/2410.05229v1, temperature in Song et al., 2024 https://arxiv.org/html/2407.10457v1, multilingual evaluation in Singh et al., 2025 https://aclanthology.org/2025.acl-long.919/).

The literature review and conceptualization of benchmarks and "workflows" would also benefit from drawing on existing work. Raji et al. 2021 (https://arxiv.org/abs/2111.15366) introduced a widely-used division of benchmarks into datasets and metrics, (or more broadly "task" and "metric" in the concurrent Bean et al. 2025 https://arxiv.org/abs/2511.04703), which is more complete than the "workload" terminology used here. (Bean et al. also provides a more comprehensive and principled taxonomy of benchmark types which might be used in the related work here.)

The primary novel contribution of this work is then the empirical analysis of all of these factors together, and the ANOVA attribution. Sweeping across the ten listed dimensions is a novel approach, but doesn't necessarily make sense. As the authors note, the costs of conducting the complete evaluation are prohibitive, so a subset needs to be chosen. However, several of the hyperparameter options tested are unusual (max_tokens=10, Shot as a binary, temperature=2.0) and generally it makes the most sense to set these parameters in line with how the model is expected to be deployed (if I don't speak Russian, I don't want Russian translations in my eval set). I imagine that the bimodal distributions of scores on MES reflects the poor performance of the models in unusual setups. This aggressive testing could make sense for robustness testing, but does not inherently make for a better evaluation.

Notes:
There is a lot of good thinking in this paper, but its pretty much all been done before without being cited. It feels like a paper written by capable authors outside of their own knowledge area.

**Reviewer Concerns:**

see "Summary" text

**Reviewer Scores:**

see "Summary" text

---

### Decision · Program_Chairs · 2026-01-26

Reject